 

# Intergenerational transport of double-stranded RNA in *C. elegans* can limit heritable epigenetic changes

Nathan M Shugarts Devanapally, Aishwarya Sathya, Andrew L Yi, Winnie M Chan, Julia A Marre, Antony M Jose*

Department of Cell Biology and Molecular Genetics, University of Maryland, College Park, United States

## eLife Assessment

In this report, the authors present **valuable** findings identifying a novel worm-specific protein (sdg-1) that is induced upon loss of dsRNA import via SID-1, but is not required to mediate SID-1 RNA regulatory effects. The genetic and genomic approaches are well-executed and the revision contains generally **solid** support for the central findings of the work. These findings will be of interest to those working in the germline epigenetic inheritance field.

***For correspondence:**
amjose@umd.edu

**Competing interest:** The authors declare that no competing interests exist.

## Abstract

RNAs in circulation carry sequence-specific regulatory information between cells in plant, animal, and host-pathogen systems. Such RNA can cross generational boundaries, as evidenced by somatic double-stranded RNA (dsRNA) in the nematode *Caenorhabditis elegans* silencing genes of matching sequence in progeny. Here we dissect the intergenerational path taken by dsRNA from parental circulation and discover that cytosolic import through the dsRNA importer SID-1 in the parental germline and/or developing progeny varies with developmental time and dsRNA substrates. Loss of SID-1 enhances initiation of heritable RNA silencing within the germline and causes changes in the expression of the **s**id-*1*-**d**ependent **g**ene *sdg-1* that last for more than 100 generations after restoration of SID-1. The SDG-1 protein is enriched in perinuclear germ granules required for heritable RNA silencing but is expressed from a retrotransposon targeted by such silencing. This auto-inhibitory loop suggests how retrotransposons could persist by hosting genes that regulate their own silencing.

## Introduction

RNAs released into circulation can act as intercellular messages that are used for gene regulation in distant cells. Examples include secretion of small RNAs within exosomes in response to pathogenic fungal infection in *Arabidopsis* (*Cai et al., 2018*), virus-like proteins with their coding mRNAs in developing *Drosophila* (*Ashley et al., 2018*) and mice (*Pastuzyn et al., 2018*), microRNAs from adipose tissue in mice (*Thomou et al., 2017*) and small RNAs from the epididymis in mice (*Sharma et al., 2016*; *Chen et al., 2016*; *Conine et al., 2018*; *Sharma et al., 2018*). Such extracellular RNAs have also been detected in humans, but their roles in gene regulation remain unclear despite their use as diagnostic markers for diseases (reviewed in *Das et al., 2019*). Furthermore, the recent development of double-stranded RNA (dsRNA)-based drugs (reviewed in *Setten et al., 2019*; *Hu et al., 2020*; *Mullard, 2022*) that can silence genes of matching sequence through RNA interference (*Fire et al., 1998*) has heightened interest in understanding the import of dsRNA into cells. A conserved dsRNA-selective importer, SID-1 (*Winston et al., 2002*; *Feinberg and Hunter, 2003*; *Shih et al., 2009*), is

required for the import of extracellular dsRNA into the cytosol of any cell in the nematode *Caenorhabditis elegans*. This entry into the cytosol is distinct from the uptake of dsRNA into cells, which can rely on other receptors (e.g. SID-2 for uptake from the intestinal lumen *Winston et al., 2007*; *McEwan et al., 2012*). SID-1 has two homologs in mammals – SIDT1 and SIDT2. Although similar cytosolic entry of dsRNA using these mammalian homologs of SID-1 is supported by studies in mice reporting entry of viral dsRNA through SIDT2 (*Nguyen et al., 2017*), enhanced dsRNA uptake when SIDT1 is overexpressed in vitro (*Nguyen et al., 2019*), and uptake of ingested dsRNA into cells through SIDT1 (*Chen et al., 2021*), alternative roles for SIDT1 and/or SIDT2 in the uptake of cholesterol have also been proposed (*Méndez-Acevedo et al., 2017*).

Secretion of dsRNA from *C. elegans* tissues expressing dsRNA from transgenes has been inferred based upon the SID-1-dependent silencing of matching genes in other tissues (*Winston et al., 2002*; *Jose et al., 2009*). Secreted dsRNA from neurons can silence genes of matching sequence in most somatic cells (*Ravikumar et al., 2019*) and within the germline (*Devanapally et al., 2015*). Extracellular dsRNA delivered into parental circulation by injection or ingestion also enters the germline and can cause silencing of matching genes in progeny (*Fire et al., 1998*; *Timmons and Fire, 1998*; *Grishok et al., 2000*; *Marré et al., 2016*; *Wang and Hunter, 2017*). In every case, the entry of dsRNA into the cytosol dictates when and where the processing of extracellular dsRNA can begin. Such intergenerational transport of RNA is an attractive mechanism for explaining endogenous, gene-specific effects in progeny that could occur in response to changes in somatic tissues of parents. However, which conditions induce transport of dsRNA into the germline, when during development this transport occurs, and what regulatory consequences ensue in progeny upon uptake of extracellular dsRNA from parents are all unknown. Despite this lack of knowledge, the analysis of transgenerational gene silencing triggered by dsRNA has revealed that a class of small RNAs called 22G RNA made using the mRNA targeted by dsRNA and bound by Argonaute proteins in the germline (*Buckley et al., 2012*; *Xu et al., 2018*) is necessary for observing silencing in every generation.

Timed dsRNA injection into animals or timed dsRNA ingestion by animals has been the only way to study mechanisms of dsRNA transport throughout *C. elegans* development since no tools to induce dsRNA secretion from cells have been developed. Injection of dsRNA into adult *sid-1(-)* animals demonstrated that extracellular dsRNA can be directly transmitted to progeny without entry into the cytosol (*Marré et al., 2016*; *Wang and Hunter, 2017*). This intergenerational transmission of dsRNA in *sid-1(-)* adult animals requires the yolk receptor RME-2 (*Grant and Hirsh, 1999*) and is independent of parental 22G RNA production because dsRNA cannot enter the cytosol in the parent. In further support of the initial intergenerational transport of extracellular dsRNA or dsRNA-derived signals not requiring 22G RNAs, *gfp*-dsRNA from parents that lack *gfp* sequences are transported to progeny (Figure 1E in *Marré et al., 2016*). Thus, the silencing signals transported from parent to progeny include the extracellular dsRNA and its derived silencing signals, independent of 22G RNAs. However, 22G RNAs are required for silencing and they may be used for intergenerational transport in subsequent generations.

All dsRNAs, regardless of length, have been assumed to be equivalent substrates for entry into the cytosol. This assumption is supported by the uptake of a variety of dsRNA substrates when *C. elegans* SID-1 is overexpressed in heterologous *Drosophila* S2 cells (*Shih et al., 2009*). Yet, two key observations suggest that dsRNA can take multiple routes in vivo before SID-1-dependent entry into the cytosol. One, even in the presence of SID-1, dsRNA ingested during early adulthood requires RME-2 to cause silencing in progeny (*Marré et al., 2016*). Two, 50 bp fluorescently labeled dsRNA requires RME-2 for entry from parental circulation into oocytes (*Marré et al., 2016*). Establishing the contexts for the use of different modes of dsRNA transport is crucial for understanding the processes regulated by endogenous extracellular dsRNA.

Here, we dissect the intergenerational transport of extracellular dsRNA and discover a role for this mechanism in modulating RNA regulation within the germline. Extracellular dsRNA is transported with developmental and substrate specificity from parental circulation to progeny, and its release from neurons can be enhanced using light-induced oxidative damage. Blocking dsRNA import into the cytosol of all cells revealed heritable changes in gene expression and led to the identification of **s**id-1-**d**ependent **g**ene-1 (*sdg-1*). The *sdg-1* coding sequence is located within a retrotransposon that is targeted by RNA silencing in the germline. Yet, the SDG-1 protein colocalizes with regulators of RNA silencing in perinuclear granules within the germline and dynamically enters the nucleus in

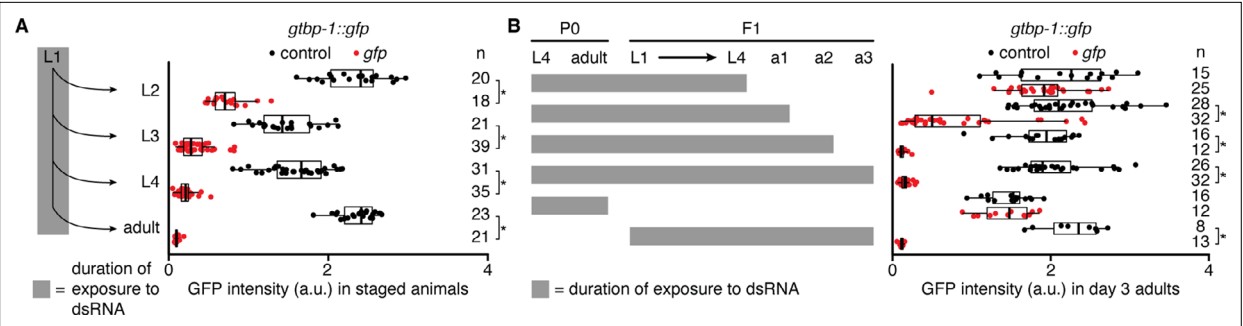

**Figure 1.** Gene silencing by ingested dsRNA during larval development can fail to persist into adulthood. Silencing in the germline was measured after exposure of *gtbp-1::gfp* animals to bacteria expressing dsRNA by imaging separate cohorts at indicated stages (**A**) or day 3 of adulthood (**B**). (**A** and **B**), *left* Schematics depicting stages and durations of exposure to dsRNA. (**A** and **B**), *right* GFP intensity (a.u.) in *gtbp-1::gfp* animals quantified in germ cells (larvae) or eggs in utero (adults) (**A**) or in day 3 adult (a3) animals (**B**) after exposure to control dsRNA (black) or *gfp*-dsRNA (red). The numbers of animals scored at each stage (n) are indicated. Asterisks indicate p<0.05 with Bonferroni correction using Mann-Whitney U test for two-sided comparisons between animals exposed to control dsRNA or *gfp*-dsRNA. Also see *Figure 1—figure supplement 1*.

The online version of this article includes the following figure supplement(s) for figure 1:

**Figure supplement 1.** Uptake of dsRNA into the proximal germline by RME-2 is required for silencing during early adulthood.

proximal oocytes and in cells of developing embryos. Measurements of *sdg-1* expression using native mRNA, a translational reporter, or a transcriptional reporter reveal that expression is easily perturbed in different mutants that impact dsRNA-mediated gene regulation. Expression varies between the two gonad arms of wild-type animals, and different mutant isolates can show an increase or decrease in expression, indicative of a loss of buffered gene expression within the germline. However, consistent with an overall role for SDG-1 (and potentially other SDGs) in promoting RNA silencing, either loss of SID-1 or overexpression of SDG-1 enhances piRNA-mediated silencing within the germline initiated by mating. Therefore, we propose that the import of extracellular dsRNA into the germline tunes intracellular pathways that cause heritable RNA silencing.

## Results

### Requirements for the entry of extracellular dsRNA into the germline change during development

A convenient method for the delivery of extracellular dsRNA into *C. elegans* at various times during larval development is the expression of dsRNA in the bacteria that the animals ingest as food (*Timmons and Fire, 1998*). To determine when ingested dsRNA can enter the germline and cause silencing, we exposed developing animals with a ubiquitously expressed protein (GTBP-1) tagged with GFP to bacteria that express *gfp*-dsRNA. Silencing was detectable within the germline from the second larval stage (L2) onwards (*Figure 1A*, *Figure 1—figure supplement 1A*), but either exposure to ingested dsRNA beyond the fourth larval stage (L4) (*Figure 1B*) or injection of dsRNA into the 1-day-old adult germline (*Figure 1—figure supplement 1B*) was required to observe silencing in the germline of 3-day-old adults. Combined with the need for exposure to dsRNA after the L4 stage (*Marré et al., 2016*; *Wang and Hunter, 2017*) for silencing in progeny, even for just 24 hr (*Figure 1—figure supplement 1C*), these observations suggest that detectable silencing within the germline is possible during early development without initiating heritable RNA silencing. One possible explanation for this observation could be that both RNAs derived from the imported dsRNA and downstream silencing signals are continually diluted by the proliferation of germ cells. Heritable silencing by dsRNA ingested from the L4 stage to the first day of adulthood likely relies on entry of dsRNA into the proximal germline because silencing of a somatic gene in progeny after parental ingestion of dsRNA during this period required RME-2 (*Figure 1—figure supplement 1C*), which is enriched in the proximal germline (*Figure 1—figure supplement 1D*; *Grant and Hirsh, 1999*), and some *gtbp-1::gfp* animals exposed to *gfp*-dsRNA up to the first day of adulthood showed more silencing in the proximal germline (*Figure 1—figure supplement 1E*).

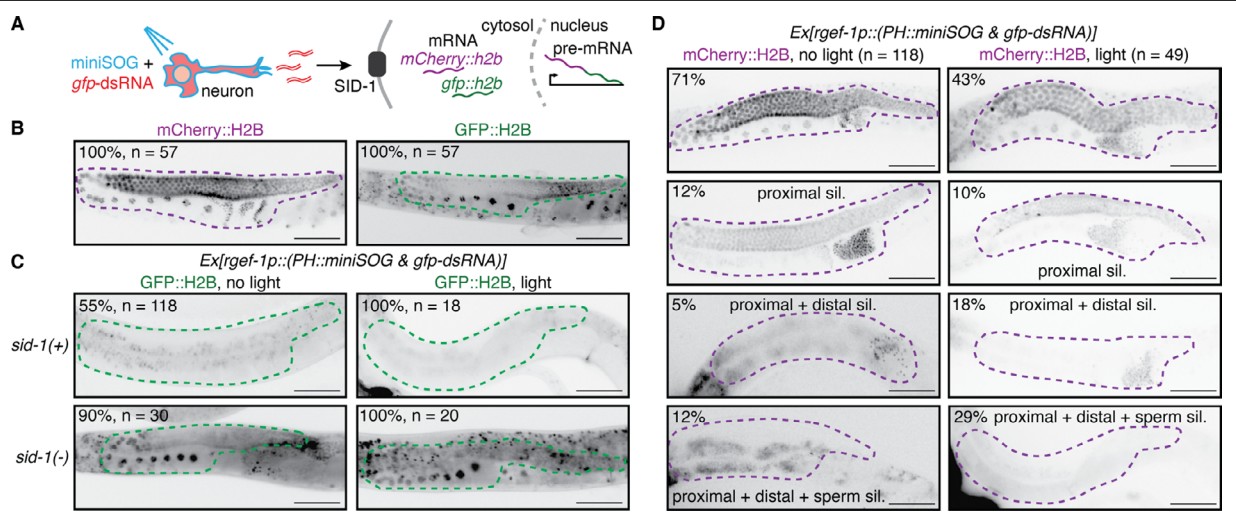

**Figure 2.** Oxidative damage of neurons enhances gene silencing by neuronal dsRNA in the adult germline. (**A**) Schematic illustrating exposure to blue light of animals expressing a singlet oxygen generator (miniSOG) and *gfp*-dsRNA in neurons, and subsequent release of dsRNA. Such extracellular dsRNA is expected to enter the cytosol of the germline through the dsRNA importer SID-1 and silence *gfp::h2b* mRNA from a two-gene operon that expresses *mCherry::h2b* and *gfp::h2b* as part of a single pre-mRNA. (**B–D**) Images of single gonad arms in adult animals with the two-gene operon (*mex-5p::mCherry::h2b::gfp::h2b*) showing fluorescence (black) of mCherry::H2B (magenta outline) or of GFP::H2B (green outline). Punctate autofluorescence from the intestine can also be seen. Numbers of animals assayed (**n**) and percentages of adult animals with the depicted expression patterns are indicated. Scale bars, 50 µm. (**B**) mCherry::H2B fluorescence is seen throughout the germline (*left*) and GFP::H2B fluorescence is seen in the oocytes and in the distal gonad (*right*). (**C**) GFP::H2B fluorescence in *sid-1(+)* and *sid-1(-)* animals expressing membrane-localized miniSOG (PH::miniSOG) and *gfp*-dsRNA driven by a neuronal promoter (*rgef-1p*) from a multi-copy transgene (*Ex*, *jamEx214*) without (*left*) or with (*right*) exposure to blue light at 48 hr post L4-stage of parent. (**D**) mCherry::H2B fluorescence in *sid-1(+)* animals with the transgene *Ex*. Silencing of mCherry is enhanced in the distal gonad (third row) and sperm (fourth row) after exposing animals to blue light at 48 hr and 54 hr post L4-stage of parent. By region, silencing after exposure to light (*right*) in the proximal germline (57%=10 + 18+29)>distal germline (47%=18 + 29)>sperm (29%). Also see ***Figure 2—figure supplements 1 and 2***.

The online version of this article includes the following figure supplement(s) for figure 2:

**Figure supplement 1.** Timed release of neuronal dsRNA by oxidative damage in neurons reveals period of enhanced gene silencing in the soma and germline.

**Figure supplement 2.** Schematics depicting mutations generated in this study.

Thus, these results reveal three periods of germline development that can be broadly distinguished based on the response to ingested dsRNA: (1) from the first larval to the fourth larval stage when exposure to dsRNA does not result in maximal silencing within the germline in adults (***Figure 1B***); (2) from the fourth larval stage to early adulthood when entry of dsRNA primarily occurs in the proximal germline through RME-2 (***Figure 1—figure supplement 1C and E***); and (3) later adulthood when germline silencing by ingested dsRNA is maximal (***Figure 1B***) and ingested dsRNA can effectively silence progeny independent of RME-2 (***Figure 1—figure supplement 1C***; ***Wang and Hunter, 2017***).

## Oxidative damage in neurons expressing dsRNA enhances silencing in the germline by neuronal dsRNA

Another approach for delivering extracellular dsRNA into the germline that better mimics dsRNA transport between cells is the secretion of dsRNA from neurons (***Devanapally et al., 2015***). However, the extent of such secretion throughout development is unpredictable. To modulate the secretion of dsRNA from somatic cells into parental circulation during development, we adapted an optogenetic approach for damaging somatic cells (***Xu and Chisholm, 2016***). Specifically, we generated animals that express the mini singlet oxygen generator (miniSOG) protein in neurons and exposed them to blue light. While animals expressing miniSOG from a single-copy transgene did not show an appreciable defect when compared with wild-type animals, those expressing miniSOG from a multi-copy transgene were paralyzed (***Figure 2—figure supplement 1A and B***, *top*) and had visibly damaged neurons (***Figure 2—figure supplement 1B***, *bottom*). Using this system, we induced oxidative damage in the neurons of animals that expressed dsRNA under the control of a neuronal promoter and evaluated

silencing of target genes with matching sequence expressed in other tissues (*Figure 2A*). By exposing animals to blue light for 60 min at different times during development (*Figure 2—figure supplement 1C*), we observed SID-1-dependent enhancement in the silencing of the hypodermal gene *bli-1* at the adult stage by neuronal *bli-1*-dsRNA, with maximal silencing when oxidative damage occurred during mid-to-late larval development (*Figure 2—figure supplement 1D*, light exposure every 6 hr from 42 to 66 hr post L4-stage of parent; *Figure 2—figure supplement 1E*, ~two-fold increase from 14.9% to 29.1% in a background with **e**nhanced **R**NA **i**nterference (*eri-1(-)*) and ~six-fold increase from ~1.6% to ~9.8% in a wild-type background). A similar period of maximal SID-1-dependent enhancement of silencing was also observed when neurons expressing *gfp*-dsRNA were damaged and silencing of a two-gene operon that expresses two fluorescent proteins, mCherry::H2B and GFP::H2B, in the germline was measured (*Figure 2B-D*, *Figure 2—figure supplement 1F* 48–60 hr post L4-stage of parent; *sid-1(-)* allele (*jam80[nonsense]*) is depicted in *Figure 2—figure supplement 2*). While silencing of *gfp::h2b* was observed throughout the germline, silencing of the other cistron *mCherry::h2b* was also observed sometimes, albeit restricted to specific regions of the germline. Silencing of *mCherry::h2b* was most frequent in the proximal germline and was not observed in any other region without silencing in the proximal germline (proximal germline – 57%, distal germline – 47%, sperm – 29%, *Figure 2D*), likely due to reduction of *mCherry::h2b::gfp::h2b* pre-mRNA (*Guang et al., 2008*) in those regions. Consistently, the silencing of both *gfp::h2b* and *mCherry::h2b* was eliminated in the absence of the nuclear Argonaute HRDE-1 (*Figure 2—figure supplement 1G*). The pattern of *mCherry::h2b* silencing is similar to the spatial pattern observed for the RME-2-dependent entry of dsRNA delivered into parental circulation (*Marré et al., 2016*) and is consistent with the pattern of target mRNA degradation in the germline by extracellular dsRNA (*Ouyang et al., 2022*). However, silencing of *gfp::h2b* in the germline by neuronal dsRNA did not show a detectable dependence on RME-2 (*Figure 2—figure supplement 1H* - difficulty in obtaining transgenic animals that also lack RME-2 resulted in a low sample size for this experiment).

Thus, by modulating the secretion of dsRNA from somatic cells for the first time, we gained two insights into the intercellular transport of dsRNA: (1) oxidative damage of neurons during particular periods in development increases the amount of dsRNA and/or changes the kinds of dsRNA in circulation either because of specific enhancement of secretion or nonspecific spillage; and (2) there is a preference for silencing by neuronal dsRNA in the proximal germline.

## Extracellular dsRNA in parental circulation can be transported through multiple routes to cause silencing in progeny

While the characteristics of extracellular dsRNA imported into the germline from ingested bacteria or from neurons are unknown, delivery of chemically defined dsRNA into the extracellular space in *C. elegans* can be accomplished using microinjection (*Fire et al., 1998*; *Marré et al., 2016*). We examined differences, if any, in the entry of in vitro transcribed dsRNA from the extracellular body cavity into the germline during the L4 and adult stages as evidenced by silencing in the progeny of injected animals. Silencing was comparable regardless of whether wild-type or *rme-2(-)* parents were injected as L4-staged or adult animals (*Figure 3A*, *Figure 3—figure supplement 1A*, *left*; also reported for adults in *Wang and Hunter, 2017*). However, some dependence on RME-2 for silencing in progeny was discernable when lower concentrations of dsRNA were used (*Figure 3—figure supplement 1A*, *right*). This result and previous results demonstrating proximal germline-specific silencing in contexts where silencing is less robust (*Figure 2D*, *Figure 1—figure supplement 1E*) are consistent with RME-2-dependent uptake of dsRNA being a route for extracellular dsRNA to enter the germline. The difference in parental RME-2 requirement for silencing in progeny observed for dsRNA ingested (*Figure 1—figure supplement 1C*) or injected (*Figure 3—figure supplement 1A*) at the L4 stage could similarly reflect the accumulation of different amounts of dsRNA in parental circulation (e.g. more upon injection than upon ingestion), and/or different kinds of dsRNA (e.g. because of modifications in bacteria or upon transit through the intestine). However, these possibilities could not be easily distinguished because sensitive northern blotting (*Choi et al., 2017*) revealed that both bacterial and in vitro transcribed dsRNA consist of a complex mix of dsRNAs (*Figure 3—figure supplement 1B–1D*; consistent with *Tabara et al., 2002* and *Jain et al., 2020*), hereafter called mixed dsRNA.

In contrast, when synthesized *gfp*-dsRNA of a defined length (50 bp) with a fluorescent label was injected into circulation in adult animals, no entry into the germline was observed in the absence of

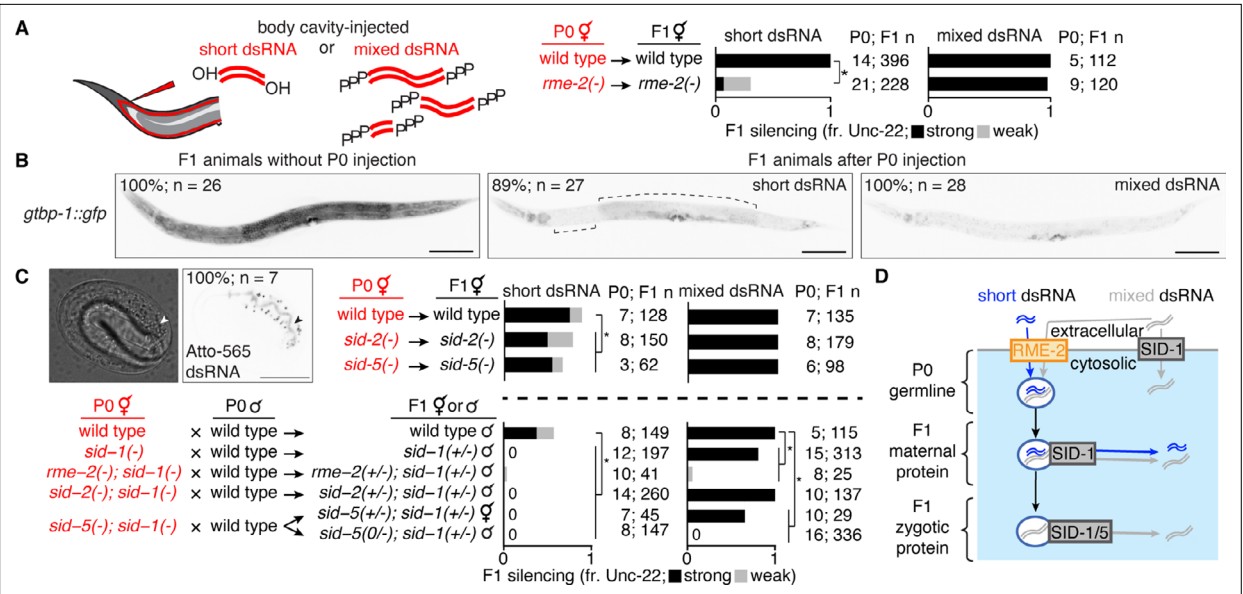

**Figure 3.** Transport of dsRNA from parental circulation to progeny occurs through two routes with distinct substrate selectivity. (**A**) Hermaphrodite animals of indicated genotypes (in red) were injected in the body cavity with 50 bp *unc-22*-dsRNA synthesized with a 5'-OH (short dsRNA, left bars) or *unc-22*-dsRNA with a 5' triphosphate transcribed from a ~1.1 kb template (mixed dsRNA, right bars). Hermaphrodite self-progeny of injected animals were scored for *unc-22* silencing (fr. Unc-22: strong, black; weak, grey). Numbers of injected parents and scored progeny (P0; F1 n) are indicated. Also see *Figure 3—figure supplements 1 and 2*. (**B**) Fluorescence images of progeny from animals with a *gfp* tag of the ubiquitously expressed gene *gtbp-1* (*gtbp-1::gfp*) that were not injected (*left*), injected with 50 bp *gfp*-dsRNA (short dsRNA injection, *middle*), or injected with dsRNA transcribed from an ~730 bp template (mixed dsRNA injection, *right*). Complete silencing is not observed in neurons or in the developing vulva; brackets indicate additional regions with dim GFP fluorescence. Numbers of animals assayed (**n**) and percentages of L4-staged animals with the depicted expression patterns are indicated. Scale bar, 100 μm. Also see *Figure 3—figure supplement 2*. (**C**) Requirements for intergenerational transport of extracellular dsRNA. (*top left*) Differential Interference Contrast (DIC) and fluorescence images of a developing embryo from an animal injected in the body cavity with 50 bp dsRNA of the same sequence as in (**B**) and labeled at the 5' end of the antisense strand with Atto-565. Accumulation within the intestinal lumen (arrowhead), number of embryos imaged (**n**), and percentage of embryos with depicted pattern of fluorescence are indicated. Scale bar, 20 μm. (*top right* and *bottom*) Hermaphrodite animals of the indicated genotypes were injected with short dsRNA (left bars) or mixed dsRNA (right bars) and self-progeny (*top right*) or cross progeny after mating with wild-type males (*bottom*) were analyzed as in (**A**). Cases of no observable silencing are indicated with '0'. (**D**) Schematic summarizing requirements for transport of dsRNA from parental circulation to developing progeny. See text for details. Asterisks in (**A**) and (**C**) indicate p<0.05 with Bonferroni correction using $\chi^2$ test.

The online version of this article includes the following source data and figure supplement(s) for figure 3:

**Figure supplement 1.** Requirement of RME-2 for silencing in progeny by dsRNA injected into parents depends on concentration, length, and 5' modification of dsRNA.

**Figure supplement 1—source data 1.** Labelled uncropped images for *Figure 3—figure supplement 1B–E*.

**Figure supplement 2.** Extent of silencing in progeny by short or mixed dsRNA injected into parental circulation varies between tissues but has similar nuclear Argonaute requirements.

RME-2 (*Marré et al., 2016*). We found that silencing of *unc-22* in progeny by similarly synthesized but unlabeled 50 bp *unc-22*-dsRNA with a 5' OH delivered into parental circulation also showed a strong requirement for RME-2, unlike mixed dsRNA (*Figure 3A*). Further comparison between the two forms of dsRNA revealed that silencing in progeny by 50 bp dsRNA injected into parental circulation was detectably less efficient in somatic cells (*Figure 3A*, *Figure 3—figure supplement 2A and B*, *left*), even when ~14X more 50 bp dsRNA was delivered into parental circulation (*Figure 3—figure supplement 2B*, *right*), and was also less efficient in the germline (*Figure 3B*, *Figure 3—figure supplement 2A and C*). Efficient silencing in response to added dsRNA requires nuclear Argonaute proteins: NRDE-3 in somatic cells (*Guang et al., 2008*) and HRDE-1 in the germline (*Buckley et al., 2012*). Both 50 bp dsRNA and mixed dsRNA relied on HRDE-1 for silencing within the germline (*Figure 3—figure supplement 2A and C*) and could silence independent of NRDE-3 in somatic cells (*Figure 3—figure supplement 2A and C*). In addition to the diversity of RNA lengths observed in mixed dsRNA, another known feature that could distinguish dsRNA transcribed in bacteria or in vitro from synthesized 50 bp

**Table 1.** Summary of constraints on intergenerational transport of extracellular dsRNA.

| Stage of exposure | dsRNA type | Genetic requirement for germline entry | Heritability |
|---|---|---|---|
| L1 to L3 | mixed, ingested | none tested | not heritable |
| early L4 | mixed, ingested | *rme-2* required | no persistent silencing in P0 adults, heritable to F1 |
| | mixed, injected | *rme-2* not required | heritable to F1 |
| early adult | mixed, ingested | *rme-2* required | partial silencing in P0 adults, heritable to F1 |
| | mixed, injected | *rme-2* not required | heritable to F1 |
| late adult | mixed, ingested | *rme-2* not required | persistent silencing in P0 adults, heritable to F1 |
| | mixed, injected | *sid-1* or *rme-2* required | heritable to F1 |
| | synthesized 50 bp, injected | *sid-1* and *rme-2* required | heritable to F1 with partial silencing |
| | Synthesized 50 bp with 5'-phosphate, injected | *rme-2* is partially required | heritable to F1 |

dsRNA is the presence of 5' triphosphates on the transcribed dsRNA species instead of the 5' OH present in synthesized 50 bp dsRNA. In support of the impact of 5' phosphates on transport and/or silencing, addition of 5' monophosphates to synthesized 50 bp dsRNA injected into parental circulation reduced the dependence on RME-2 for silencing in progeny (*Figure 3—figure supplement 1E and F*), potentially by enhancing the ability of synthesized dsRNA to be imported by other dsRNA importers (e.g. SID-1) in the absence of RME-2. Thus, the requirements for dsRNA entry into the germline and subsequent silencing vary for different lengths and/or chemical forms of dsRNA (see summary in *Table 1*).

Fluorescently labeled 50 bp dsRNA delivered into parental circulation localized within intestinal cells in progeny (*Figure 3C*, *top left*), as has been observed for vitellogenin proteins (*Sharrock, 1983*) and fluorescent dyes (*Bossinger and Schierenberg, 1996*). Accumulation of fluorescently labeled dsRNA was also detected at the apical membrane of the intestine, which could reflect exocytosis of dsRNA into the lumen of developing intestinal cells. However, separation of the fluorescent label from dsRNA catalyzed by cellular enzymes cannot be excluded. Therefore, to dissect differences, if any, between the transport of unlabeled short dsRNA (synthesized 50 bp with 5'OH) and mixed dsRNA (mixture transcribed in vitro using ~1 kb DNA template) we injected *unc-22*-dsRNA into animals with mutations in genes that play roles in the import of dsRNA. We found that maternal SID-1 was required for silencing by short dsRNA in progeny (*Figure 3C*, *bottom*, left bars), suggesting that the SID-1-dependent entry of short dsRNA into the cytosol likely occurs in the injected parent or during early development in progeny. Uptake of dsRNA from the intestinal lumen requires SID-2, a transmembrane protein located in the apical membranes of intestinal cells (*Winston et al., 2007*; *McEwan et al., 2012*). We found that SID-2 was not required for most silencing in progeny by short or mixed dsRNA injected into parental circulation (*Figure 3C*, *top right* and *bottom*). Exit of dsRNA from intracellular vesicles requires SID-5, a transmembrane protein located in endolysosomal membranes (*Hinas et al., 2012*). Silencing in wild-type animals was comparable to silencing in *sid-5(-)* animals (*Figure 3C*, *top right*). However, when animals that lacked SID-1 were injected, SID-5 was required in progeny for silencing by mixed dsRNA from parental circulation (*Figure 3C*, *bottom*, right bars; as also reported in *Wang and Hunter, 2017*). Since dsRNA is expected to be present in vesicles upon entry through RME-2 in the absence of SID-1 (*Marré et al., 2016*; *Wang and Hunter, 2017*), this observation suggests that SID-5 is required for the release of mixed dsRNA from inherited vesicles in progeny.

In summary, injected extracellular dsRNA can enter the germline in parents and be transmitted to progeny through two routes with different substrate selectivity. One route is preferentially used by short dsRNA and relies on RME-2-mediated endocytosis of dsRNA into oocytes, where early exit from vesicles is required for silencing in progeny as evidenced by the need for maternal SID-1 (*Figure 3D*, blue). The other route appears to exclude short dsRNA but allows mixed dsRNA entry into the cytosol

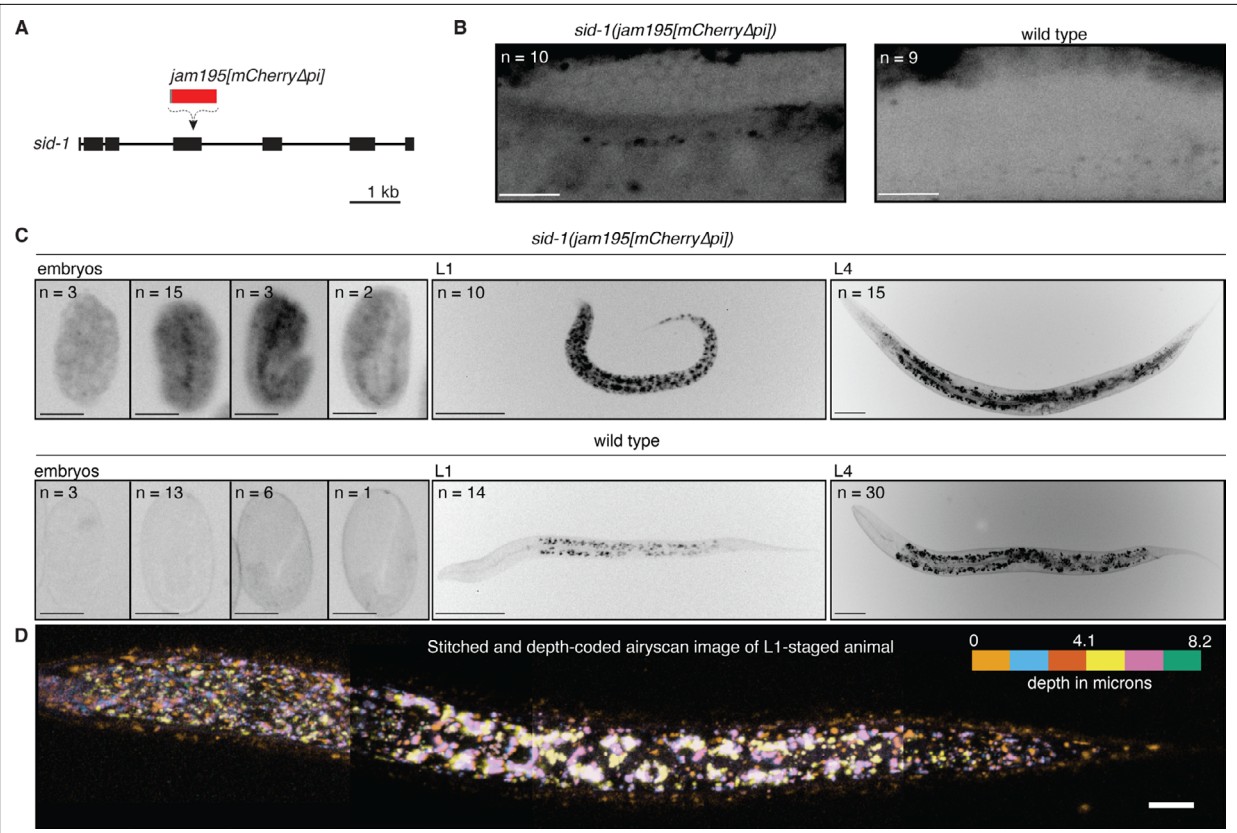

**Figure 4.** The expression pattern of SID-1 varies during development. (**A**) Schematic depicting insertion of *mCherry* sequence that lacks piRNA binding sites (*jam195[mCherryΔpi]*; *Zhang et al., 2018*; *Devanapally et al., 2021*) into the *sid-1* gene using Cas9-mediated genome editing. (**B and C**) Representative images showing fluorescence from SID-1::mCherry (black) in (**B**) the adult gonad arm, (**C**, *left*) developing embryos, (**C**, *middle*) L1-stage animals, or (**C**, *right*) L4-stage animals with *sid-1(jam195[mCherryΔpi])* compared to autofluorescence in wild-type animals of the same stages. Numbers (**n**) of each stage imaged are indicated (100% of animals exhibited the depicted expression patterns). For animals imaged in (**B**), the distal germline was obstructed by the intestine in 1/10 *sid-1(jam195[mCherryΔpi])* and 5/9 wild-type animals. (**D**) Airyscan image of an L1-staged animal assembled by stitching four different Z-stacks after depth-coding and taking maximum projections, illustrating the expression of SID-1::mCherry throughout the animal. Scale bar for adult gonad arms in (**B**) and embryos in (**C**), 20 µm; scale bar for larvae in (**C**), 50 µm and in (**D**), 10 µm. Also see *Technical comments* on 'Making a *sid-1* translational reporter' in Materials and methods.

The online version of this article includes the following figure supplement(s) for figure 4:

**Figure supplement 1.** Unsuccessful attempts to functionally tag SID-1 and to identify SID-1-dependent genes.

**Figure supplement 2.** Tetracycline-induced functional rescue of *sid-1* expression is evident in somatic tissues but not within the germline.

in the parental germline through SID-1 and exit from inherited vesicles in progeny through a process that requires both zygotic SID-1 and SID-5 (*Figure 3D*, grey; *Wang and Hunter, 2017*).

## Expression of SID-1 is consistent with a role in the intergenerational transport of extracellular dsRNA

All routes of dsRNA transport deduced using the experimental addition of dsRNA ultimately require SID-1 for entry into the cytosol. The proposed model (*Figure 3D*) for dsRNA transport into the germline and to progeny suggests that the expression pattern of SID-1 is likely to include the germline. We used Cas9-mediated genome editing to insert a piRNA-resistant *mCherry* sequence (*Zhang et al., 2018*; *Devanapally et al., 2021*) into the *sid-1* coding sequence (*Figure 4A*) to observe the endogenous expression pattern of SID-1::mCherry. This fusion protein was detectably functional in contrast to other attempts at tagging SID-1 (see *Technical comments* in Materials and methods and *Figure 4—figure supplement 1*). Fluorescence from SID-1::mCherry was visible in the proximal and distal regions of the adult germline (*Figure 4B*). Expression also progressively increased during development with tissue-specific enrichment in the developing embryo (*Figure 4C*, *left*), becoming ubiquitous in hatched

L1 larvae (*Figure 4C*, *middle*, and *Figure 4D*). SID-1::mCherry was not easily detectable in the germline during later larval development (*Figure 4C*, *middle* and *right*). In combination with the expression of RME-2 observed in the proximal germline (*Figure 1—figure supplement 1D*; *Grant and Hirsh, 1999*), this expression pattern of SID-1 is consistent with the entry of most dsRNA from circulation of adult animals into the germline, followed by activity of transport mechanisms in developing embryos and early larvae that inherit parental dsRNA.

To determine if acute, induced expression rather than developmental expression of SID-1 can be sufficient for the import of dsRNA into the germline, we engineered the endogenous *sid-1* gene to transcribe a fusion transcript with an aptamer-regulated ribozyme (*Figure 4—figure supplement 2A*, *left*) that cleaves itself when not bound to tetracycline (*Figure 4—figure supplement 2A*, *right*) (based on *Wurmthaler et al., 2019*). Exposing these animals to tetracycline enabled silencing by dsRNA in somatic tissues (hypodermis: *Figure 4—figure supplement 2B*, *left*; body-wall muscles: *Figure 4—figure supplement 2B*, *right*), indicative of stabilization of *sid-1* mRNA, production of SID-1 protein, and subsequent dsRNA import in somatic cells. However, such tetracycline-induced silencing was not detectable in the germline (*Figure 4—figure supplement 2C-F*). Yet, similar tagging of the ubiquitously expressed gene *gtbp-1::gfp* resulted in detectable rescue of expression within the germline by tetracycline (*Figure 4—figure supplement 2G*). A possible explanation for the poor rescue of SID-1 activity within the germline is that post-transcriptional mechanisms targeting *sid-1* mRNA in the germline but not in somatic cells interfere with tetracycline-dependent stabilization of the *sid-1* transcript (e.g. piRNA-based regulation of *sid-1* mRNA, *Ouyang et al., 2019*; *Dodson and Kennedy, 2019*), or that acute stabilization of the *sid-1* transcript does not override developmental regulation of SID-1 translation.

Additional attempts to tag the SID-1 protein guided by structure and to modulate *sid-1* transcripts guided by post-transcriptional regulatory interactions could improve control of dsRNA transport between cells. Nevertheless, the developmentally regulated expression observed for both SID-1 and RME-2 in the germline is consistent with intergenerational or transgenerational effects of endogenous dsRNA from parental circulation after development of the adult germline.

## Temporary loss of *sid-1* causes a transgenerational increase in the levels of mRNA from two germline genes

To understand how the dsRNA importer SID-1 might be used in endogenous gene regulation across generations, we searched for *sid-1*-dependent changes in gene expression that could be heritable (*Figure 5*, *Figure 2—figure supplement 2* and *Figure 5—figure supplement 1*). To control for genetic background (see *Technical comments* in Materials and methods), we used Cas9-mediated genome editing to delete the entire *sid-1* coding sequence or introduce a nonsense mutation in cohorts of the same wild-type animals. By comparing polyA+ RNA from this wild type with that of the newly generated *sid-1(jam113[deletion])* (*Figure 5A and B*, *Figure 5—figure supplement 1A*) or *sid-1(jam80[nonsense])* (*Figure 5A–C*) animals, we found that 26 genes were significantly (q<0.05) misregulated in *sid-1(jam113[deletion])* (*Figure 5—figure supplement 1B*) and 6 in *sid-1(jam80[nonsense])* (*Figure 5D*, *top*), both including *sid-1* (a list of significantly altered genes is in *Table 2*). The most upregulated gene in *sid-1(jam113[deletion])*, *F14F9.5* (*Figure 5—figure supplement 1B*), which is located immediately 3' to *sid-1* in the genome, was only misregulated in the deletion mutant and not in the nonsense mutant (*Figure 5—figure supplement 1C*, *left*). Both mutants, however, were equally defective for silencing by ingested dsRNA (*Figure 5B*). This observation suggests that while both mutations result in loss of SID-1 protein, the deletion of *sid-1* also changes local regulatory sequences (potentially explaining upregulation of the neighboring gene *F14F9.5*) and eliminates *sid-1* mRNA, which could participate in RNA-based regulatory interactions within the germline (*Ouyang et al., 2019*; *Dodson and Kennedy, 2019*). Nevertheless, we could detect two genes that were upregulated in both *sid-1(jam113[deletion])* and *sid-1(jam80[nonsense])* animals (red in *Figure 5D*, *top*, and *Figure 5—figure supplement 1B*): the identical loci *W09B7.2/F07B7.2* (*Figure 5—figure supplement 1C*, *middle*), and *Y102A5C.36* (*Figure 5—figure supplement 1C*, *right*). Intriguingly, another gene *cls-3* also changed in both mutants (*Table 2*) but in different directions (~3.4-fold decrease in the *sid-1(jam80[nonsense])* mutant but a ~5.8-fold increase in the *sid-1(jam113[deletion])* mutant), suggesting that the direction of change in expression can vary. Conservatively, we began by analyzing only the two genes with mRNA levels that changed in the same direction in both *sid-1* mutants. Both



**Figure 5.** Ancestral loss of SID-1 causes transgenerational changes in the mRNA levels of two germline genes that are subject to RNA regulation. (**A**) Schematic of modifications at the *sid-1* gene generated using Cas9-mediated genome editing. Deletion of the entire coding sequence (*jam113[deletion]*), a nonsense mutation (*jam80[nonsense]*), and its reversion to wild-type sequence (*jam86[revertant]*) are depicted. (**B**) Fractions of animals with the indicated genotypes that show silencing in response to *unc-22*-dsRNA (grey) or *bli-1*-dsRNA (black). Numbers of animals scored (**n**), significant differences using two-tailed test with Wilson's estimates for single proportions (asterisks, $P<0.05$ with Bonferroni correction) and 95% CI (error bars) are indicated. (**C**) Principal components explaining the variance between wild type (black), *sid-1(jam80[nonsense])* (red), and *sid-1(jam86[revertant])* (grey) polyA+ RNA samples. Almost all of the variance between samples is explained by PC 1. (**D**) Volcano plots of changes in the abundance of polyA+ RNA in *sid-1(jam80[nonsense])* (*top*) and *sid-1(jam86[revertant])* (*bottom*) animals compared with wild-type animals (black, $q<0.05$; red, both $q<0.05$ and change in the same direction in *sid-1(jam80[nonsense])* and *sid-1(jam113[deletion])*; see ***Figure 5—figure supplement 1***). While *sid-1* transcript levels in *sid-1(jam86[revertant])* are comparable to that in wild type (grey), *sdg-1* (*W09B7.2/F07B7.2*) and *sdg-2* (*Y102A5C.36*) transcript levels remain elevated in *sid-1(jam86[revertant])* (red). (**E**) Levels of spliced *sid-1* (*top*), *sdg-1* (*middle*) and *sdg-2* (*bottom*) transcripts measured using RT-qPCR. The median of three technical replicates is plotted for each of three biological replicates (bar indicates median) assayed before and after 1 year of passaging animals (year 1, dark grey; year 2, light grey). Asterisks indicate p<0.05 with Bonferroni correction using two-tailed Student's t-test. (**F**) Heatmap showing changes in the levels of transcripts (total RNA or mRNA) or antisense small RNAs (22G RNA) from *sid-1*, *sdg-1*, *sdg-2*, and *tbb-2* (abundant germline transcript for comparison). Fold changes (expressed as LogFC, indicating $\log_2$ for (m)RNA, $\log_{10}$ for piRNA binding, and $\log_{10}$ for 22G RNA) were deduced by integrating reports (study) of 21 experiments that identify subsets of genes as being subject to RNA-mediated regulation within the germline (# genes). These prior studies include comparisons of RNA or 22G RNA from wild-type animals with that from mutant animals (e.g. *mut-16(-)* 22G RNA), biochemical detection of piRNA binding to transcripts (piRNA-bound mRNA), and biochemical detection of 22G RNA binding to an Argonaute (HRDE-1-bound 22G RNA). 'NS' indicates cases where changes, if any, were not significant based on the criteria used in the study. A conservative value of two-fold is assigned to all genes reported as changing >two-fold in ***Ni et al., 2016***.

The online version of this article includes the following figure supplement(s) for figure 5:

**Figure supplement 1.** Selective disruption of *sid-1* followed by restoration to wild type reveals *sid-1*-dependent transcripts expressed in the germline that show heritable change.

**Table 2.** List of genes changed in *sid-1(jam80[nonsense])* animals or in *sid-1(jam113[deletion])* animals compared with wild-type animals.

| Genes | Change compared with wild type |
| --- | --- |
| *sid-1* | Down in *sid-1(jam80)* and *sid-1(jam113)* |
| *sdg-1* (W09B7.2/F07B7.2) | Up in *sid-1(jam80)* and *sid-1(jam113)* |
| *sdg-2* (Y102A5C.36) | Up in *sid-1(jam80)* and *sid-1(jam113)* |
| *cls-3* | Down in *sid-1(jam80)* and Up in *sid-1(jam113)* |
| *sax-2* | Down in *sid-1(jam80)* |
| Y46G5A.23 | Up in *sid-1(jam80)* |
| F14F9.5 | Up in *sid-1(jam113)* |
| T10D4.6 | Up in *sid-1(jam113)* |
| F47D12.9 | Down in *sid-1(jam113)* |
| C07G1.7 | Up in *sid-1(jam113)* |
| Y48G1BL.5 | Up in *sid-1(jam113)* |
| Y20F4.4 | Up in *sid-1(jam113)* |
| ZK177.9 | Up in *sid-1(jam113)* |
| C27C7.1 | Up in *sid-1(jam113)* |
| Y38H6C.4 | Up in *sid-1(jam113)* |
| C40A11.8 | Up in *sid-1(jam113)* |
| C24H11.2 | Up in *sid-1(jam113)* |
| C18D4.6 | Up in *sid-1(jam113)* |
| F15B9.10 | Up in *sid-1(jam113)* |
| F07B7.1 | Up in *sid-1(jam113)* |
| ZC204.14 | Down in *sid-1(jam113)* |
| Y47D7A.19 | Up in *sid-1(jam113)* |
| Y26G10.5 | Down in *sid-1(jam113)* |
| B0554.1 | Down in *sid-1(jam113)* |
| F13A2.1 | Down in *sid-1(jam113)* |
| C10C6.13 | Down in *sid-1(jam113)* |
| H25K10.141 | Down in *sid-1(jam113)* |
| Y43D4A.1 | Up in *sid-1(jam113)* |

*W09B7.2/F07B7.2* and *Y102A5C.36* have been reported (*Reed et al., 2020*) to be expressed within the germline (*Figure 5—figure supplement 1D*, *left*) and regulated by endogenous small RNAs (*Figure 5—figure supplement 1D*, *middle* and *right*). Spliced mRNA levels measured at a later generation using RT-qPCR demonstrated that both transcripts were upregulated in *sid-1(jam80[nonsense])* animals compared to wild-type animals as expected (*Figure 5E*), but no upregulation was detectable in *sid-1(jam113[deletion])* animals (*Figure 5—figure supplement 1E*). This difference between the two *sid-1* mutants could reflect increased variation in expression (as was observed for *cls-3* using RNAseq) or could reflect complex effects caused by deletion of *sid-1* DNA (e.g. *F14F9.5* overexpression, loss of *sid-1* mRNA, etc.) that could be independent of SID-1 protein function.

To determine if changes in *W09B7.2/F07B7.2* and *Y102A5C.36* expression were heritable, we reverted the *sid-1* nonsense mutation to wild-type sequence using Cas9-mediated genome editing. This immediately restored silencing by ingested dsRNA (*Figure 5B*) with concomitant recovery of *sid-1* mRNA to wild-type levels (*Figure 5E*, *top*). In contrast, changes in both *W09B7.2/F07B7.2* and

*Y102A5C.36* expression persisted (*Figure 5D*, *bottom*) even after one year of passaging the reverted animals (*sid-1(jam86[revertant]*); i.e. after >100 generations, *Figure 5E*, *middle* and *bottom*). Thus, the *sid-1*-dependent accumulation of mRNA from these two germline genes persisted for many generations, likely through mechanisms that maintain heritable epigenetic changes. We hereafter refer to these **s**id-1-**d**ependent **g**enes (*sdg*) that show heritable epigenetic changes in response to temporary loss of SID-1 as *sdg-1* (*W09B7.2/F07B7.2*) and *sdg-2* (*Y102A5C.36*).

## The *sid-1*-dependent gene *sdg-1* is affected by many factors that regulate RNA silencing in the germline

To determine if expression of *sdg-1* and *sdg-2* is regulated by other proteins that play a role in RNA silencing within the germline, we examined 21 published datasets (*Buckley et al., 2012*; *Dodson and Kennedy, 2019*; *Welker et al., 2007*; *Spike et al., 2008*; *Wan et al., 2021*; *Suen et al., 2020*; *Wahba et al., 2021*; *Shen et al., 2018*; *Goh et al., 2014*; *Lee et al., 2012*; *Batista et al., 2008*; *Kim et al., 2021*; *Ni et al., 2016*) that reported changes that depend on such proteins. For each dataset, we determined if the lists of genes reported as showing significant changes in mutants compared to the respective wild types included *sdg-1* and/or *sdg-2*. This analysis revealed that changes in mRNA and/or antisense small RNAs of *sdg-1* were detected in 20 of the 21 datasets while changes in *sdg-2* were observed in 9 of 21 (*Figure 5F*). When detected, changes in *sdg-2* were in the same direction as changes in *sdg-1*, suggestive of similar regulation of both genes.

RNAs transcribed in the germline can be recognized as they exit the nuclear pores by piRNAs bound to the Argonaute PRG-1, which recruits them for regulation by antisense small RNAs called 22G RNA made by proteins organized within perinuclear germ granules (reviewed in *Sundby et al., 2021*). Interaction with piRNAs was detected for RNA from *sid-1*, *sdg-1*, and *sdg-2*, and the control gene *tbb-2* using crosslinking, ligation, and sequencing of hybrids (*Shen et al., 2018*; *Figure 5F*), consistent with their germline expression. Depletion of downstream 22G RNAs targeting both *sid-1* and *sdg-1* was detectable upon loss of the germ granule component MUT-16 (*Suen et al., 2020*; *Figure 5F*). Both genes were among the top 500 genes targeted by 22G RNAs bound by the secondary Argonaute HRDE-1/WAGO-9 (*Buckley et al., 2012*; *Figure 5F*), suggesting similar downregulation of both genes using 22G RNAs. Furthermore, multiple datasets support downregulation of *sdg-1* within the germline by HRDE-1/WAGO-9-bound 22G RNAs in the absence of PRG-1. One, loss of HRDE-1/WAGO-9 increased *sdg-1* RNA in whole animals (*Ni et al., 2016*; *Figure 5F*) and in dissected gonads (*Kim et al., 2021*; *Figure 5F*). Two, loss of PRG-1 decreased *sdg-1* RNA (*Figure 5F*) and increased 22G RNAs that are antisense to *sdg-1* (*Figure 5F*) in dissected gonads (*Wahba et al., 2021*). Three, although animals that lack PRG-1 become progressively sterile, the increase in *sdg-1* 22G RNA persisted in near-sterile animals (*Figure 5F*, near-sterile in *Wahba et al., 2021*), and this increase was eliminated upon additional loss of HRDE-1/WAGO-9 (*Figure 5F*, near-sterile in *Wahba et al., 2021*).

As expected for *sid-1*-dependent downregulation of *sdg-1*, multiple datasets support an inverse relationship between the two genes. In animals lacking PRG-1, *sid-1* RNA levels increased and *sid-1* 22G RNAs decreased (*Lee et al., 2012*; *Figure 5F*), but both *sdg-1* RNA and *sdg-2* RNA levels decreased along with an increase in 22G RNAs (*Suen et al., 2020*; *Wahba et al., 2021*; *Shen et al., 2018*; *Goh et al., 2014*; *Lee et al., 2012*; *Batista et al., 2008*; *Figure 5F*). This inverse relationship between *sid-1* and *sdg-1* RNA regulation is also observed when many components of germ granules are mutated as indicated by changes in 22G RNA upon loss of the embryonic P granule components MEG-3/-4 (*Dodson and Kennedy, 2019*; *Figure 5F*), the PRG-1 interactor DEPS-1 (*Suen et al., 2020*; *Figure 5F*), or the Z granule component ZSP-1 (*Wan et al., 2021*; *Figure 5F*; also known as PID-2, *Placentino et al., 2021*).

In addition to the above studies, pioneering studies that used microarrays identified *sdg-1* as upregulated in animals lacking the germ granule component DEPS-1 (*Spike et al., 2008*; *Figure 5F*) and in animals lacking the dsRNA-binding protein RDE-4 (*Figure 5F*; second-most upregulated in *Welker et al., 2007*), which recruits dsRNA imported through SID-1 and other intracellular dsRNA for processing and eventual gene silencing. Animals that lack RDE-4 show a ~47.5-fold increase in *sdg-1* RNA (*Welker et al., 2007*). A reduction in RDE-4 activity could also contribute to the ~11.6-fold increase in *sdg-1* RNA seen in *deps-1(-)* animals because these animals also show a ~3.2-fold decrease in *rde-4* RNA (one of 13 downregulated genes in *Spike et al., 2008*). These observations support the

idea that appropriate regulation of *sdg-1* RNA requires both piRNA-mediated processes that act via germ granule components such as DEPS-1 and dsRNA-mediated processes that use SID-1 and RDE-4.

In summary, the levels of *sdg-1* RNA are detectably regulated by the dsRNA-selective importer SID-1, the dsRNA-binding protein RDE-4, and the piRNA-binding Argonaute PRG-1. Presence of dsRNA-mediated regulation or loss of piRNA-mediated regulation enhances MUT-16-dependent production of secondary small RNAs that bind the secondary Argonaute HRDE-1/WAGO-9. Consistent with downregulation of these *sid-1*-dependent transcripts by SID-1, disruption of many components of germ granules results in opposite effects on these transcripts and *sid-1* RNA. Intriguingly, a search of protein interaction studies revealed that the SDG-1 protein is among the interactors of two germ granule components: PID-2 by immunoprecipitation (*Placentino et al., 2021*; also known as ZSP-1, *Wan et al., 2021*) and DEPS-1 by proximity labeling (*Price et al., 2021*). This interaction raises the possibility that the *sdg-1* gene is not only regulated by germ granule components, but also that its protein product associates with one or more components of germ granules.

## Regulation of *sdg-1* RNA is susceptible to epigenetic changes that last for many generations

Given these observations on the regulation of *sdg-1* and the potential for the SDG-1 protein to be a regulator of RNA silencing, we focus here on the analysis of this gene. The *sdg-1* gene is located within a retrotransposon (*Figure 6—figure supplement 1A*) that is within a duplicated ~40 kb region and has two recognizable paralogs (*Figure 6—figure supplement 1B*). However, there are no detectable SDG-1 homologs in species other than *C. elegans*, suggesting that it is a recently evolved gene. Potentially because of this lack of conservation, AlphaFold is unable to predict a confident structure for SDG-1 (CELE_F07B7.2 on the Database; *Varadi et al., 2022*). To facilitate analysis of SDG-1 expression, we tagged both loci that encode SDG-1 with *mCherry* coding sequences lacking piRNA-binding sites (*mCherryΔpi*; *Zhang et al., 2018*; *Devanapally et al., 2021*; *Figure 6—figure supplement 1C and D*). This tagging resulted in the expression of *sdg-1::mCherryΔpi* mRNA being ~16-fold higher than *sdg-1* mRNA (*Figure 6—figure supplement 1E*), potentially because of the reduction in the overall density of piRNA-binding sites per transcript, the additional introns included in *mCherryΔpi* (based on *Okkema et al., 1993* and *Crane et al., 2019*), and/or other unknown factors. Fluorescence from SDG-1::mCherry was observed in the germline of adult animals (*Figure 6A*). However, animals showed variation in SDG-1::mCherry expression between their two gonad arms (*Figure 6A*, *middle* shows bright anterior [20% of animals] and *right* shows bright posterior [6% of animals]). A contributing feature for the observed stochasticity could be the location of *sdg-1* within a duplicated region (*Figure 6—figure supplement 1A*), as suggested by similar stochastic RNA silencing of multicopy transgenes but not single-copy transgenes (*Le et al., 2016*). Despite this variation, unbiased passaging of self-progeny for more than 18 generations continuously preserved SDG-1::mCherry expression in an otherwise wild-type background (*Figure 6B*). In contrast, mating, which can perturb RNA regulation within the germline in cross progeny (*Devanapally et al., 2021*), caused dramatic changes in *sdg-1* expression that persisted in descendants (*Figure 6C*). Mating animals that express SDG-1::mCherry with wild-type animals resulted in heritable changes along lineages that express *sdg-1::mCherryΔpi* mRNA or that express *sdg-1* mRNA (*Figure 6C* and *Figure 6—figure supplement 2A*). This discovery of mating-induced perturbation in gene expression raises caution in interpreting past studies (summarized in *Figure 5F*) where changes in multiple independent isolates were not examined. Nevertheless, when we used genetic crosses to determine the impact of mutations on *sdg-1* expression, we observed reduced SDG-1::mCherry fluorescence in mutants predicted to have reduced levels of intracellular dsRNA (*Figure 6D*, *sid-2(-)*, *sid-5(-)*) or reduced processing of intracellular dsRNA (*Figure 6D*, *eri-1(-)*). In contrast, we observed an increase in SDG-1::mCherry fluorescence in animals lacking MUT-16 (*Figure 6D*). Finally, animals lacking RME-2, which lack the ability to import many maternal factors (e.g. lipids, proteins, RNAs, etc.), also showed an increase in SDG-1::mCherry fluorescence (*Figure 6D*).

To avoid mating-induced perturbations of RNA regulation within the germline, we used Cas9-mediated genome editing to introduce mutations into animals that express SDG-1::mCherry in an otherwise wild-type background. Use of this approach to mutate a control gene with no known roles in RNA regulation within the germline resulted in similar levels of SDG-1::mCherry fluorescence in multiple isolates of animals with and without the mutation (*Figure 6—figure supplement 2B*). In



**Figure 6.** The *sdg-1* gene is prone to stochastic changes in gene expression that can become heritable. (**A**) Representative images showing fluorescence of SDG-1::mCherry (black) in a wild-type background. While most animals showed symmetric expression in the germline (*left*), animals with >two-fold difference in fluorescence between both gonad arms (bright anterior, *middle* and bright posterior, *right*) were also observed. Punctate fluorescence in the intestine likely represents autofluorescence. Scale bar, 50 μm. (**B**) Quantification of SDG-1::mCherry fluorescence intensity (arbitrary units, a.u.) in adult gonad arms (anterior arm, dark grey; posterior arm, light grey) of *sdg-1(jam137[mCherryΔpi])* animals starting in one generation (**x**) and continuing in successive generations as indicated. Numbers of gonad arms quantified (**n**) is indicated. Expression in one generation was not significantly different when compared to that in the previous tested generation using Mann-Whitney U test for two-sided comparisons and Bonferroni correction. (**C**) Lineages and estimated relative *sdg-1* expression 10 generations after mating wild-type (open circle) males with *sdg-1::mCherryΔpi* (filled circle) hermaphrodites and vice versa, and isolating *sdg-1(+)* or *sdg-1::mCherry* animals from F1 heterozygotes (half-filled circle). Expression of *sdg-1* in the F10 generation was measured by RT-qPCR of *sdg-1* mRNA purified from pooled wild-type animals of mixed stages or by quantification of SDG-1::mCherry fluorescence in gonad arms of adult *sdg-1::mCherryΔpi* animals. Relative levels of *sdg-1* mRNA and SDG-1::mCherry fluorescence intensity were converted to units of estimated relative *sdg-1* expression (see Materials and methods) for comparison. See **Figure 6—figure supplement 2A** for raw data. (**D–F**) Fluorescence intensity measurements (quantified as in **B**) in adult animals with *sdg-1::mCherryΔpi* (+) and additionally with mutations in genes introduced through genetic crosses (in regulators of dsRNA import *rme-2*, *sid-2* or *sid-5*, or in regulators of RNA silencing *mut-16* or *eri-1*) or through genome editing (in regulators of dsRNA import *sid-1* or *sid-3*, or in regulators of RNA silencing *rde-1* or *deps-1*). Asterisks indicate p<0.05 with Bonferroni correction using Mann-Whitney U test for two-sided comparisons between animals with *sdg-1::mCherryΔpi* (+) and animals with additional mutations. Nonsense mutations (nonsense) or deletion utations (deletion) introduced through genetic crosses (isolate numbers #1, #2, etc. in **D**) or genome editing (different alleles in **E** and **F**) and numbers of gonad arms (**n**) quantified for each isolate are indicated. Mutations in genes required for dsRNA import or subsequent silencing resulted in fewer animals with asymmetric fluorescence between gonad arms (a combined proportion of 21/197 for *sid-1*, *sid-3*, *rde-1* and *deps-1* mutants versus 22/84 for wild type, p=0.0009 using two-tailed test with Wilson's estimates for single proportions). Animals with at least one gonad arm brighter than the dimmest wild-type gonad arm in (**A**) and with asymmetric gonad arms were found in different genotypes (anterior bright: *sid-1(-)* – 5/122, *sid-3(-)* – 1/29, *rde-1(-)* – 2/22, *deps-1(-)* – 4/24, and posterior bright: *sid-1(-)* – 6/122, *rde-1(-)* – 2/22, *deps-1(-)*

*Figure 6 continued on next page*

*Figure 6 continued*

– 1/24). (**G**) Fluorescence intensity measurements as in (**B**) of animals with *sdg-1::mCherryΔpi* that show loss of fluorescence when a nonsense mutation is introduced in *sid-1* using genome editing ~30 generations (gen.) later remain changed despite reversion of *sid-1* nonsense mutation to wild-type sequence after ~20 additional generations. Subsequent mutation of *deps-1* after another ~110 generations restored SDG-1::mCherry fluorescence to wild-type levels. Also see *Figure 6—figure supplements 1 and 2*.

The online version of this article includes the following source data and figure supplement(s) for figure 6:

**Figure supplement 1.** The *sid-1*-dependent gene *sdg-1* is expressed from two identical loci (*W09B7.2/F07B7.2*) and loss of its expression in *sid-1(nonsense)* animals fails to recover in *sid-1(revertant)* animals.

**Figure supplement 1—source data 1.** Labelled uncropped images for *Figure 6—figure supplement 1D*.

**Figure supplement 2.** Mating but not genome editing can initiate distinct heritable changes in *sdg-1* expression.

contrast, mutating *sid-1* using Cas9-mediated genome editing caused a range of expression levels in different isolates when compared with *sid-1(+)* animals (*Figure 6E* - 6 isolates lower, 2 isolates comparable, and 1 isolate higher), which differs from the increase in *sdg-1* mRNA observed upon SID-1 loss in the single isolate examined earlier (*Figure 5*). Mutating *sid-3* also lowered the levels of SDG-1::mCherry in one isolate, but caused no detectable change in another (*Figure 6F*). While both isolates with loss of RDE-1 showed lower levels of SDG-1::mCherry, both isolates with loss of the germ granule component DEPS-1 showed higher levels of SDG-1::mCherry (*Figure 6F*). These experiments also reveal that in most isolates of animals expected to have reduced levels of intracellular dsRNA (*sid-3(-)*, *sid-1(-)*) or reduced processing of intracellular dsRNA (*rde-1(-)*), SDG-1::mCherry fluorescence is reduced. In contrast, isolates lacking DEPS-1 showed increased expression of SDG-1::mCherry.

Collectively, the observations on SDG-1 expression using mutants suggest that the uptake and processing of intracellular dsRNA (influenced by SID-1, SID-2, SID-3, SID-5, RDE-1, and ERI-1) and the function of germ granules (influenced by MUT-16 and DEPS-1) are both necessary to maintain intermediate levels of SDG-1 expression across generations. Once the levels of SDG-1::mCherry were reduced upon loss of SID-1, downregulation persisted across generations even after restoration of wild-type SID-1 (*Figure 6G*), just as the upregulation of untagged *sdg-1* mRNA also persisted (*Figure 5*). Despite >100 generations of such persistent silencing, the expression of SDG-1::mCherry could be restored by mutating *deps-1* (*Figure 6G*), implicating small RNA-based regulation and germ granules in the maintenance of the new epigenetic states established upon loss of SID-1.

## SID-1-dependent genes, including SDG-1, could alter RNA-mediated regulation in the germline

Since SDG-1 interacts with regulators of RNA silencing (ZSP-1/PID-2, *Placentino et al., 2021*) and DEPS-1 (*Price et al., 2021*), loss of SID-1 could both block the entry of extracellular dsRNA into the cytosol and change intracellular RNA regulation through SID-1-dependent genes such as *sdg-1*. In support of this possibility, intracellular delivery of dsRNA through injection into the syncytial germline of *sid-1(-)* animals showed a weak defect in silencing (Figure 2D in *Marré et al., 2016* and Figure 1 in *Wang and Hunter, 2017*). To examine if changes in the levels of SDG-1 alone could account for such a defect in silencing by intracellular dsRNA in the germline, we exposed independently-generated *sdg-1* deletion animals and animals that overexpress *sdg-1* (i.e. with *sdg-1::mCherryΔpi*) to dsRNA matching the germline gene *pos-1* for ~16 hr. This short exposure to *pos-1*-dsRNA caused intermediate levels of silencing in wild-type animals and comparable intermediate silencing in *sdg-1* mutant and *sdg-1* overexpressing animals (*Figure 7A*), suggesting that changes in *sdg-1* levels alone are not sufficient to cause a defect in silencing by dsRNA that is detectable using this assay. Alternatively, the previously described defect in silencing by intracellular dsRNA in *sid-1(-)* animals (*Marré et al., 2016*; *Wang and Hunter, 2017*) could be through the promotion of competing piRNA-mediated gene regulation in the absence of SID-1. This notion that dsRNA-mediated and piRNA-mediated gene regulation compete in the germline is supported by a study that demonstrated that loss of PRG-1 enhances heritable RNA silencing by dsRNA (*Shukla et al., 2021*). To test if loss of SID-1 or a *sid-1*-dependent gene enhances piRNA-mediated silencing, we used an experimental system which initiates piRNA-mediated silencing of the two-gene operon described in *Figure 2* through mating, independent of externally provided dsRNA (*Devanapally et al., 2021*). We found that *sid-1(-)* animals exhibited enhanced mating-induced silencing (*Figure 7B*, *top right*:~50% off in *sid-1(+)* vs



**Figure 7.** SID-1 modifies RNA regulation within the germline, potentially through *sdg-1* and other *sid-1*-dependent genes. (**A**) (*left*) Schematic of assay for sensitive detection of *pos-1* silencing by ingested dsRNA. (*right*) Numbers of developed progeny (> 3rd larval stage) laid by wild-type animals, animals with a deletion (Δ) in *sdg-1* (*jam232*, *jam241*, *jam242*) or animals with overexpression (*sdg-1::mCherryΔpi*) of *sdg-1* exposed to *pos-1* dsRNA (red) or control dsRNA (black) for 16 hr are plotted. Asterisks indicate p<0.05 using Mann-Whitney U test for two-sided comparisons with Bonferroni

*Figure 7 continued on next page*

**eLife** Research article

Genetics and Genomics

*Figure 7 continued*

correction. (**B**) Cross progeny males that inherited the *mex-5p::mCherry::h2b::gfp::h2b* transgene (T; *Devanapally et al., 2021*; also used in *Figure 2*) from maternal (*left*) or paternal (*right*) parents, both of wild-type, *sid-1(-)*, or *sdg-1(-)* background, were scored for expression of mCherry and GFP (bright, dim, off). Wild-type data for top set (n=77 and n=33) are replotted from *Devanapally et al., 2021* for comparison. Dashed line separates independent experiments. Asterisk indicates p<0.05 with Bonferroni correction using $\chi^2$ test; n.s. indicates not significant. (**C**) Representative AiryScan images of the distal germline (*left*; scale bar, 10 µm) or single germline nuclei (*right*; scale bar, 2 µm) showing SDG-1::mCherry alone (*top*) or with GFP::ZNFX-1 (*bottom*, merge and single channel images). The number of animals imaged (**n**) and the percentage that show enrichment of SDG-1::mCherry in perinuclear foci are indicated. Sites of SDG-1::mCherry enrichment coincide with GFP::ZNFX-1 localization. Boxes in left mark the nuclei shown in right. (**D**) Representative images showing entry of SDG-1::mCherry into the nucleus in −1 oocytes (*left*) and upon pronuclear fusion in early embryos during the time course indicated (*right*). Numbers of germlines and embryos imaged are indicated. Scale bars, 20 µm. Also see *Videos 1–4*. (**E**) Representative image of the hermaphrodite germline in animals with a translational (*left*) or transcriptional (*right*) reporter of *sdg-1*. Scale bar, 20 µm. Apparent extracellular punctae of SDG-1::mCherry and mCherry surrounding the proximal germline requires further study, but could be non-specific because similar localization is observed in animals with other promoters driving mCherry expression, but not GFP expression, in the germline (data not shown). The numbers of animals with the depicted fluorescence pattern are indicated. (**F and G**) Response of the transcriptional *sdg-1* reporter (*sdg-1p::mCherryΔpi[sdg-1(Δ)]::sdg-1 3' UTR*) to the addition of *unc-22*-dsRNA (**F**) or loss of *rde-4* (**G**). Quantification and asterisk are as in *Figure 6*. (**H**) Models for dsRNA import into the germline (*top*) and subsequent RNA-mediated regulation of *sdg-1* (*bottom*). See text for details.

100% off in *sid-1(qt9[nonsense])*), while animals lacking *sdg-1* showed a small reduction in mating-induced silencing that was not statistically significant (*Figure 7B, bottom right*, ~40% off in *sdg-1(+)* vs ~30% off in *sdg-1(jam232[deletion])*). Taken together, these results support the model that loss of SID-1 weakly inhibits silencing by intracellular dsRNA (*Marré et al., 2016*; *Wang and Hunter, 2017*) but enhances silencing by piRNAs within the germline.

RNA regulation within the germline using piRNAs relies on phase-separated granules (reviewed in *Dodson and Kennedy, 2020*). To determine if the previous identification of SDG-1 as a putative interactor of the Z-granule component PID-2 (*Placentino et al., 2021*)/ZSP-1 (*Wan et al., 2021*) and potentially of the P-granule-adjacent protein DEPS-1 (*Price et al., 2021*) could be seen as colocalization with Z-granules in vivo, we examined the localization of SDG-1::mCherry within the cytoplasm at high resolution using AiryScan imaging (*Huff et al., 2017*). SDG-1::mCherry was enriched in perinuclear foci in many animals (*Figure 7C*, *top*; 7 of 9 animals) and these sites of enrichment overlapped with the Z-granule marker GFP::ZNFX-1 (*Ishidate et al., 2018*; *Wan et al., 2018*; *Figure 7C*, *bottom*; 100% colocalized in 10 of 12 animals with enrichment). However, given the resolution of our imaging, we cannot distinguish between a specific interaction with one of the many Z granule components (*Zhao et al., 2024*) and an interaction with components of the many other nearby perinuclear granules that continue to be reported (*Huang et al., 2024*).

Therefore, we conservatively conclude that SDG-1::mCherry colocalizes with perinuclear germ granules. Time-course imaging revealed re-localization of SDG-1::mCherry into the nucleus from the cytoplasm in the −1 oocyte, which will be the first to be fertilized (*Figure 7D*, *left* and *Video 1*), along with subsequent exclusion from the nucleus before the maternal and paternal pronuclei meet (*Figure 7D*, *right* and *Video 2*). Such dynamic entry into the nucleus followed by exclusion from the nucleus also occurred during early cell divisions in the developing embryo (*Videos 3 and 4*). The timing of nuclear entry of SDG-1::mCherry coincides with the nuclear envelope breakdown events that occur during fertilization and early development (*Huelgas-Morales and Greenstein, 2018*). The *sdg-1* coding sequence was required for regulated nuclear entry because deletion of the *sdg-1* open reading frame in *sdg-1::mCherryΔpi* animals resulted in mCherry expression throughout the germline in both the cytoplasm and nuclei (*Figure 7E*). Nuclear

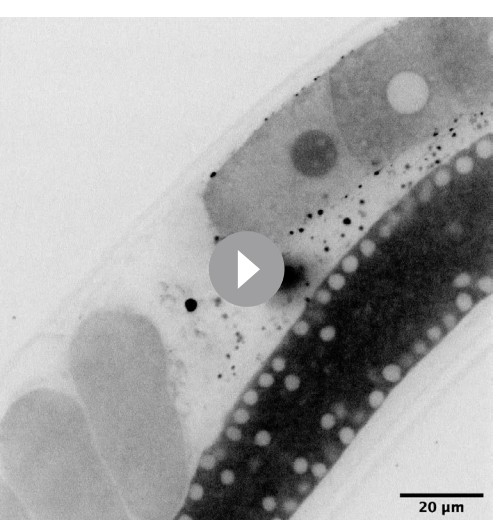

**Video 1.** Animals expressing SDG-1::mCherry (black) showing nuclear localization in −1 oocytes, but cytoplasmic localization in other oocytes and in the distal germline.
https://elifesciences.org/articles/99149/figures#video1

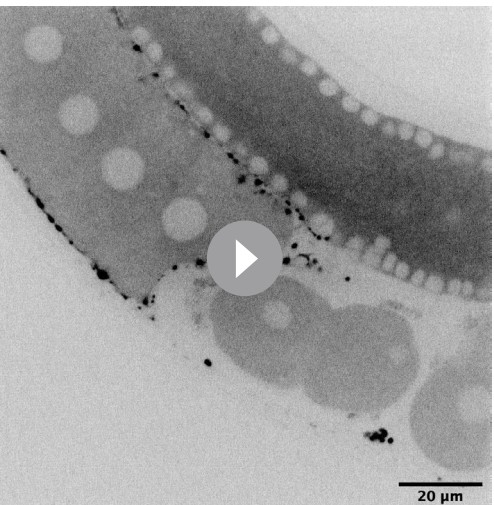

**Video 2.** Animals expressing SDG-1::mCherry (black) showing dynamic entry of SDG-1::mCherry into the nucleus in a zygote in utero after the maternal and paternal pronuclei meet.
https://elifesciences.org/articles/99149/figures#video2

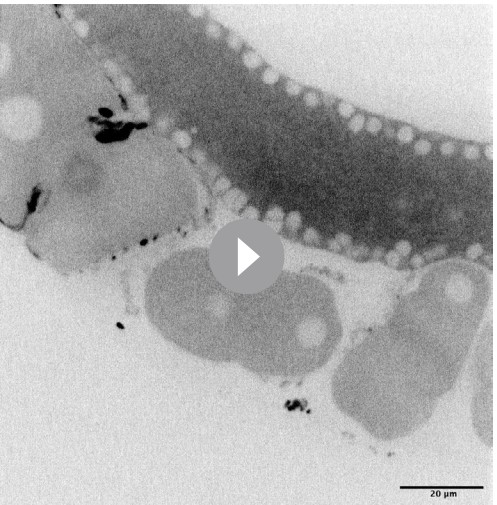

**Video 4.** Animals expressing SDG-1::mCherry (black) showing nuclear localization in oocytes during fertilization and in embryos in utero during cell divisions.
https://elifesciences.org/articles/99149/figures#video4

localization of the SDG-1 protein in the –1 oocyte is like that of the essential Argonaute CSR-1b (*Charlesworth et al., 2021*), thought to play a role in protecting transcripts from silencing. Exposure to ingested dsRNA did not alter the expression of *sdg-1p::mCherryΔpi[sdg-1(Δ)]::sdg-1 3' UTR* (*Figure 7F*) but loss of *rde-4* perturbed expression such that one isolate showed loss of expression while the other showed enhanced expression (*Figure 7G*). These results suggest that while the *sdg-1* open reading frame is required for exclusion from germline nuclei, it is not required for the response of the *sdg-1* gene to changes in intracellular dsRNA. Together, these observations on the levels and localizations of SDG-1 raise the possibility that this protein is actively regulated by extracellular and intracellular dsRNA with a role in heritable RNA silencing, potentially with perinuclear germ granules, and additional cell-cycle coupled roles during early development, potentially through nucleocytoplasmic shuttling.

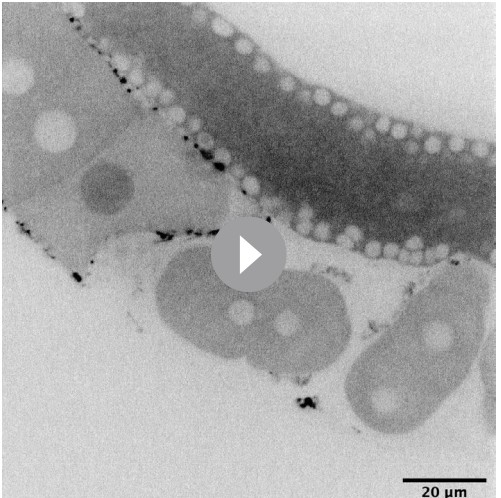

**Video 3.** Animals expressing SDG-1::mCherry (black) showing nuclear localization in –1 oocytes and in an early-staged embryo in utero during cell divisions.
https://elifesciences.org/articles/99149/figures#video3

## Discussion

Our analyses suggest a model for the impact of dsRNA from parental circulation on descendants (*Figure 7H*). Extracellular dsRNA can accumulate in parental circulation through regulated secretion from neurons (e.g. oxidative damage promotes accumulation) and potentially other tissues. Uptake into the germline is RME-2-dependent for some forms of dsRNA (blue in *Figure 7H*), but RME-2-independent for other forms of dsRNA (grey in *Figure 7H*). While all forms of dsRNA require SID-1 for the entry of dsRNA into the cytosol in progeny, RME-2-dependent dsRNA also require SID-5. Such dsRNA from parental circulation along with other intracellular dsRNA are processed with the help of the dsRNA-binding protein RDE-4 in progeny (*Figure 7H*, *bottom*). This processing of dsRNA regulates the SID-1-dependent gene *sdg-1* by reducing variation between animals in the levels

of *sdg-1* mRNA, although the dsRNA sequences likely need not match *sdg-1* because no such dsRNA was reported in the published dsRNAome (*Whipple et al., 2015*) and because this regulation can occur in the absence of the *sdg-1* open reading frame. The *sdg-1* gene is located within a retrotransposon that is also targeted by heritable RNA silencing, but the SDG-1 protein is enriched near perinuclear germ granules (observed using the Z granule component ZNFX-1), which are required for heritable RNA silencing. Consistent with the SDG-1 protein promoting heritable RNA silencing, the *sdg-1::mCherryΔpi* gene is highly susceptible to mating-induced silencing, potentially owing to the ~16-fold higher levels of SDG-1::mCherry compared with that of SDG-1 in wild-type animals. In agreement with this proposal, SID-1 limits heritable RNA silencing because loss of SID-1 enhances mating-induced silencing (*Figure 7B*, *top*). Since the *sdg-1* gene is located within a retrotransposon that is targeted by heritable RNA silencing, this mechanism for regulating the regulators of heritable RNA silencing such as SDG-1 (and potentially other SDGs) reveals a strategy for tuning an autoregulatory loop for heritable RNA silencing by using competing dsRNA processing. Intriguingly, SDG-1 becomes enriched within nuclei upon nuclear envelope breakdown during fertilization and during early cell divisions in embryos (*Figure 7H*, bottom right) with active exclusion from nuclei after each reformation of the nuclear envelope, suggestive of additional roles for this retrotransposon-encoded protein.

## Import of extracellular dsRNA

The temporal and/or spatial preferences observed for silencing by both ingested dsRNA (*Figure 1*) and neuronal dsRNA (*Figure 2*) could be because of unknown characteristics of the exported neuronal dsRNA or ingested dsRNA expressed from bacteria (e.g. modifications, lengths, structures, etc.) that influence import or subsequent silencing. This hypothesis is supported by the different genetic requirements reported for silencing by neuronal *gfp*-dsRNA compared to other sources of *gfp*-dsRNA (*Ravikumar et al., 2019*). Alternatively, these preferences could reflect universal constraints for silencing using any extracellular dsRNA (e.g. expression patterns of factors that promote the import or processing of dsRNA). Furthermore, for both neuronal and ingested dsRNA it is unknown if additional extracellular molecules impact the uptake of dsRNA into cells. Regardless of the presence of such unknown molecules that could be released from neurons to a different degree upon oxidative damage, the gene regulatory effects of the released neuronal dsRNA provide convenient readouts that can be analyzed to understand neuronal damage and its consequences in animals. Pathologies of the central nervous system in humans, including cancer, stroke, multiple sclerosis, neurodegenerative disease, and brain injury, have been associated with extracellular RNAs detected in circulation (reviewed in *Tielking et al., 2019*), although their origins and regulatory consequences, if any, remain unknown. While the physiological conditions that promote secretion of dsRNA from *C. elegans* neurons are not known, the discovery that oxidative damage of neurons can enhance the secretion of dsRNA suggests that disruption of cell structures by oxidative damage (e.g. membrane integrity) or initiation of cellular processes that repair oxidative damage (e.g. through ejection of damaged macromolecules; *Melentijevic et al., 2017*) also promote the release of dsRNA.

The trafficking of extracellular dsRNA from parent to progeny has spatial specificity, as evidenced by more silencing within the proximal germline (*Figure 2*), temporal specificity, as evidenced by the need for dsRNA beyond the fourth larval stage (*Marré et al., 2016*; *Wang and Hunter, 2017*; *Figure 1*), and substrate specificity, as evidenced by the differential requirements for 50 bp dsRNA with 5′ OH versus a mix of longer dsRNAs with 5′ triphosphates (*Figure 3*). The observed difference in the extent of silencing by these dsRNAs delivered into parental circulation cannot be attributed to differential engagement of the nuclear Argonautes HRDE-1 and NRDE-3 in progeny (*Figure 3—figure supplement 2*), but rather could be the result of differences in the ability of each type of dsRNA to bind to upstream factors in the RNA interference pathway (e.g. RDE-4), differences in the stability of each type of dsRNA, and/or differences in the intergenerational transport of each type of dsRNA. Drivers of temporal specificity during development include changes in the uptake of dsRNA into the intestine, distribution of dsRNA to other tissues from the intestine, import of dsRNA into the germline, and availability of RNA silencing factors within the germline. Proteins mediating dsRNA transport or subsequent silencing also differ in their availability during development and in their affinities for different substrates. For example, SID-1 was not detectable in the developing larval germline but was detected in the adult germline (*Figure 4*) and has an extracellular domain

that binds dsRNA (*Li et al., 2015*) but could prefer dsRNA molecules with 5′ phosphates and/or long dsRNA. Another consideration for substrate specificity is the composition of 'mixed dsRNAs' generated through in vitro transcription versus defined short dsRNAs. In addition to a variety of lengths, mixed dsRNAs could also include a variety of undefined secondary structures. Such structures could recruit or titrate away RNA-binding proteins in addition to the dsRNA structures engaging the canonical RNAi pathway, resulting in mixed mechanisms of silencing. Additionally, although the selectivity uncovered here could apply to all dsRNA delivered into the extracellular space of *C. elegans* from any source (see *Table 1* for constraints on intergenerational dsRNA transport), the chemistry of the delivered dsRNA could be modified by yet unidentified enzymes in vivo to overcome these requirements. Tracking labeled dsRNA with diverse chemistries from parental circulation to progeny could allow correlation of differences observed in progeny silencing to differences in intergenerational trafficking.

## Physiological role(s) of SID-1

The germline is a major site of dsRNA import in *C. elegans* as evidenced by three key observations: the expression of SID-1 in the germline (*Figure 4*), heritable misregulation of germline genes in *sid-1(-)* animals (*Figures 5 and 6*), and accumulation of fluorescently labeled dsRNA from the extracellular space in the germline (*Marré et al., 2016*; *Wang and Hunter, 2017*). As a result, *sid-1(-)* animals could have a defect in the germline that is detectable only under conditions that promote dsRNA transport (e.g. oxidative damage). Multiple physiological defects in the germline and soma of *sid-1(-)* animals have been reported but have not been widely reproduced, have only been characterized within single generations, or have not been attributed to any specific *sid-1*-dependent gene(s). These include defects in animals with some misfolded proteins in the endoplasmic reticulum (*Long et al., 2014*), in animals exiting the dauer stage (*Ow et al., 2018*; *Kadekar and Roy, 2019*), in animals exposed to pathogenic *P. aeruginosa* (*Palominos et al., 2017*; *Moore et al., 2019*; *Kaletsky et al., 2020*), in animals exposed to an odorant (*Posner et al., 2019*), in intestinal cells that develop in the presence of a multi-copy transgene (*Ohno and Bao, 2022*), and in animals that overexpress α-synuclein (*Gaeta et al., 2022*). RNA-seq experiments in this study and comparisons to those of previous studies suggest that genetic background-dependent changes can obscure genuine *sid-1*-dependent changes (see *Figure 4—figure supplement 1*, *Figure 5—figure supplement 1* and *Technical comments* in Materials and methods), raising caution in the interpretation of putative *sid-1*-dependent defects. Comparing multiple *sid-1* mutants generated using genome editing with animals in which the mutated sequence has been reverted to wild-type sequence in the same genetic background could provide a firmer basis for the identification of additional *sid-1*-dependent processes while avoiding the potential for mating-induced changes in gene regulation (*Devanapally et al., 2021*) that can result when genetic crosses are used instead of genome editing.

## Retrotransposon-encoded protein regulators

Genes expressed within the germline are likely regulated by positive feedback loops that are required to continually produce factors for maintaining germline immortality and for preserving form and function across generations (*Jose, 2020*; *Chey and Jose, 2022*). Thus, germline genes could be particularly vulnerable to heritable epigenetic changes, where deviations in the expression levels of a gene that is regulated by or is part of such feedback loops has the potential to become permanent in descendants. Prior data on *sdg-1* mRNA (*Figure 5F*) and the SDG-1 protein (*Placentino et al., 2021*; *Price et al., 2021*) suggested the hypothesis that reduction of *sdg-1* RNA via SID-1 alters the amount of SDG-1 protein, which could interact with components of germ granules to mediate RNA regulation within the germline of wild-type animals. Our analysis of *sdg-1* expression suggests that it is part of a regulatory architecture that is susceptible to heritable epigenetic changes through the perturbation of RNA regulation (*Figures 5–7*). Such architectures within the germline could be exploited by 'selfish' genetic elements such as retrotransposons to persist across evolution if one of these elements also include genes encoding a regulator. In support of a wider use of such a strategy, a paralog of SDG-1, ZK262.8 (*Figure 6—figure supplement 1B*), is also encoded by a gene located within a retrotransposon and its loss along with that of the miRNA-associated Argonaute ALG-2 was reported to be synthetic lethal (*Tops et al., 2006*).

## Buffering of heritable epigenetic change by extracellular dsRNA

Loss of SID-1 weakly inhibits silencing by intracellular dsRNA (*Marré et al., 2016*; *Wang and Hunter, 2017*) but enhances the initiation of mating-induced silencing within the germline (*Figure 7B*). However, loss of SDG-1 alone does not account for these effects (*Figure 7A and B*), suggesting that additional SID-1-dependent genes could be involved. Furthermore, the ultimate consequence of SID-1 loss could be different in different isolates as exemplified by the differential changes in *cls-3* in *sid-1(jam80[nonsense])* and *sid-1(jam113[deletion])* animals (*Table 2*). Indeed, the levels of SDG-1 also vary in different isolates of *sid-1(-)* animals (*Figure 6*), suggesting that in the absence of SID-1 the *sdg-1* gene becomes prone to heritable epigenetic changes, resulting in either high or low expression states being stabilized in different isolates. Together these observations suggest that SID-1-dependent import of extracellular dsRNA could play a role in buffering against heritable epigenetic changes in the germline. Such buffering could be necessary to maintain levels of gene expression within a particular range for each regulatory context. Given the enrichment of SDG-1 protein with perinuclear germ granules (*Figure 7C*) and the maintenance of heritable changes in *sdg-1::mCherryΔpi* expression by a DEPS-1-dependent process (*Figure 6G*), buffering against changes in gene expression could involve both RNA- and protein-based regulation that tunes the function of perinuclear germ granules. We therefore speculate that one role for extracellular RNAs that enter germ cells in other systems (e.g. tRNA fragments in mammals; *Sharma et al., 2016*; *Chen et al., 2016*; *Sharma et al., 2018*) could be to similarly buffer against heritable changes in gene expression.

## Materials and methods

### Strains and oligonucleotides

All strains (listed in *Table 3*) were cultured on Nematode Growth Medium (NGM) plates seeded with 100 µl of OP50 *E. coli* at 20 °C and strains made through mating were generated using standard methods (*Brenner, 1974*). Oligonucleotides used are in *Table 4* (for genotyping *sid-1(qt9)*: P1-P2, *ttTi5605*: P3-P5, *eri-1(mg366)*: P6-P7, *sid-1(tm2700)*: P8-P10, *hrde-1(tm1200)*: P11-P13, *nrde-3(tm1116)*: P14-P16, and *rde-4(ne301)*: P156-P157). Strains made through mating existing mutant strains and genotyping using the above primers are listed below.

#### To create *gtbp-1::gfp* animals with *hrde-1(tm1200)* in the background

AMJ577 (*Devanapally et al., 2015*) was crossed with JH3197 males to create AMJ1220 and one other independent isolate.

#### To create *gtbp-1::gfp* animals with *nrde-3(tm1116)* in the background

JH3197 was crossed with WM156 males to create AMJ1383.

### Transgenesis

Animals were transformed with plasmids and/or PCR products using microinjection (*Mello et al., 1991*) to generate extrachromosomal arrays or single-copy transgenes. All plasmids were purified from bacterial culture using QIAprep Spin Miniprep Kit (QIAGEN) and all PCR products were generated with Phusion High-Fidelity DNA Polymerase (New England BioLabs) and purified using NucleoSpin Gel and PCR Clean-up Kit (Macherey-Nagel).

#### To express *sid-1::DsRed* in the muscle from an integrated array

pAJ53a (*myo-3p::sid-1::DsRed::unc-54 3'UTR*, made by AMJ while in Hunter Lab, Harvard University) was generated by amplifying part of *sid-1* cDNA from pHC355 (*Jose et al., 2009*) with primers P27 and P18, *DsRed* and *unc-54 3'UTR* from pHC183 (*Winston et al., 2002*) with primers P17 and P30, fusing the fragments using PCR with primers P30 and P31, and then cloning the fusion product into the pHC355 vector backbone using the restriction enzymes NruI and EagI. pAJ53a (40 ng/µl) was then injected into HC196 and animals expressing DsRed were isolated. AMJ3 was isolated as a spontaneous integrant. AMJ3 males were then crossed with AMJ308 hermaphrodites to generate AMJ327.

**Table 3.** Strains.

| Strains | Genotype |
| --- | --- |
| AMJ3 | *sid-1(qt9) V; jamIs2[myo-3p::sid-1::gfp]* |
| AMJ308 | *ccIs4251[myo-3p::gfp::lacZ::nls & myo-3p::mito-gfp & d py-20(+)] I; sid-1(qt9) V* |
| AMJ327 | *ccIs4251[myo-3p::gfp::lacZ::nls & myo-3p::mito-gfp & dpy-20(+)] I; sid-1(qt9) V; jamIs2[myo-3p::sid-1 cDNA::DsRed]* |
| AMJ471 | *jamEx140[rgef-1p::gfp-dsRNA & myo-2p::DsRed]* |
| AMJ477 | *qtEx136[rgef-1p::unc-22-dsRNA & rgef-1p::DsRed]* |
| AMJ576 | *jamSi12[mex-5p::sid-1::DsRed::sid-1 3'UTR]; unc-119(ed3) III; sid-1(qt9) V* |
| AMJ577 | *hrde-1(tm1200[4 X outcrossed]) III* |
| AMJ581 | *oxSi487[mex-5p::mCherry::h2b::gfp::h2b & Cbr-unc-119(+)] dpy-2(e8) II; unc-119(ed3) III* |
| AMJ592 | *hrde-1(tm1200) III; jamEx140[rgef-1p::gfp-dsRNA & myo-2p::DsRed]* |
| AMJ602 | *oxSi487[mex-5p::mCherry::h2b::gfp::h2b & Cbr-unc-119(+)] dpy-2(e8) II; unc-119(ed3) hrde-1(tm1200) III* |
| AMJ706 | *sid-1(qt9) V; jamEx193[myo-3p::sid-1::gfp]* |
| AMJ819 | *eri-1(mg366) gtbp-1(ax2053[gtbp-1::gfp]) IV* |
| AMJ837 | *jamEx209[rgef-1p::PH::miniSOG & myo-2p::DsRed]* |
| AMJ936 | *jamEx210[rgef-1p::PH::miniSOG & rgef-1p::DsRed]* |
| AMJ1007 | *eri-1(mg366) IV; jamEx213[rgef-1p::PH::miniSOG & rgef-1p::bli-1-dsRNA & myo-2p::DsRed]* |
| AMJ1009 | *eri-1(mg366) gtbp-1(ax2053[gtbp-1::gfp]) IV; jamEx214 [rgef-1p::PH::miniSOG & rgef-1p::gfp-dsRNA & myo-2p::DsRed]* |
| AMJ1019 | *jamSi36[rgef-1p::PH::miniSOG & Cbr-unc-119(+)] II; unc-119(ed3) III* |
| AMJ1108 | *eri-1(mg366) IV; sid-1(qt9) V; jamEx213[rgef-1p::PH:: miniSOG & rgef-1p::bli-1-dsRNA & myo-2p::DsRed]* |
| AMJ1114 | *sid-1(qt9) V; jamEx213[rgef-1p::PH::miniSOG & rgef-1 p::bli-1-dsRNA & myo-2p::DsRed]* |
| AMJ1120 | *rme-2(jam71[deletion]) IV; sid-1(qt9) V* |
| AMJ1123 | *jamEx213[rgef-1p::PH::miniSOG & rgef-1p::bli-1-dsRNA & myo-2p::DsRed]* |
| AMJ1131 | *rme-2(jam71[deletion]) IV* |
| AMJ1134 | *jamEx214[rgef-1p::PH::miniSOG & rgef-1p::gfp-dsRNA & myo-2p::DsRed]* |
| AMJ1146 | *oxSi487[Pmex-5::mCherry::h2b::gfp::h2b]; unc-119(ed9) III; rme- 2(jam71[deletion]) IV* |
| AMJ1204 | *rme-2(jam71[del]) IV; jamEx140[Prgef-1:: gfp-dsRNA & Pmyo- 2::DsRed]* |
| AMJ1151 | *sid-1(tm2700) V; jamEx213[rgef-1p::PH::miniSOG & rgef-1p::bli-1-dsRNA & myo-2p::DsRed]* |
| AMJ1153 | *sid-1(tm2700)[3 X outcrossed] V* |
| AMJ1159 | *sid-1(jam80[nonsense]) V* |
| AMJ1173 | *eri-1(mg366) IV; sid-1(tm2700) V; jamEx213[rgef-1p ::PH::miniSOG & rgef-1p::bli-1-dsRNA & myo-2p::DsRed]* |
| AMJ1217 | *sid-1(jam86[revertant]) V* |
| AMJ1220 | *hrde-1(tm1200) III; gtbp-1(ax2053[gtbp-1::gfp]) IV* |
| AMJ1280 | *sid-1(jam115[sid-1::wrmScarlet13]) V* |

*Table 3 continued on next page*

*Table 3 continued*

| Strains | Genotype |
| --- | --- |
| AMJ1281 | *rme-2(jam116[rme-2::wrmScarlet13]) IV* |
| AMJ1282 | *sid-1(jam117[sid-1::wrmScarlet]) V* |
| AMJ1284 | *rme-2(jam119[rme-2::wrmScarlet]) IV* |
| AMJ1312 | *sid-1(jam80[nonsense]) V; jamEx214[rgef-1p::PH ::miniSOG & rgef-1p::gfp-dsRNA & myo-2p::DsRed]* |
| AMJ1323 | *sid-1(jam112[sid-1::tetracycline-K4-aptazyme::3'UTR]) V* |
| AMJ1324 | *sid-1(jam113[deletion]) V* |
| AMJ1330 | *sid-1(jam112[sid-1::tetracycline-K4-aptazyme::3'UTR]) V; qtEx136[rgef-1p::unc-22-dsRNA & rgef-1p::DsRed]* |
| AMJ1332 | *sid-5(jam122[deletion]) X* |
| AMJ1350 | *sid-1(jam112[sid-1::tetracycline-K4-aptazyme::3'UTR]) V; jamEx140[rgef-1p::gfp-dsRNA & myo-2p::DsRed]* |
| AMJ1355 | *gtbp-1(ax2053[gtbp-1::gfp]) IV; sid-1(jam112[sid-1: :tetracycline-K4-aptazyme::3'UTR]) V* |
| AMJ1365 | *hrde-1(tm1200) III; sid-1(jam117[sid-1::wrmScarlet]) V* |
| AMJ1366 | *rme-2(jam71[deletion]) IV; sid-1(jam113[deletion]) V* |
| AMJ1367 | *sid-1(jam113[deletion]) V; sid-5(jam122[deletion]) X* |
| AMJ1368 | *sid-2(jam134[deletion]) III* |
| AMJ1372 | *W09B7.2/F07B7.2(jam137[W09B7.2/F07B7.2::mCherryΔpi]) V* |
| AMJ1380 | *sid-2(jam134[deletion]) III; sid-1(jam113[deletion]) V* |
| AMJ1383 | *gtbp-1(ax2053[gtbp-1::gfp]) IV; nrde-3(tm1116) X* |
| AMJ1389 | *sid-1(jam150[nonsense]) W09B7.2/F07B7.2(jam137 [W09B7.2/F07B7.2::mCherryΔpi]) V* |
| AMJ1399 | *sid-1(jam157[nonsense]) V* |
| AMJ1405 | *sid-1(jam163[revertant]) V* |
| AMJ1406 | *sid-1(jam164[revertant]) V* |
| AMJ1407 | *sid-1(jam165[revertant]) V* |
| AMJ1408 | *sid-1(jam166[revertant]) V* |
| AMJ1409 | *sid-1(jam167[revertant]) V* |
| AMJ1410 | *sid-1(jam168[revertant]) V* |
| AMJ1412 | *sid-1(jam170[revertant]) W09B7.2/F07B7.2(jam137 [W09B7.2/F07B7.2::mCherryΔpi]) V* |
| AMJ1413 | *sid-1(jam171[revertant]) W09B7.2/F07B7.2(jam137 [W09B7.2/F07B7.2::mCherryΔpi]) V* |
| AMJ1438 | *sid-1(jam172[sid-1 N-term::mCherryΔpi::sid-1 C-term]) V* |
| AMJ1442 | *sid-1(jam173[nonsense]) W09B7.2/F07B7.2(jam137 [W09B7.2/F07B7.2::mCherryΔpi]) V* |
| AMJ1443 | *sid-1(jam174[nonsense]) W09B7.2/F07B7.2(jam137 [W09B7.2/F07B7.2::mCherryΔpi]) V* |
| AMJ1444 | *sid-1(jam175[nonsense]) W09B7.2/F07B7.2(jam137[W09B7.2/F07B7.2::mCherryΔpi]) V* |
| AMJ1445 | *sid-1(jam176[nonsense]) W09B7.2/F07B7.2(jam137[W09B7.2/F07B7.2::mCherryΔpi]) V* |
| AMJ1446 | *sid-1(jam177[nonsense]) W09B7.2/F07B7.2(jam137 [W09B7.2/F07B7.2::mCherryΔpi]) V* |

*Table 3 continued*

| Strains | Genotype |
| --- | --- |
| AMJ1447 | *W09B7.2/F07B7.2(jam137[W09B7.2/F07B7.2::mCherryΔpi]) rde-1(jam178[nonsense]) V* |
| AMJ1448 | *W09B7.2/F07B7.2(jam137[W09B7.2/F07B7.2::mCherryΔpi]) rde-1(jam179[nonsense]) V* |
| AMJ1449 | *W09B7.2/F07B7.2(jam137[W09B7.2/F07B7.2::mCherryΔpi]) V; sid-3(jam180[nonsense]) X* |
| AMJ1450 | *W09B7.2/F07B7.2(jam137[W09B7.2/F07B7.2::mCherryΔpi]) V; sid-3(jam181[nonsense]) X* |
| AMJ1451 | *deps-1(jam182[nonsense]) I; W09B7.2/F07B7.2(jam137[W09B7.2/F07B7.2::mCherryΔpi]) V* |
| AMJ1452 | *deps-1(jam183[nonsense]) I; W09B7.2/F07B7.2(jam137[W09B7.2/F07B7.2::mCherryΔpi]) V* |
| AMJ1479 | *sid-1(jam189[deletion]) W09B7.2/F07B7.2(jam137[W09B7.2/F07B7.2::mCherryΔpi]) V* |
| AMJ1480 | *sid-1(jam190[deletion]) W09B7.2/F07B7.2(jam137[W09B7.2/F07B7.2::mCherryΔpi]) V* |
| AMJ1481 | *sid-1(jam191[deletion]) W09B7.2/F07B7.2(jam137[W09B7.2/F07B7.2::mCherryΔpi]) V* |
| AMJ1482 | *sid-1(jam192[deletion]) W09B7.2/F07B7.2(jam137[W09B7.2/F07B7.2::mCherryΔpi]) V* |
| AMJ1485 | *sid-1(jam195[sid-1 N-term::linker::mCherryΔpi::sid-1 C-term]) V* |
| AMJ1504 | *oxSi487[mex-5p::mCherry::h2b::gfp::h2b & Cbr-unc-119 (+)] dpy-2(e8) II; unc-119(ed3) III; sid-1(jam80[nonsense]) V* |
| AMJ1542 | *gtbp-1(jam210[gtbp-1::gfp::tetracycline-K4-aptazyme::3'UTR]) IV* |
| AMJ1574 | *deps-1(jam229[nonsense]) I; sid-1(jam170[revertant]) W09B7.2/F07B7.2 (jam137[W09B7.2/F07B7.2::mCherryΔpi]) V* |
| AMJ1575 | *deps-1(jam230[nonsense]) I; sid-1(jam170[revertant]) W09B7.2/F07B7.2 (jam137[W09B7.2/F07B7.2::mCherryΔpi]) V* |
| AMJ1577 | *W09B7.2/F07B7.2(jam232[deletion]) V* |
| AMJ1612 | *W09B7.2/F07B7.2(jam241[deletion]) V* |
| AMJ1613 | *W09B7.2/F07B7.2(jam242[deletion]) V* |
| AMJ1615 | *W09B7.2/F07B7.2(jam244[sdg-1 ORF deleted from jam137]) V* |
| AMJ1616 | *W09B7.2/F07B7.2(jam245[sdg-1 ORF deleted from jam137]) V* |
| AMJ1617 | *W09B7.2/F07B7.2(jam246[sdg-1 ORF deleted from jam137]) V* |
| AMJ1662 | *znfx-1(gg544[3xflag::gfp::znfx-1]) II; W09B7.2/F07B7.2(jam137[W09B7.2/F07B7.2::mCherryΔpi]) V* |
| AMJ1766 | *rde-4(ne301) III; W09B7.2/F07B7.2(jam244[sdg-1 ORF deleted from jam137]) V* |
| AMJ1767 | *rde-4(ne301) III; W09B7.2/F07B7.2(jam244[sdg-1 ORF deleted from jam137]) V* |
| AMJ1770 | *W09B7.2/F07B7.2(jam244[sdg-1 ORF deleted from jam137 1 X outcrossed]) V* |
| DH1390 | *rme-2(b1008) IV* |
| EG4322 | *ttTi5605 II; unc-119(ed9) III* |
| EG6787 | *oxSi487[mex-5p::mCherry::h2b::gfp::h2b & Cbr-unc-119(+)] II; unc-119(ed3) III* |
| FX02700 | *sid-1(tm2700) V* |
| FX15992 | *sid-1(tm2700) V; tmIs1005[sid-1(+) & vps-45 mini]* |
| GR1373 | *eri-1(mg366) IV* |
| HC196 | *sid-1(qt9) V* |
| HC731 | *eri-1(mg366) IV; sid-1(qt9) V* |
| JH3197 | *gtbp-1(ax2053[gtbp-1::gfp]) IV* |
| N2 | wild type |
| WM49 | *rde-4(ne301) III* |
| YY916 | *znfx-1(gg544[3xflag::gfp::znfx-1]) II* |

**Table 4.** Oligonucleotides.

| Name | Sequence |
| --- | --- |
| P1 | caccttcgccaattatcacctc |
| P2 | cgtcagcttctgattcgacaac |
| P3 | ataaggagttccacgcccag |
| P4 | ctagtgagtcgtattataagtg |
| P5 | tgaagacgacgagccacttg |
| P6 | ggaacatatggggcattcg |
| P7 | cagacctcacgatatgtggaaa |
| P8 | gcttcacctgtcttatcactgc |
| P9 | cgcggcgactttggttaaatc |
| P10 | ggcttgacaaacgtcagcttc |
| P11 | tcatctcggtacctgtcgttg |
| P12 | agaggcggatacggaagaag |
| P13 | cataaccgtcgcttggcac |
| P14 | aatgggtgagatgggcttaag |
| P15 | gcacttcgatatttcgcgccaa |
| P16 | gaaccaatgtggcacgaaac |
| P17 | gcaaaacttcgattaacattttcatggcctcctccgagaacg |
| P18 | cgttctcggaggaggccatgaaaatgttaatcgaagttttgc |
| P19 | ggtaccctctagtcaaggcctatagaaaagttgaaatatcagtttttaaaaa |
| P20 | cacgaatcattctctgtctgaaacattcaattg |
| P21 | cagacagagaatgattcgtgtttatttgataattttaatg |
| P22 | cggaggaggccatgaaaatgttaatcgaagttttgc |
| P23 | taacattttcatggcctcctccgagaac |
| P24 | aattactctactacaggaacaggtggtgg |
| P25 | gttcctgtagtagagtaattttgttttccctatc |
| P26 | ggctacgtaatacgactcacagtggctgaaaatttatgc |
| P27 | gagcagcagaatacgagctc |
| P28 | gaaaagttcttctcctttactcatgaaaatgttaatcgaagttttgc |
| P29 | gcaaaacttcgattaacattttcatgagtaaaggagaagaacttttc |
| P30 | ctctcagtacaatctgctctg |
| P31 | gaatacgagctcagaactcg |
| P32 | atgccgcatagttaagccag |
| P33 | atcgacgacgacgacgatcagcagtaaagaagcttgcatgcctgcag |
| P34 | atgttgaagagtaattggacgtcatccatccagcagcac |
| P35 | gtccaattactcttcaacatcccta |
| P36 | ctttactgctgatcgtcg |
| P37 | tctctccctaggcacaacgatggatacgctaac |
| P38 | gagagacctaggcacgatgagcatgatttgacg |
| P39 | atttaggtgacactatagctaccataggcaccacgaggttttagagctagaaatagcaag |

*Table 4 continued on next page*

*Table 4 continued*

| Name | Sequence |
| --- | --- |
| P40 | gcaccgactcggtgcca |
| P41 | cacttgaacttcaatacggcaagatgagaatgactggaaaccgtaccgcatgcggtgcctatggtagcggagcttcacatggcttcagaccaacagccta |
| P42 | atttaggtgacactatagcaaggcgcatggttctcagttttagagctagaaatagcaag |
| P43 | atttaggtgacactatagcaactttcatgcaataaatgtttagagctagaaatagcaag |
| P44 | ttctttcattcttttcataatctcactcaccatgatattgcatgaaagttgataatgtctactagtactg |
| P45 | aaacaccaacaacgcaatcc |
| P46 | tgacctcatcatctcctccag |
| P47 | tccgaatctgaaccacgaatg |
| P48 | atttaggtgacactatagcattcaatcgagactgcagttttagagctagaaatagcaag |
| P49 | agcctataatctatatcagcattcaatcaaggctacacggttacgatcaggttttgatggaaatgagggt |
| P50 | atttaggtgacactatagcattcaatcaaggctacagttttagagctagaaatagcaag |
| P51 | aagcctataatctatatcagcattcaatcgagactgcacggttacgatcaggttttgatggaaatgaggg |
| P52 | tgaaatatgaaaaaccggat |
| P53 | tcattaatacacgcaaaacttcgattaacattttcatggtcagcaagggagaggcagttatcaaggagttcatgcgtttcaaggtccaacgagcgttccga gggacgtcactccaccggaggaatggacgagctctacaagtagagtaattttgttttccctattcgtttcttcatatttcaacttttttctcctgcctta |
| P54 | actcggcttcttcggttcc |
| P55 | aacaccagatcactgcgtagag |
| P56 | aaggtccaacgagcgttccg |
| P57 | atggtcagcaagggagagg |
| P58 | cttgtagagctcgtccattcct |
| P59 | attgtgaacctggaaaaatg |
| P60 | tttcactatcagtggcttcacctgtcttatcactgcttcttgtatactgaacgacgttaaacacatctcactttaacatttagaaattaaaactcctcatcggtt tttcatatttcaacttttttctcctgccttaatacgtagcccatctctcatttcttcatgttttaagaactttctgaatctatgtaattagttgg |
| P61 | tttttggcacagttttttgct |
| P62 | ggaattagagactagagctt |
| P63 | cgtgtctctcacaacagccgtttctctaacagaaaaaccttcttttgttgatgtttgtctaaaatcgattttttcagcaagaaatcgagaaactgga acgagctttggtaagttttttgttcctcgaagtgtaaataattgagtaaaagctttcttattgaaaaaaaaaacgaatgttcaaattatgaagattgaaaaatg |
| P64 | tttcccgcgtactcctctc |
| P65 | ctaagaccaacatccaagctcg |
| P66 | tcacatttggcgaggagcca |
| P67 | aatcgaatgactccagcgaa |
| P68 | cagacgtttggctatacgcc |
| P69 | caactggtttcgtcagatcggcttccgcaccatttgccggtgtgatccgtttcgaaaatgatagtttattaatggtcagcaagggagaggcagttatcaaggagttcat gcgtttcaagttccgagggacgtcactccaccggaggaatggacgagctctacaagtgaattctactacaaaattactaaatcagatgtct |
| P70 | ctgctttgatggccgaatactg |
| P71 | aaacaaaaatatacaaatcg |
| P72 | ccttcgctacattggaaagc |
| P73 | catatgaaattttttaaataaagttgttttctaactgttcccaatattcttaaatcccattgaacagaatttcattttcaaaaccctgatattttcaggaatttttattccaataatatgatttt gaaaaactattaatcttacctgtgcatcaataaagatcttgtgagtatatcatcgatcacagtctccgatttgtctg |
| P74 | ggtcttacccattccaacatcg |
| P75 | ttcgctacattggaaagctgg |

*Table 4 continued*

| Name | Sequence |
| --- | --- |
| P76 | cacgcctatgttcccttgtc |
| P77 | ttcatgcgtttcaagttccg |
| P78 | tcgattaacattttctagagtaattttgtttcccaaacaaacaaaggcgcgtcctggattcgtacaaaacataccagatttcgatctggagaggtgaagaatacgaccacct gtacatccagctgatgagtcccaaataggacgaaacgcgctcaaacaaacaaactatccggtttttcatatttcaactttttctcct |
| P79 | tctcccacttgaatccctctg |
| P80 | ccaaatgttgagccagtcac |
| P81 | ttgaggaaatgcagacgctcgttatcgacctccagatggtctccaagggagagga |
| P82 | tgttattttgagggagccaaatgttgagccagtcagccactacctgatcccttgt |
| P83 | gctgaaggtggatagtgtctc |
| P84 | gagttcggaagtaaaccgtgg |
| P85 | tcaccatctgggaggtgttcacatttggcgaggagccataggtcggctgtcgagccatcgatgtgctcaa |
| P86 | agacgaaagggtgagaactttg |
| P87 | cgcgaggatatgcagttcac |
| P88 | agcattcaatcgagactgca |
| P89 | acaagaaggaaaaaggagaa |
| P90 | aatgcgggacaaaattagaagctttccgttctcccaaacaaacaaaggcgcgtcctggattcgtacaaaacataccagatttcgatctggagaggtgaagaata cgaccacctgtacatccagctgatgagtcccaaataggacgaaacgcgctcaaacaaacaaattttttccttcttgtaagaattgcacatccattag |
| P91 | cacatggtccttcttgagtttg |
| P92 | acggtgaggaaggaaaggag |
| P93 | agcattcaatcaaggctaca |
| P94 | cgaagtaaaacaattcatgt |
| P95 | gcttcgatctttaaaaagcgaagtaaaataatttatgtcagaacgggatggagaagatccagagccgaag |
| P96 | tggctcatggacgggaaag |
| P97 | ggaacaggcaacgagatgg |
| P98 | cgtggcacatactttccgttgttg |
| P99 | gtcatctccgacgagcac |
| P100 | ttccgttgttggcttcgttg |
| P101 | tgcacggcgtatcaaactg |
| P102 | ggccattgggagaacttcg |
| P103 | tgacggcctcttctacatatcg |
| P104 | ccgcaagtctctcctgtatg |
| P105 | gctgaaggtggatagtgtctc |
| P106 | attgctccgcaaatgtagtgg |
| P107 | gctgctcaagcaaatcgaatg |
| P108 | ttatcacggtggagaacagc |
| P109 | ttggtagggaatcggctgg |
| P110 | tcaaattgttgaagagatca |
| P111 | cagcagaaaatcaaattgttgaagagatcacagctatggtctccaagggagagga |
| P112 | cggtttccctcttctacgctcgtttcttgattttcgccactacctgatcccttgt |
| P113 | caacgggacatggatttgag |

*Table 4 continued on next page*

*Table 4 continued*

| Name | Sequence |
|------|----------|
| P114 | ttgaatttcccggtttccctc |
| P115 | tgttgaagagatcacagcta |
| P116 | cagcagaaaatcaaattgttga agagatcacagctggtggcggtggatcgggaggaggaggttcgggtggc ggaggcagtatggtctccaagggagaggaagataacatggctat |
| P117 | taatacgactcactatagg |
| P118 | cccacactaccatcggcgctac |
| P119 | cactcttactgctaccaacgcttctggaagcgacaaacat |
| P120 | atgtttgtcgcttccagaagcgttggtagcagtaagagtg |
| P121 | tcgttgttccaggagatcagaaaacagcaactgttccaaa |
| P122 | tttggaacagttgctgttttctgatctcctggaacaacga |
| P123 | acccacttcacagtcgattcactcaacaagggagatcatt |
| P124 | aatgatctcccttgttgagtgaatcgactgtgaagtgggt |
| P125 | tagaaaaaatgagtaaaggagaagaacttttcactggagt |
| P126 | actccagtgaaaagttcttctcctttactcatttttttcta |
| P127 | agtttgaaggtgataccttgttaatagaatcgagttaaa |
| P128 | tttaactcgattctattaacaagggtatccttcaaact |
| P129 | ggattacacatggcatggatgaactatacaaatgcccggg |
| P130 | cccgggcatttgtatagttcatccatgccatgtgtaatcc |
| P131 | acauuccagucaguggugaaccaacuccaacaauuacuuggacuuucgaa |
| P132 | uucgaaaguccaaguaauuguuggaguugguucaccacugacuggaaugu |
| P133 | ugguccuucuugaguuuguaacagcugcugggauuacacauggcauggau |
| P134 | auccaugccaugaguguaaucccagcagcuguuacaaacucaagaaggacca |
| P135 | 5'Atto 565-auccaugccaugaguguaaucccagcagcuguuacaaacucaagaaggacca |
| P136 | 5'Atto 488-ugguccuucuugaguuuguaacagcugcugggauuacacauggcauggau |
| P137 | aggcgacccgtgcggagccagacgtttggctatacgcctgaattcgattcgaaactaccatgaagagtgg |
| P138 | cgtttggctatacgccggg |
| P139 | tccgttgacagaggttacatgc |
| P140 | agcgtcttccagcagaaatg |
| P141 | cttcatggtagtttcgaatcgactt |
| P142 | gctaccataggcaccgcatg |
| P143 | ctggttgagcttctcattct |
| P144 | ccaaatgttgagccagtcac |
| P145 | tccgttttttcgaaacttttcgtaatattttttgtttcttcaattgatctcttgaatattcatcgtgaatta |
| P146 | gagttcggaagtaaaccgtgg |
| P147 | gctgaaggtggatagtgtctc |
| P148 | cgcagtacgcagagtgaac |
| P149 | gatggtctccaagggagagg |
| P150 | ttacagtaaaacagccggatcccaccgagaatggtctccaagggagaggaagataacatg |
| P151 | tctcccacttgaatccctctg |

*Table 4 continued on next page*

*Table 4 continued*

| Name | Sequence |
| --- | --- |
| P152 | atcgtcttgatcgacggaacac |
| P153 | ttgaggtggtttatctctggac |
| P154 | cttgtagttcccgtcatctttg |
| P155 | atttcgttctgattccgtgagg |
| P156 | ttcctgcaactttccgacc |
| P157 | gaacttcctgaaggcttcg |
| P158 | atcgtcttgatcgacggaacac |
| P159 | acccaggattcctccgtaag |
| P160 | gagttcggaagtaaaccgtgg |

### To express *sid-1::DsRed* in the germline from a single-copy transgene

The *mex-5* promoter was amplified from pJA252 (Addgene #21512) using the primers P19 and P20. The *sid-1* gene was amplified from N2 genomic DNA using the primers P21 and P22. The *DsRed* gene was amplified from pAJ53a (*myo-3p::sid-1(+)::DsRed::unc-54 3'UTR*; made by AMJ and Tessa Kaplan while in Hunter Lab, Harvard University) using the primers P23 and P24. The *sid-1* 3'UTR was amplified using the primers P25 and P26. Using NEBuilder HiFi DNA Assembly (New England BioLabs), these four amplicons were placed into pCFJ151 (Addgene #19330) digested with AflII (New England BioLabs) and treated with CIP (New England BioLabs) to generate pJM10. pJM10 (50 ng/µl) and the coinjection markers pCFJ601 (50 ng/µl), pMA122 (10 ng/µl), pGH8 (10 ng/µl), pCFJ90 (2.5 ng/µl), and pCFJ104 (5 ng/µl) (plasmids described in *Frøkjær-Jensen et al., 2012*) were injected into the germline of adult EG4322 animals. One transgenic line was isolated as described previously (*Frøkjær-Jensen et al., 2012*) and crossed with HC196 males to generate AMJ576. The integration of *mex-5p::sid-1(+)::DsRed::sid-1 3'UTR* in AMJ576 was verified by genotyping with primers P3-P5 and Sanger sequencing of the insertion.

### To express *sid-1::gfp* in the muscle from an extrachromosomal array

pTK2 (*myo-3p::sid-1::gfp*, made by AMJ and Tessa Kaplan while in Hunter Lab, Harvard University) was constructed by amplifying part of *sid-1* cDNA from pHC355 (*Jose et al., 2009*) with primers P27 and P28, *gfp* and *unc-54 3'UTR* from pPD95.75 (Addgene #1494) using primers P29 and P30, and then fusing the fragments using PCR with primers P30 and P31 and cloning the product into the pHC355 vector backbone using the restriction enzymes NruI and EagI. pTK2 (10 ng/µl) was injected into HC196 and animals expressing GFP were isolated as AMJ706.

### To express *PH::miniSOG* in neurons from an extrachromosomal array

pNMS03 (*rgef-1p::PH::miniSOG::unc-54 3'UTR*) was generated by amplifying the vector backbone of pHC337 excluding the *gfp*-dsRNA hairpin sequence using primers P35 and P36, and assembling it with *PH::miniSOG(Q103L)* amplified from pCZGY2851 (gift from Andrew Chisholm) with primers P33 and P34 using NEBuilder HiFi DNA Assembly (New England BioLabs). pNMS03 (40 ng/µl) was injected into N2 animals with pHC448 (*Jose et al., 2011*) (*myo-2p::DsRed2::unc-54 3'UTR*; 40 ng/µl) as a coinjection marker to create AMJ837 and two other isolates.

pNMS03 (40 ng/µl) was also injected into N2 animals with PCR products forming *rgef-1p::DsRed* (40 ng/µl) generated previously (*Jose et al., 2011*) as a coinjection marker to create AMJ936 and two other isolates.

### To express *PH::miniSOG* in neurons from a single-copy transgene

pNMS05 (*rgef-1p::PH::miniSOG::unc-54 3'UTR* with *ttTi5605* homology arms and *Cbr-unc-119(+)*) was generated by amplifying the transgene *rgef-1p::PH::miniSOG::unc-54 3'UTR* from pNMS03 with primers P37 and P38 containing AvrII restriction sites and cloning the fragment into pCFJ151 after



AvrII (New England BioLabs) digestion. pNMS05 (50 ng/µl) and the coinjection markers pCFJ601 (50 ng/µl), pMA122 (10 ng/µl), pGH8 (10 ng/µl), pCFJ90 (2.5 ng/µl), and pCFJ104 (5 ng/µl) (plasmids described in *Frøkjær-Jensen et al., 2012*) were injected into the germline of adult EG4322 animals. One transgenic line was isolated as described previously (*Frøkjær-Jensen et al., 2012*) and designated as AMJ1019. The integration of *rgef-1p::PH::miniSOG::unc-54 3'UTR* in AMJ1019 was verified by genotyping with primers P3-P5 and Sanger sequencing of the insertion.

### To express *PH::miniSOG* with *bli-1*-dsRNA in neurons from an extrachromosomal array

pNMS03 (40 ng/µl) was injected with *rgef-1p::bli-1-sense* (40 ng/µl) and *rgef-1p::bli-1-antisense* (40 ng/µl) PCR products generated previously (*Raman et al., 2017*) into GR1373 animals with pHC448 (*myo-2p::DsRed2::unc-54 3'UTR*) as a coinjection marker (40 ng/µl) to create AMJ1007 and one other independent isolate. AMJ1007 was crossed with HC731 males to create AMJ1108 and two other isolates. AMJ1108 was crossed with HC196 males to create AMJ1114 and one other isolate. AMJ1007 was crossed with N2 males to create AMJ1123 and one other isolate. AMJ1123 males were crossed with 3X outcrossed FX02700 (designated as AMJ1153) to create AMJ1151 and two other isolates. AMJ1151 was crossed with GR1373 males to create AMJ1173 and two other isolates.

### To express *PH::miniSOG* with *gfp*-dsRNA in neurons from an extrachromosomal array

pNMS03 (40 ng/µl) and pHC337 (*rgef-1p::gfp-dsRNA::unc-54 3'UTR*; 40 ng/µl) were injected into AMJ819 (*Devanapally et al., 2021*) with pHC448 (*myo-2p::DsRed2::unc-54 3'UTR*; 40 ng/µl) as a coinjection marker to create AMJ1009 and one other independent isolate. AMJ1009 was crossed with N2 males to create AMJ1134. AMJ1159 was crossed with AMJ1134 males to create AMJ1312 and two other isolates.

All other transgenes were generated previously (*ccIs4251 Fire et al., 1998*); *oxSi487* (*Frøkjær-Jensen et al., 2012*); *tmIs1005* (*Kage-Nakadai et al., 2014*); *jamEx140* (*Devanapally et al., 2015*); *qtEx136* (*Ravikumar et al., 2019*).

## Cas9-mediated genome editing

Genome editing was performed by injecting nuclear-localized Cas9 (PNA Bio) preincubated at 37 °C for 10 min with either a single-guide RNA (sgRNA) generated by in vitro transcription (SP6 RNA Polymerase, New England BioLabs) or hybridized crRNA/tracrRNA (IDT), as well as an oligonucleotide or PCR-amplified homology repair template, into the *C. elegans* distal gonad. Screening for plates with successfully edited F1 animals was performed using either *dpy-10* co-CRISPR (*Arribere et al., 2014*; *Paix et al., 2015*) or the pRF4 plasmid used as a co-injection marker (*Dokshin et al., 2018*). All plasmids were purified from bacterial culture using QIAprep Spin Miniprep Kit (QIAGEN) and all PCR products were generated with Phusion High-Fidelity DNA Polymerase (New England BioLabs) and purified using NucleoSpin Gel and PCR Clean-up Kit (Macherey-Nagel). Alleles generated by genome editing are schematized in *Figure 4A* (*sid-1*), *Figure 5A* (*sid-1*), *Figure 1—figure supplement 1D* (*rme-2*), *Figure 2—figure supplement 2* (*deps-1*, *mut-16*, *sid-2*, *rme-2*, *sid-1*, *rde-1*, *sid-5*, and *sid-3*), *Figure 4—figure supplement 2A* (*sid-1*), and *Figure 6—figure supplement 1C* (*W09B7.2/F07B7.2* (*sdg-1*)), and oligonucleotides used are in *Table 4*.

### To delete the *rme-2* coding sequence

Two sgRNAs targeting the start and end of the *rme-2* coding sequence were in vitro transcribed from a SP6 transcription template amplified from pDD162 (Addgene #47549) using primers P42 (start sgRNA) or P43 (end sgRNA) as forward primers and P40 as a universal reverse primer. An sgRNA targeting *dpy-10* for co-CRISPR was also in vitro transcribed using a similar template amplified from pDD162 with primers P39 and P40. All sgRNAs were purified using organic extraction, were precipitated using ethanol, and resuspended in water prior to injection. Injection into HC196 with all sgRNAs, Cas9 and the homology repair templates for *rme-2* (P44) and *dpy-10* (P41), and screening for edited animals were performed as described above. Genotyping for *rme-2(deletion)* was performed using a triplex PCR with primers P45-P47 to isolate AMJ1120 and one other isolate and the *rme-2* deletion

was verified using Sanger sequencing. AMJ1120 was crossed with N2 males to isolate AMJ1131. AMJ1131 males were crossed with EG6787 and AMJ471 hermaphrodites to isolate AMJ1146 and AMJ1204 animals, respectively.

### To delete the *sid-1* coding sequence

Injection of crRNAs targeting the start (P59) and end (P52) of the *sid-1* coding sequence (IDT), tracrRNA, Cas9, a *sid-1(deletion)* homology repair template (P60) and pRF4 into N2 and AMJ1372, and subsequent screening were performed as described above. Genotyping for *sid-1(deletion)* was performed using triplex PCR with primers P8, P54 and P55 to isolate AMJ1324 and one other independent isolate from N2 and AMJ1479-AMJ1482 from AMJ1372. The *sid-1* deletion was verified by Sanger sequencing in all isolates. AMJ1324 was crossed with AMJ1131 males to create AMJ1366.

### To delete the *sid-2* coding sequence

Injection of crRNAs targeting the start (P71) and end (P72) of the *sid-2* coding sequence (IDT), tracrRNA, Cas9, a *sid-2(deletion)* homology repair template (P73) and pRF4 into N2, and subsequent screening were performed as described above. Genotyping for *sid-2(deletion)* was performed using triplex PCR with primers P74-P76 to isolate AMJ1368 and one other independent isolate. The *sid-2* deletion was verified by Sanger sequencing in both isolates. AMJ1368 was crossed with AMJ1324 males to create AMJ1380.

### To delete the *sid-5* coding sequence

Injection of crRNAs targeting the start (P61) and end (P62) of the *sid-5* coding sequence (IDT), tracrRNA, Cas9, a *sid-5(deletion)* homology repair template (P63) and pRF4 into N2, and subsequent screening were performed as described above. Genotyping for *sid-5(deletion)* was performed using duplex PCR with primers P64 and P65 to isolate AMJ1332 and three other independent isolates. The *sid-5* deletion was verified by Sanger sequencing in all four isolates. AMJ1332 was crossed with AMJ1324 males to create AMJ1367.

### To introduce a nonsense mutation into *sid-1* coding sequence

An sgRNA was designed to introduce into *sid-1* a nonsense mutation mimicking the *qt9* allele (***Winston et al., 2002***). This sgRNA was in vitro transcribed from a SP6 transcription template amplified from pDD162 (Addgene #47549) using primers P48 and P40. An sgRNA targeting *dpy-10* for co-CRISPR was also in vitro transcribed using a similar template amplified from pDD162 with primers P39 and P40. Both sgRNAs were purified using organic extraction and were precipitated using ethanol prior to injection. Both sgRNAs, Cas9 and the homology repair templates for *sid-1(nonsense)* (P49) that includes a missense mutation (S155P) and nonsense mutation (R156*) downstream that would prevent recutting of edited DNA by Cas9, and for *dpy-10* (P41) were injected into N2. Screening for edited animals was performed as described above. Genotyping for *sid-1(nonsense)* was performed using a duplex PCR with primers P1 and P2 followed by restriction digestion with HpyCH4V to isolate AMJ1159. The nonsense mutation was confirmed using Sanger sequencing. AMJ1159 males were crossed with AMJ581 (***Devanapally et al., 2015***) to create AMJ1504 and two other independent isolates.

Injection of a crRNA with the same target sequence (P88) (IDT) as the sgRNA described above, tracrRNA, Cas9, the same *sid-1(nonsense)* homology repair template (P49) and pRF4 into N2 and AMJ1372 and subsequent screening were performed as described above. Genotyping for *sid-1(nonsense)* was performed using duplex PCR with primers P1 and P2 followed by restriction digestion with HpyCH4V to distinguish AMJ1399 from N2, and AMJ1389 and AMJ1442-AMJ1446 from AMJ1372. The nonsense mutation was verified using Sanger sequencing in all isolates.

### To revert the mutation in *sid-1(nonsense)* animals

An sgRNA was designed to revert the nonsense mutation described above back to wild-type *sid-1* sequence. The sgRNA was in vitro transcribed from a SP6 transcription template amplified from pDD162 (Addgene #47549) using primers P50 and P40. An sgRNA targeting *dpy-10* for co-CRISPR was also in vitro transcribed using a similar template amplified from pDD162 with primers P39 and

P40. Both sgRNAs were purified using organic extraction and were precipitated using ethanol prior to injection. Injection into AMJ1159 with both sgRNAs, Cas9 and the homology repair template for *sid-1(revertant)* (P51), which also reverted the missense mutation (S155P) and nonsense mutation (R156*) downstream of *sid-1(nonsense)* to wild-type sequence, and *dpy-10* (P41). Screening for edited animals was performed as described above. Genotyping for *sid-1(revertant)* was performed using duplex PCR with primers P1 and P2 followed by restriction digestion with HpyCH4V to isolate AMJ1217 and two other independent isolates. The revertant was verified using Sanger sequencing in all isolates.

Injection of a crRNA with the same target sequence (P93) (IDT) as the sgRNA described above, tracrRNA, Cas9, a *sid-1(revertant)* homology repair template (P51) and pRF4 into AMJ1389 and AMJ1399, and subsequent screening were performed as described above. Genotyping for *sid-1(revertant)* was performed using duplex PCR with primers P1 and P2 followed by restriction digestion with HpyCH4V to distinguish AMJ1412 and AMJ1413 from AMJ1389, and AMJ1405-AMJ1410 from AMJ1399. The revertant was verified using Sanger sequencing in all isolates.

### To tag *W09B7.2/F07B7.2* with *mCherry*

Injection of a crRNA with the target sequence listed as P80 (IDT), tracrRNA, Cas9, an *mCherry* sequence lacking piRNA binding sites amplified using primers P81 and P82 from pSD6 (*Devanapally et al., 2021*) as a homology repair template with homology arms to the C-terminus of *W09B7.2/F07B7.2*, and pRF4 into N2, and subsequent screening were performed as described above. Genotyping for identical tags *W09B7.2::mCherryΔpi* and *F07B7.2::mCherryΔpi* in isolate AMJ1372 was performed using triplex PCR with primers P79, P83, and P84. Tagging of both loci is evident in *Figure 6—figure supplement 1D*. The *mCherryΔpi* insertion was verified by Sanger sequencing. AMJ1372 hermaphrodites and males generated by heatshock were mated with N2 males and hermaphrodites, respectively, to examine expression in cross progeny and in homozygosed wild-type and *W09B7.2/F07B7.2(-jam137[mCherryΔpi])* animals across generations in six independent F1 lineages from each cross. See *Figure 6C*, *Figure 6—figure supplement 2A* for associated data. YY916 males were crossed with AMJ1372 to generate AMJ1662. The *3xflag::gfp::znfx-1* locus was genotyped using primers P153, P154, and P155.

### To introduce a nonsense mutation into *rde-1* coding sequence

Injection of a crRNA with the target sequence listed as P94 (IDT), tracrRNA, Cas9, a *rde-1(nonsense)* homology repair template (P95) mimicking *rde-1(ne300)* (*Tabara et al., 1999*), and pRF4 into AMJ1372 and subsequent screening were performed as described above. Genotyping for *rde-1(nonsense)* was performed using duplex PCR with primers P96 and P97 and restriction digestion with NlaIII to isolate AMJ1447 and AMJ1448. The nonsense mutation was verified by Sanger sequencing for all isolates.

### To tag *sid-1* with *wrmScarlet* at the 3' end

Injection of a crRNA with the target sequence listed as P52 (IDT), tracrRNA, Cas9, a *sid-1::wrmScarlet13* homology repair template with the beginning (1) and end (3) but not the middle (2) of the coding sequence (*Vicencio et al., 2019*) (P53), and pRF4 into N2 and subsequent screening were performed as described above. Genotyping for *wrmScarlet13* was performed using duplex PCR with primers P54 and P55 to isolate AMJ1280. The *wrmScarlet13* insertion was verified by Sanger sequencing. Subsequent injections were performed into AMJ1280 with a *wrmScarlet13* specific crRNA with the target sequence listed as P56 (IDT), a complete *wrmScarlet* coding sequence amplified from pSEM89 (made in Boulin Lab – gift from Kevin O'Connell) with primers P57 and P58 and the same components as described previously. After similar screening, genotyping for full *wrmScarlet* insertion was performed using duplex PCR with primers P54 and P55 to isolate AMJ1282 and one other independent isolate. The full *wrmScarlet* insertion was verified by Sanger sequencing. AMJ1282 was crossed with AMJ577 males to create AMJ1365.

### To tag *rme-2* with *wrmScarlet* at the 3' end

Injection of a crRNA with the target sequence listed as P67 (IDT), tracrRNA, Cas9, a *rme-2::wrmScarlet13* homology repair template with the beginning (1) and end (3) but not the middle (2) of the coding sequence (*Vicencio et al., 2019*) (P69), and pRF4 into N2 and subsequent screening were performed

as described above. Genotyping for *wrmScarlet13* was performed using duplex PCR with primers P70 and P47 to isolate AMJ1281. The *wrmScarlet13* insertion was verified by Sanger sequencing. Subsequent injections were performed into AMJ1281 with a *wrmScarlet13* specific crRNA with the target sequence listed as P77 (IDT), a complete *wrmScarlet* coding sequence amplified from pSEM89 (made in Boulin Lab – gift from Kevin O'Connell) with primers P57 and P58 and the same components as described previously. After similar screening, genotyping for full *wrmScarlet* insertion was performed using duplex PCR with primers P54 and P55 to isolate AMJ1284 and two other independent isolates. The full *wrmScarlet* insertion was verified by Sanger sequencing.

### To tag *sid-1* internally with *mCherry*

Injection of a crRNA with the target sequence listed as P110 (IDT), tracrRNA, Cas9, an *mCherry* lacking piRNA binding sites amplified from pSD6 (*Devanapally et al., 2021*) as a homology repair template with homology arms to exon 4 of *sid-1* with primers P111 and P112, and pRF4 into N2 and subsequent screening were performed as described above. Genotyping for *mCherryΔpi* was performed using triplex PCR with primers P113, P114, and P79 to isolate AMJ1438 and one other isolate from the same lineage. The *mCherryΔpi* insertion was verified by Sanger sequencing. Subsequent injections were performed into AMJ1438 with a crRNA targeting the 5'-end of *mCherryΔpi* (P115) (IDT), a homology repair template containing a 45-nt linker sequence (P116) and the same components as described previously. After similar screening, genotyping for the linker insertion was performed using duplex PCR with primers P113 and P79 to isolate AMJ1485 and two other independent isolates. Insertion of the linker was verified by Sanger sequencing in all three isolates.

### To introduce a nonsense mutation into *sid-3* coding sequence

Injection of a crRNA with the target sequence listed as P66 (IDT), tracrRNA, Cas9, a *sid-3(nonsense)* homology repair template (P85) mimicking *sid-3(qt31)* (*Jose et al., 2012*) and pRF4 into AMJ1372 and subsequent screening were performed as described above. Genotyping for *sid-3(nonsense)* was performed using duplex PCR with primers P86 and P87, and restriction digestion with StyI to isolate AMJ1449 and AMJ1450. The nonsense mutation was verified by Sanger sequencing for both isolates.

### To introduce a nonsense mutation into *deps-1* coding sequence

Injection of a crRNA with the target sequence listed as P68 (IDT), tracrRNA, Cas9, a *deps-1(nonsense)* homology repair template (P137) mimicking *deps-1(bn124)* (*Spike et al., 2008*) and pRF4 into AMJ1372 and AMJ1412 and subsequent screening were performed as described above. Genotyping for *deps-1(nonsense)* was performed using allele-specific PCR with primers P138 and P139 amplifying the wild-type sequence and primers P140 and P141 amplifying the mutant allele to isolate AMJ1451-AMJ1452 from AMJ1372 and AMJ1574-AMJ1575 from AMJ1412. The nonsense mutation was verified by Sanger sequencing for both isolates.

### To insert the tetracycline K4 aptazyme (*Wurmthaler et al., 2019*) into the 3'UTR of *sid-1*

Injection of a crRNA with the target sequence listed as P52 (IDT), tracrRNA, Cas9, a *sid-1::tetracycline-K4-aptazyme* homology repair template (P78) and pRF4 into N2 and subsequent screening were performed as described above. Genotyping for insertion of the aptazyme sequence was performed using duplex PCR with primers P54 and P55 to isolate AMJ1323. The aptazyme insertion was verified by Sanger sequencing. AMJ1323 hermaphrodites was crossed with AMJ477 (*Ravikumar et al., 2019*) males to create AMJ1330 and with AMJ471 (*Devanapally et al., 2015*) males to create AMJ1350. AMJ1323 males were crossed with JH3197 to create AMJ1355.

### To insert the tetracycline K4 aptazyme (*Wurmthaler et al., 2019*) into the 3'UTR of *gtbp-1(ax2053[gtbp-1::gfp])*

Injection of a crRNA with the target sequence listed as P89 (IDT), tracrRNA, Cas9, a *gtbp-1::gfp::tetracycline-K4-aptazyme* homology repair template (P90) and pRF4 into JH3197 and subsequent screening were performed as described above. Genotyping for insertion of the aptazyme

sequence was performed using duplex PCR with primers P91 and P92 to isolate AMJ1542. The apta-zyme insertion was verified by Sanger sequencing.

### To introduce a missense mutation into *dpy-10* coding sequence

Injection of crRNA with the target sequence listed as P142 (IDT), tracrRNA, Cas9, and a *dpy-10(mis)* homology repair template (P41) mimicking *dpy-10(cn64)* (*Levy et al., 1993*) into AMJ1372 was performed as described above and heterozygous F1 animals were screened for by passaging 'rolling' animals. Animals that appeared wild-type and those that appeared Dpy (homozygous *dpy-10(-)*) were isolated from three independently edited F1 animals. See *Figure 6—figure supplement 2B* for associated data.

### To delete the *W09B7.2/F07B7.2* coding sequence

Injection of crRNAs targeting the start (P143) and end (P144) of the *W09B7.2/F07B7.2* coding sequence (IDT), tracrRNA, Cas9, a *W09B7.2/F07B7.2(deletion)* homology repair template (P145) and pRF4 into N2, and subsequent screening were performed as described above. Genotyping for *W09B7.2/F07B7.2(deletion)* was performed using triplex PCR with primers P146-P148 to isolate AMJ1577, AMJ1612, and AMJ1613. Deletion of both *W09B7.2/F07B7.2* loci was verified by absence of wild-type band by PCR (see *Figure 6—figure supplement 1D*) and Sanger sequencing in all three isolates.

### To delete the *W09B7.2/F07B7.2* coding sequence from *W09B7.2/F07B7.2::mCherryΔpi*

Injection of crRNAs targeting the start (P143) and end (P149) of the *W09B7.2/F07B7.2* coding sequence (IDT), tracrRNA, Cas9, a *W09B7.2/F07B7.2(deletion)* homology repair template (P150) and pRF4 into AMJ1372, and subsequent screening were performed as described above. Genotyping for *W09B7.2/F07B7.2(deletion)* was performed using triplex PCR with primers P148, P151, and P152 to isolate AMJ1615, AMJ1616, and AMJ1617. Deletion of both *W09B7.2/F07B7.2* loci was verified by absence of wild-type band by PCR (see *Figure 6—figure supplement 1D*) and Sanger sequencing in all three isolates. AMJ1615 was outcrossed 1X with N2 males to generate AMJ1770. Genotyping was performed using primers P158-P160 and expression was verified by widefield microscopy. AMJ1615 was crossed with WM49 males to generate AMJ1766 and AMJ1767. Many of the AMJ1770 and AMJ1766 animals had defective germline morphology and therefore only animals with apparently normal morphology were selected for quantification of *sdg-1p::mCherryΔpi* expression. Genotyping for *sdg-1p::mCherryΔpi* was performed using the same primers (P158-P160) and initial expression was verified by widefield microscopy.

## Light-induced damage of neurons

### Optimizing duration of light exposure

20–30 animals expressing PH::miniSOG in neurons (multi copy, AMJ837; single copy, AMJ1019) were placed on an unseeded NGM plate and exposed to blue light (470 nm wavelength) at a distance of approximately 7.5 cm from an LED (Cree Xlamp XP-E2 Color High Power LED Star – Single 1 UP, LED supply) producing light at a power of ~2 mW/mm$^2$ flashing at a frequency of 2 Hz for different durations of time. Animals were then scored for movement defects immediately after light exposure, OP50 was seeded onto the plate, and animals were scored again 24 hr post light exposure (*Figure 2—figure supplement 1A*). Wild-type animals were exposed to blue light for the same durations as control. Representative widefield images of unparalyzed (wild type) and paralyzed (coiled, AMJ837) animals were taken using a Nikon AZ100 microscope and Photometrics Cool SNAP HQ$^2$ camera (*Figure 2—figure supplement 1B, top*). Confocal images of animals expressing PH::miniSOG and DsRed in neurons (AMJ936) with and without 30 min of blue light exposure were taken using a Leica TCS SP8 DLS microscope with HyVolution using a 40X oil objective lens. DsRed was excited using a 638 nm laser and fluorescence was collected through a 598 nm emission filter (*Figure 2—figure supplement 1B, bottom*). Images were adjusted for display using Fiji (*Schindelin et al., 2012*; NIH).

## Silencing by *bli-1*-dsRNA

Five L4 animals with an extrachromosomal array expressing PH::miniSOG and *bli-1*-dsRNA in neurons were placed on seeded NGM plates and allowed to lay progeny for 24 hr. P0 animals were then removed and F1 progeny were exposed to blue light as described above for 60 min at different time points after initial P0 L4 animals were passaged. 96 hr post light exposure F1 progeny with the array were scored for *bli-1* silencing (presence of blisters) in gravid adults (*Figure 2—figure supplement 1C*, *top*, 1D and 1E).

## Silencing by *gfp*-dsRNA

L4 animals with the *mex-5p::mCherry::h2b::gfp::h2b* transgene (*oxSi487*) (*Figure 2B*) were mated with L4 male animals with an extrachromosomal array expressing PH::miniSOG and *gfp*-dsRNA in neurons (*Figure 2A*). After 36 hr of mating and laying progeny, P0 animals were removed from plates and F1 progeny were exposed to blue light as described above for 60 min at different time points after initial P0 L4 animals were mated. 96 hr after mating, F1 cross progeny hermaphrodites with the array were imaged as adults (*Figure 2—figure supplement 1C*, *bottom*) under a coverslip in 10 µl of 3 mM levamisole on a 2% agarose pad using a Nikon AZ100 microscope and Photometrics Cool SNAP HQ$^2$ camera. A C-HGFI Intensilight Hg Illuminator was used to excite GFP (filter cube: 450–490 nm excitation, 495 dichroic, and 500–550 nm emission) and mCherry (filter cube: 530–560 nm excitation, 570 dichroic, and 590–650 nm emission). Animals were scored as bright if fluorescence was easily detectable without adjusting levels, dim if fluorescence could be observed after level was adjusted to saturation, and not detectable if fluorescence was still not observed after level adjustments (*Figure 2—figure supplement 1F*). Representative images were adjusted in Adobe Photoshop to identical levels for presentation (*Figure 2B–D*).

## Sensitive northern blotting

Double-stranded RNA was in vitro transcribed from a PCR amplicon using T7 RNA Polymerase (New England BioLabs; *Figure 3—figure supplement 1D*) or expressed in HT115 *E. coli* after IPTG induction during exponential growth (*Figure 3—figure supplement 1B and C*) and extracted using TRIzol (Thermo Fisher Scientific). RNA was then separated by size using fully denaturing formaldehyde polyacrylamide gel electrophoresis (FDF-PAGE; *Harris et al., 2015*) wherein 10 µg RNA samples were heated with formaldehyde to disrupt dsRNA duplexes and run on a 4% denaturing polyacrylamide gel next to 1 kb and 100 bp DNA ladders for size comparison. After migration, the ladder lanes were stained with ethidium bromide and imaged, and the RNA lanes were transferred to a positively charged nitrocellulose membrane using a Trans-Blot Turbo Transfer System (Bio-Rad) and crosslinked using 120 mJ/cm$^2$ UV radiation. Blots were then exposed to 2.5 pmol of 40-nt HPLC purified DNA oligonucleotides conjugated to digoxigenin (DIG) using the DIG Oligonucleotide Tailing Kit (Roche) hybridized to the nitrocellulose membrane at 60 °C overnight (42 °C for 2 hr for 5S rRNA) in ULTRAhyb buffer (Invitrogen) to probe the sense or antisense strands of *unc-22* (*Figure 3—figure supplement 1B and D*) or *gfp*-dsRNA (*Figure 3—figure supplement 1C*) at different positions (protocol adapted from *Choi et al., 2017*). After hybridization, the membrane was washed and blocked using the DIG Wash and Block Buffer Set (Roche), incubated with Anti-DIG-AP, Fab fragments (Roche) and developed with CSPD (Roche) at 37 °C for 15 min. Chemiluminescence from the AP/CSPD reaction was imaged using a LAS-3000 (Fujifilm) or iBright CL1000 (Invitrogen) imager. Blots were compared to ethidium bromide-stained ladders after imaging to visualize fragment size. Blots were stripped using two washes with 5% SDS (Sigma Aldrich) and two washes with 2X SSC (Sigma Aldrich) and the hybridization, blocking and development procedures were repeated for each probe (5S RNA probe: P118; *unc-22* probes: P119-P124; *gfp* probes: P125-P130).

## Injection of dsRNA

### Injection of synthetic dsRNA

RNA oligonucleotides were purchased from IDT and resuspended in IDT Duplex Buffer (*unc-22*: P131 and P132; *gfp*: P133 and P134; fluorescently labeled *gfp*: P135 and P136). 1 µg each of HPLC purified 50-nt sense and antisense oligonucleotide was diluted to 100–350 ng/µl with IDT Duplex Buffer at a final volume of 10 µl. Alternatively, *unc-22* single-stranded RNA was treated with polynucleotide

kinase and annealed in equal proportion at a final concentration of ~97 ng/µl of *unc-22*-dsRNA in IDT Duplex Buffer (*Figure 3—figure supplement 1E and F*). This mixture was heated to 95 °C for 1 min and cooled at a rate of 1 °C/min to a final temperature of 25 °C. The mix was centrifuged at 16,500 x *g* for 20–30 min and loaded into a microinjection needle. Young adult animals were injected 24 h after the L4 stage in the body cavity just beyond the bend of the posterior gonad arm (*Marré et al., 2016*). Injected animals were recovered with M9 buffer and isolated onto NGM plates and allowed to lay progeny. In cases where animals were mated with N2 males post injection, two adult N2 males were placed on each plate with an injected hermaphrodite.

### Injection of in vitro transcribed dsRNA

Templates for transcription were amplified from RNAi vectors using one common primer specific to the T7 promoter sequence (P117). PCR products were purified using column purification (Macherey-Nagel, ref 740609.50) and subsequently used for transcription by T7 RNA Polymerase (New England BioLabs). Many transcription reactions were pooled and purified using one column to produce concentrated RNA samples. Annealing, centrifugation, and injection into the body cavity of animals staged as L4s (injected between pharynx and anterior intestine) or young adults were performed as described for synthetic dsRNA with identical concentrations unless otherwise indicated in figure legends. In cases where animals were mated with N2 males post injection, two adult N2 males were placed on each plate with an injected hermaphrodite.

### Scoring of gene silencing

For scoring silencing by *unc-22* dsRNA, 10–30 L4 animals were passaged into 10 µl of 3 mM levamisole and scored for twitching, observed as rapid movement of the head and/or tail (as in *Marré et al., 2016*), 3–4 days after injection for progeny of *rme-2(+)* parents and 4–5 days after injection for progeny of *rme-2(-)* parents with no appreciable difference between days in which animals were scored post injection. Weak and strong twitching were scored as in Movies S1-S3 of *Marré et al., 2016*. Numbers of silenced animals and total animals scored were summed across all days of scoring and experimental replicates.

When scoring silencing of *gfp*, animals were either scored by eye in comparison to animals injected with duplex buffer only (i.e. buffer; *Figure 1—figure supplement 1B*) or were mounted in 10 µl of 3 mM levamisole on a 2% agarose pad and imaged under a coverslip as P0 adults (2 days post injection) or F1 L4s (3 days post P0 injection) using a Nikon AZ100 microscope and Photometrics Cool SNAP HQ$^2$ camera. A C-HGFI Intensilight Hg Illuminator was used to excite GFP (filter cube: 450–490 nm excitation, 495 dichroic, and 500–550 nm emission). Representative images for *gfp* expression in F1 animals after P0 injection were adjusted to identical levels in Adobe Photoshop for presentation (*Figure 3B*, *Figure 1—figure supplement 1E*). See "Imaging and quantification of reporters using widefield microscopy" for other methods of scoring *gfp* expression after imaging.

### Imaging of fluorescently labeled dsRNA

Embryos were imaged 22 hr post P0 injection with labeled dsRNA. Laid embryos were picked off plates and placed into 5 µl of 3 mM levamisole on a coverslip for at least 5 min before placing on a 2% agarose pad on a slide. Embryos were imaged using the Eclipse Ti Spinning Disk Confocal (Nikon) with the 60X objective lens. Atto 565 was excited using a 561 nm laser and fluorescence was collected through a 415–475 nm and 580–650 nm emission filter. Images were adjusted for display using Fiji (*Schindelin et al., 2012*; NIH).

### Feeding RNAi

#### P0 and F1 feeding

*E. coli* (HT115) expressing dsRNA was cultured in LB media with 100 µg/µl carbenicillin overnight at 250 rpm. 100 µl of cultured bacteria was then seeded onto RNAi plates and incubated at room temperature for approximately 24 hr. L4 animals were passaged onto seeded RNAi plates and progeny were scored for silencing by bacteria expressing dsRNA targeting *unc-22* (twitching in levamisole), *bli-1* (blisters), *pos-1* (dead eggs) or expressing L4440 as an empty vector control.

### P0 only feeding

RNAi bacteria were cultured and seeded as described above. L4-stage or young adult-stage (24 hr post L4) animals were passaged onto seeded RNAi plates and cultured at 20 °C for approximately 24 hr. In some cases, fed P0 animals were then scored for silencing as described above and subsequently imaged under widefield microscopy (*Figure 7F*). To score unfed progeny, fed P0 animals were picked into 1 ml of M9 buffer and washed four times to remove any residual bacteria (as in *Marré et al., 2016*). After washing, animals were resuspended in 200 µl of remaining M9 buffer and placed onto a seeded NGM plate. 1 hr later, animals were isolated onto single NGM plates and their progeny were scored for silencing as described above.

### Limited P0 only feeding

RNAi bacteria were cultured and seeded as described above. L4-stage animals were passaged onto seeded RNAi plates and cultured at 20 °C for approximately 16 hr. Animals were then passaged onto NGM plates seeded with *E. coli* (OP50) and cultured for 1.5 hr at room temperature. Animals were then again passaged to new OP50 seeded plates (1 animal on each plate) and progeny (only L3 larvae, L4 larvae and adults) were counted after 4 days of being cultured at 20 °C (~96 hr after moving to new OP50 plates).

### F1 only feeding

L4-staged animals were passaged onto RNAi plates seeded with 10 µl of *E. coli* (OP50). Animals were allowed to develop into adults and lay eggs over 24 hr at 20 °C and then removed from plates. Plates with eggs were then seeded with RNAi bacteria cultured and seeded as described above and further cultured at 20 °C. Hatched progeny were imaged throughout development or as adults 3 days after being staged as L4 animals (day 3 adults).

## Tetracycline-induced expression

### For animals cultured with OP50 *E. coli*

81.6 µl of a 500 µM solution of tetracycline in water was added to 4 mL NGM plates previously seeded with OP50 *E. coli* (at least two days earlier) to create plates with ~10 µM tetracycline (concentration based on *Wurmthaler et al., 2019*). Volumes of 166.7 µl and 444.4 µl of tetracycline solution were used to create plates with final concentrations of ~20 µM or ~50 µM, respectively (see *Figure 4— figure supplement 2D*). Control plates were also made by adding the same amount of water to seeded NGM plates without tetracycline. Tetracycline plates and control plates were incubated at room temperature out of direct light overnight to allow any remaining liquid to dry. Animals were passaged to tetracycline or water plates with or without previous injection of 10 µM tetracycline or water into adult gonads. Progeny expressing neuronal *unc-22* or *gfp*-dsRNA were scored for silencing on the first day of adulthood. In the case of silencing of *gtbp-1::gfp* by neuronal *gfp*-dsRNA, animals with the array expressing *gfp*-dsRNA were passaged as L4s onto new tetracycline or water plates to be imaged as day 1 adults. The brood size of animals cultured on OP50 with 10 µM tetracycline or water was scored by staging single L4 animals on NGM plates with 10 µM tetracycline or water and moving animals every 24 hr to new 10 µM tetracycline or water plates. Progeny laid on each of the 4 days were counted after growing to adulthood, continuously cultured under either condition.

### For animals cultured on HT115 *E. coli* expressing dsRNA

Bacteria expressing *bli-1*-dsRNA, *gfp*-dsRNA, *pos-1*-dsRNA or L4440 control vector were cultured overnight to a maximum time of 24 hr (for *gfp*-dsRNA and L4440 only) and 100 µl of bacteria was seeded onto RNAi plates. Plates were incubated for 1–2 days at room temperature to allow for growth and drying of bacteria. 10 µM tetracycline or water was added to newly seeded plates as described above. After drying of tetracycline and water, P0 animals were added to plates and F1 animals were scored for silencing by *bli-1*-dsRNA or *gfp*-dsRNA as adults in the next generation. Silencing by *pos-1*-dsRNA was scored by measuring the brood of three L4 animals staged on a single RNAi plate with *pos-1*-dsRNA and 10 µM tetracycline or water. Brood size over four days was measured after moving all P0 animals every 24 hr to new 10 µM tetracycline or water plates and scoring adult progeny cultured under either condition.

In all experiments, animals expressing *unc-22*-dsRNA in neurons were exposed to the same tetracycline and water solutions used and scored for *unc-22* silencing as adults as a control for effectiveness of tetracycline (see summary of data in *Figure 4—figure supplement 2B*).

## Imaging and quantification of reporters using widefield microscopy

All animals and embryos expressing fluorescent reporters were imaged in 10 µl of 3 mM levamisole on a 2% agarose pad using a Nikon AZ100 microscope and Photometrics Cool SNAP HQ$^2$ camera. A C-HGFI Intensilight Hg Illuminator was used to excite mCherry (filter cube: 530–560 nm excitation, 570 nm dichroic, and 590–650 nm emission), GFP or other autofluorescent molecules in the green channel (filter cube: 450–490 nm excitation, 495 nm dichroic, and 500–550 nm emission) and autofluorescent molecules in the blue channel (filter cube: 325–375 nm excitation, 400 nm dichroic, 435–485 nm emission). Intensity of GFP and mCherry were quantified in Fiji (*Schindelin et al., 2012*; NIH) using the methods described below. Representative images were adjusted in Fiji (*Schindelin et al., 2012*; NIH) and/or Adobe Photoshop to identical levels for presentation (*Figure 2B-D, 3B, 4B and C, 6A and 7E*, *Figure 1-figure supplement 1E* and *Figure 4-figure supplement 2G*).

### For GTBP-1::GFP quantification post dsRNA injection

Somatic *gfp* expression was quantified between the pharynx and anterior gonad arm by drawing a circle or ventral to dorsal line within the boundaries of the animal (*Figure 3—figure supplement 2A*) on a brightfield image, creating a mask, imposing that mask onto the GFP channel image and measuring average intensity or intensity along the line, respectively. To measure background fluorescence, the same circle or a new circle was used to measure average intensity outside of the animal. Germline GFP expression was quantified by freely selecting part of the distal or proximal region of the anterior or posterior gonad arm (*Figure 3—figure supplement 2A*) excluding the intestine to avoid intestinal autofluorescence. Selection was performed using a brightfield image, a mask was created and imposed onto the GFP channel image and average intensity was measured. To measure background fluorescence, the same selection boundary was moved outside of the animal and average background intensity was measured. To plot average GFP intensity measured by a circle or free selection, average background intensity was subtracted from GFP intensity for each image and plotted with a box plot (*Figure 3—figure supplement 2C*). To plot GFP intensity along the ventral to dorsal axis in the anterior soma, the average intensity in each tenth of the axis was calculated for each animal and plotted with a shaded region representing 95% confidence intervals (*Figure 3—figure supplement 2B*, *top*). To calculate differences in intensity between the interior and exterior of animals, the average intensity of the 0.4–0.6 region of the axis was divided by the average intensity of the 0.1 and 0.9 points of the axis. These values were calculated and shown for each animal as a box plot (*Figure 3—figure supplement 2B*, *bottom*). All plotting was done using custom R scripts.

### For GTBP-1::GFP quantification after exposure to dsRNA via feeding or neuronal expression

Animals fed L4440 or *gfp*-dsRNA for different durations of the P0 and/or F1 generation were scored for silencing in the germline and soma at different stages during the F1 generation (*Figure 1*, *Figure 1—figure supplement 1A*). Somatic GFP intensity (a.u.) was quantified in the tail region by drawing a ventral to dorsal line within the boundaries of the animal (*Figure 4—figure supplement 1C and E*) on a brightfield image, creating a mask, imposing that mask onto the GFP channel image and measuring average intensity or intensity along the line. To measure background fluorescence, a circle was used to measure average intensity outside of the animal. Germline GFP intensity (a.u.) was measured by free selection of germ cells but avoiding intestinal cells at each stage, selecting a region around the primordial vulva in L2 animals, in one of two extending gonad arms in L3 and L4 animals, in the proximal or distal gonad in young adults, and of eggs in utero in gravid adults. To measure background fluorescence, the same selection or a new selection was used to measure average intensity outside of the animal. To plot average GFP intensity measured by free selection, average background intensity was subtracted from GFP intensity for each image and shown as a box plot (*Figure 1*, *Figure 1—figure supplement 1A* and *Figure 4—figure supplement 1C and E*). All plotting was done using custom R scripts.

## Adjustment of fluorescence images of *sid-1::mCherryΔpi* animals for comparison to images of wild-type animals

Representative images of *sid-1(jam195[linker::mCherryΔpi])* and wild type animals at different stages were adjusted to the same maximum and minimum displayed values of intensity using Fiji (*Schindelin et al., 2012*; NIH) to highlight each region of interest below saturation (*Figure 4B and C*).

## For quantification of SDG-1::mCherry and mCherry expressed under the *sdg-1* promoter

Germline mCherry intensity was quantified by freely selecting part of the distal (for *Figure 7F and G*) or proximal region of the anterior or posterior gonad arm excluding the intestine to avoid quantifying intestinal autofluorescence. Selection was performed using a brightfield image, a mask was created and imposed onto the mCherry channel image and average intensity was measured. To measure background fluorescence, the same selection boundary was moved outside of the measured gonad arm and average background intensity was measured. To plot average mCherry intensity, average background intensity was subtracted from mCherry intensity for each gonad arm and shown as a box plot using custom R scripts (*Figures 6B-G and 7F and G*, *Figure 6—figure supplement 2*). In *Figure 6C*, SDG-1::mCherry intensity measurements, adjusted by subtracting background intensity and intensity measurements made in a wild-type animal lacking mCherry, were normalized to RT-qPCR measurements by multiplying each median intensity value by a conversion factor. This conversion factor was calculated by dividing the median SDG-1::mCherry intensity in AMJ1372 animals by the median relative *sdg-1* mRNA level in AMJ1372 RNA samples. All estimated relative *sdg-1* expression values were then normalized to those of wild-type animals by dividing all values by the wild-type value.

## Imaging and quantification of reporters using confocal microscopy

### For the endogenous gene tag *sid-1::mCherryΔpi*

SID-1::mCherry fluorescence from an L1-staged animal was imaged using LSM 980 Airyscan 2 Laser Scanning Confocal (Zeiss) with a 63X oil objective lens after paralyzing the worm as above. mCherry was excited using a 561 nm laser and fluorescence was collected through a 422–477 nm and 573–627 nm emission filter. For *Figure 4D*, after removing noise using a 3D gaussian blur with 2.0 sigma in X, Y, and Z, depth-coded maximum intensity projections of Z-stacks were stitched together for display as described earlier (*Ravikumar et al., 2019*).

### For the endogenous gene tag *W09B7.2/F07B7.2::mCherryΔpi*

Adult animals were placed in 10 µl of 3 mM levamisole and imaged using the Eclipse Ti Spinning Disk Confocal (Nikon) with a 60X objective lens or the LSM 980 Airyscan 2 Laser Scanning Confocal (Zeiss) with a 63X oil objective lens. GFP was excited using a 488 nm laser and fluorescence was collected through a 499–557 nm and 659–735 nm emission filter, and mCherry fluorescence was excited and collected as described above. Images and videos were adjusted in Fiji (*Schindelin et al., 2012*; NIH) and Adobe Photoshop to identical levels for presentation (*Figure 7C and D* and *Videos 1–4*).

## RNA sequencing, principal component analysis, and differential expression analysis

### For analysis of previously generated *sid-1(-)* alleles

Mixed-stage animals were washed from 10 plates in biological duplicate 5 days after passaging L4-staged animals. Total RNA was extracted from pellets using TRIzol (Fisher Scientific). PolyA+ RNAs were purified and converted to DNA libraries by the University of Maryland Genomics Core using the Illumina TruSeq Library Preparation Kit. FASTQ files were processed (*Martin, 2011*) using the command "`cutadapt -j 0 a AGATCGGAAGAGCACACGTCTGAACTCCAGTCA -m 20 -q 20 -o cutread.gz fasta1.gz`". Reads were assigned transcript IDs and counted (*Patro et al., 2017*) using the command "`salmon quant -i celegans.index -l A -r cutread.gz -p 8 -validateMappings -o quant_file`". For conversion of transcript IDs to gene IDs, a table of matching transcript and gene IDs was generated from a GTF file using the command "`grep "^[^#]' Caenorhabditis_elegans.Wbcel235.101.gtf | awk '{if($3 = "transcript") {print}}' | awk '{print $12,$14}' | tr -d '";'>transcript_id_gene_id.tsv`".

Conversion was then made using this table with tximport (*Soneson et al., 2015*) in R, whereafter only genes with more than 0.1 counts per million for at least two samples were used in subsequent analyses with pairs of sample types *sid-1(qt9[nonsense])* vs. wild type and *sid-1(tm2700[deletion]); tmIs1005sid-1(+)* vs. *sid-1(tm2700[deletion])*. After normalizing samples using the trimmed mean of M-values method (*Robinson and Oshlack, 2010*), principal component analysis was performed in R by comparing samples based on the 500 genes with the largest standard deviations in their $\log_2$-fold change between each set of samples (see *Technical comments*). Differential expression analysis was performed using limma(voom) (*Law et al., 2014*) in R (example available at https://github.com/AntonyJose-Lab/Shugarts_et_al_2023; *AntonyJose-Lab, 2023*). Volcano plots of differential expression for all genes compared were plotted using custom R scripts with genes having an adjusted *p*-value threshold (*q*-value) less than 0.05 in black and those greater than 0.05 in grey (see *Technical comments*).

### For analysis of newly generated *sid-1(-)* alleles

Total RNA >200 nt was extracted using RNAzol (Sigma-Aldrich) from 200 µl pellets of mixed-stage animals collected from 6 non-starved but crowded plates in biological triplicate for each strain. PolyA+ RNAs were purified and converted to DNA libraries using the Illumina TruSeq Stranded mRNA Library Preparation Kit. Library quality was assayed using TapeStation (Agilent) and libraries were sequenced using a HiSeq X10 (Illumina) by Omega Bioservices. FASTQ files were processed (*Martin, 2011*) using the command '`cutadapt -j 0 a AGATCGGAAGAGCACACGTCTGAACTCCAGTCA -A AGATCGGA AGAGCGTCGTGTAGGGAAAGAGTGT -m 20 -q 20 -o cutread1.gz -p cutread2.gz read1.gz read2.gz`'. Reads were assigned transcript IDs and counted (*Patro et al., 2017*) using the command '`salmon quant -i celegans.index -l A -1 cutread1.gz -2 cutread2.gz -p 8 - validateMappings -o quant_files`'. For conversion of transcript IDs to gene IDs, a table of matching transcript and gene IDs was generated as described above. Conversion was then made using this table with tximport (*Soneson et al., 2015*) in R, whereafter only genes with more than 0.1 counts per million for at least three samples were used in subsequent analyses. Normalization, principal component analysis (*Figure 5C*, *Figure 5—figure supplement 1A*) and differential expression analysis were performed as described above. Volcano plots of differential expression were plotted as described above (*Figure 5D*, *Figure 5—figure supplement 1B*). Genes that were similarly misregulated in *Figure 5D*, *Figure 5—figure supplement 1B* are in red.

### For analysis of data from *Reed et al., 2020*

FASTQ files were processed (*Martin, 2011*) using the command '`cutadapt -j 0 a AGATCGGA AGAGCACACGTCTGAACTCCAGTCA -m 20 -q 20 -o cutread.gz fasta1.gz`'. Reads were assigned transcript IDs and counted (*Patro et al., 2017*) using the command '`salmon quant -i celegans.index -l A -r cutread.gz -p 8 -validateMappings -o quant_file`'. For conversion of transcript IDs to gene IDs, a table of matching transcript and gene IDs was generated as described above. Conversion was then made using this table with tximport (*Soneson et al., 2015*) in R. Normalization and differential expression analysis were performed as described above. Volcano plots of differential expression were plotted as described above with *sid-1*, *sdg-1* (*W09B7.2/F07B7.2*) and *sdg-2* (*Y102A5C.36*) in red and all other genes in grey (*Figure 5—figure supplement 1D*).

## Genome mapping and visualization of sequencing reads for *sid-1-dependent genes*

After RNA sequencing samples were processed as described above, reads were mapped to the *C. elegans* genome (*Kim et al., 2019*) using the command '`hisat2 -p 8 x Celegans98index -1 cutread1.gz -2 cutread2.gz -S sam1`'. The SAM file outputs were then converted to BAM files (*Li, 2009*) using the command '`samtools view -b sam1 | samtools sort ->bam1.bam`' and BAM index files were created for visualization using '`samtools index bam1.bam`'. Reads for the *sid-1* and *F14F9.5* locus, *W09B7.2/F07B7.2* locus, and *Y102A5C.36* locus were plotted using custom R scripts and axes were normalized for each sample based on its total mapped reads, calculated using the command '`samtools view -c -F 4 bam1.bam`' (*Figure 5—figure supplement 1C*).

## Comparisons with published datasets

Datasets in 21 published studies were collected and compared based on the gene names to identify changes in *sid-1*, *sdg-1*, *sdg-2* and *tbb-2* (control), if reported (*Figure 5F*). After standardizing the names across all datasets, the fold-changes reported, if available, were used to plot a heatmap. Cases where fold-changes were not available were set conservatively as $\log_2$(fold change)=2. The R script used is available at https://github.com/AntonyJose-Lab/Shugarts_et_al_2023 (*AntonyJose-Lab, 2023*).

## Reverse transcription and quantitative PCR

Total RNA was extracted using TRIzol (Fisher Scientific) from 200 µl pellets of mixed-stage animals collected from 3 to 6 non-starved but crowded plates in biological triplicate for each strain. The aqueous phase was then washed with an equal amount of chloroform and precipitated overnight on ice with 100 µl of 3 M sodium acetate, 1 ml of 100% ethanol and 10 µg glycogen (Invitrogen). RNA pellets were washed twice with 70% ethanol and resuspended in 22 µl nuclease-free water. RNA samples were then Dnase-treated in Dnase buffer (10 mM Tris-HCl pH 8.5, 2.5 mM $MgCl_2$, 0.5 mM $CaCl_2$) with 0.5 U Dnase I (New England BioLabs) at 37 °C for 60 min followed by heat inactivation at 75 °C for 10 min. RNA concentration was measured and 1 µg of total RNA was used as input for reverse transcription using 50 U SuperScript III Reverse Transcriptase (Invitrogen) (+RT) or no reverse transcriptase as a negative control (-RT) (RT primers: *tbb-2* (P98), *sid-1* (P101), *W09B7.2/F07B7.2* (P104), *Y102A5C.36* (P107)). For qPCR, each +RT biological replicate was assayed in technical triplicate for each gene target, along with a single -RT sample for each corresponding biological replicate using 2 µl cDNA and a no template control (NTC) with the LightCycler 480 SYBR Green I Master kit (Roche). Ct values were measured with the Bio-Rad C1000 CFX96 Real-Time System and Bio-Rad CFX Software (qPCR primers: *tbb-2* (P99 and P100), *sid-1* (P102 and P103), *W09B7.2/F07B7.2* (P105 and P106), *Y102A5C.36* (P108 and P109)). To calculate relative change in mRNA abundance compared to wild type, we calculated $\log_2(2^{(-(\text{gene Ct} - \text{tbb-2 Ct}))})$ using the median of technical replicates for the biological triplicates of each genotype. Ct values were only used if they were lower than corresponding -RT and NTC Ct values. The median value of wild-type biological replicates was then subtracted from the value for each sample to plot calculated values with respect to wild-type levels (*Figure 5E*, *Figure 5—figure supplement 1E*, *Figure 6—figure supplements 1E and 2A*).

## BLAST searches and protein alignment

BLAST (NCBI) searches were performed using the W09B7.2/F07B7.2 (SDG-1) amino acid sequence with default parameters and any homologs identified were aligned to SDG-1 using Clustal Omega (*Madeira et al., 2019*) with default parameters. Alignments produced are shown in *Figure 6—figure supplement 1B* with residues shared by two proteins (grey highlight) or all three proteins (black highlight) indicated.

## Annotation of the Cer9 retrotransposon containing W09B7.2/F07B7.2

The *Cer9* retrotransposon containing *W09B7.2/F07B7.2* (*sdg-1*) was annotated using sequence features from UCSC Genome Browser and amino acid sequences obtained from *Bowen and McDonald, 1999*. The 5' and 3' LTR sequences were identified using RepeatMasker and were confirmed to have TC and GA dinucleotides at the beginning and end of each sequence, respectively (*Bowen and McDonald, 1999*). Amino acid sequences from *Bowen and McDonald, 1999* corresponding to *gag* and *pol* (PR: protease, RT: reverse transcriptase, RH: RNaseH, IN: integrase) elements of *Cer9* were used in tblastn (NCBI) searches to determine their positions in the *Cer9* retrotransposon sequence that also contains *sdg-1*.

## Mating-induced silencing

Mating-induced silencing was assayed by crossing males with the transgene labeled *T* (*oxSi487*) encoding *mex-5p::mCherry::h2b::gfp::h2b* to hermaphrodites lacking the transgene, both in otherwise wild-type backgrounds or indicated mutant backgrounds. Reciprocal control crosses were performed in parallel where hermaphrodites with *T* were crossed to males lacking *T*. Animals were imaged and scored as described for this transgene in the 'Light-induced damage of neurons' section.

## Technical comments

### Making a *sid-1* translational reporter

Previous attempts at observing SID-1 localization relied on multi-copy transgenes (*Winston et al., 2002*), which can become silenced within the germline (*Kelly et al., 1997*) and could produce a variety of tagged and untagged proteins (*Le et al., 2016*). When using multi-copy transgenes to express a SID-1 fusion protein tagged at the C-terminus with DsRed or GFP (*Figure 4—figure supplement 1A*) under the control of a promoter that drives expression within body-wall muscles, we observed intracellular localization of SID-1::DsRed or SID-1::GFP (*Figure 4—figure supplement 1B*) along with rescue of gene silencing by ingested dsRNA in body-wall muscles by both arrays (for SID-1::DsRed – silencing in wild type = 100% (n=10), *sid-1(qt9)*=0% (n=11), *sid-1(qt9); jamIs2[myo-3p::sid-1(+)::DsRed]*=100% (n=6); for SID-1::GFP – silencing in wild type = 100% (n=50), *sid-1(qt9); jamEx193[myo-3p::sid-1(+)::gfp]*=100% [n=60]). However, similar tagging to express SID-1 fusion proteins from either a single-copy transgene expressed in the germline (SID-1::DsRed) or the endogenous locus (SID-1::wrmScarlet) did not enable gene silencing by ingested dsRNA (for evaluating function of *mex-5p::sid-1(+)::DsRed::sid-1 3' UTR (jamSi12)*: silencing of *pos-1* in wild-type=100% (n=7), *sid-1(qt9)*=0% (n=7), *jamSi12; sid-1(qt9)*=0% (n=15); for evaluating function of *sid-1::wrmScarlet(jam117)*: silencing of *pos-1* in wild-type=100% (n=8), *sid-1(jam80)*=0% (n=8), *sid-1(jam117)*=0% [n=8]), suggesting that the C-terminal fusions of SID-1 were likely non-functional and that apparent function when using multi-copy transgenes reflects production of untagged variants. In support of our rationale, a recent prediction of SID-1 structure (*Jumper et al., 2021*; *Varadi et al., 2022*) suggests that the C-terminus is sequestered (*Figure 4—figure supplement 1C*), a feature that may be disrupted by the addition of C-terminal fluorophores, potentially leading to misfolded proteins that are degraded. Consistently, we found that internal tagging of the *sid-1* gene using Cas9-mediated genome editing to express SID-1::mCherry (*Figure 4*) resulted in a fusion protein with detectable function (percent *unc-22* silencing - wild type = 100% (n=59), *sid-1(jam195[sid-1::mCherryΔpi])* = ~98% (n=52); percent *bli-1* silencing – wild type = ~87% (n=833), *sid-1(jam195[sid-1::mCherryΔpi])* = ~0.01% [n=634]).

### RNA sequencing analysis of existing *sid-1* mutants

We initially analyzed polyA+ RNAs extracted from wild-type animals, two available *sid-1* loss-of-function mutants (*Winston et al., 2002*; *Kage-Nakadai et al., 2014*) (*sid-1(-)*) and one available rescue strain where *sid-1(-)* was rescued with a transgene that overexpresses *sid-1(+)* (*Kage-Nakadai et al., 2014*), but found that pairwise comparisons between wild-type and mutant samples with otherwise similar genetic backgrounds did not yield any significantly misregulated genes present in both comparisons (*Figure 4—figure supplement 1E*). Strains with similar genotypes (*sid-1(+)* or *sid-1(-)*) did not cluster together when using principal component analysis (*Figure 4—figure supplement 1D*), suggesting that other differences (e.g. genetic background) obscure or misrepresent differences between *sid-1(+)* and *sid-1(-)* animals.

## Rationale for inferences

### Prior models and assumptions

All dsRNA is trafficked similarly. Entry of dsRNA into the germline can initiate transgenerational RNA silencing of some but not all genes. No SID-1-dependent germline genes are known, suggesting that SID-1 could be used solely in response to viral infection by analogy with roles of other members of RNA interference pathways.

## Evidence supporting key conclusions

Temporal selectivity of dsRNA transport was probed using three approaches for delivery of dsRNA (damage-induced release from neurons, ingestion, and injection). Spatial selectivity of dsRNA import and/or subsequent silencing was inferred based on differences in the frequency of patterns of silencing within the germline. Substrate selectivity of dsRNA transport pathways was probed using genetic mutants and dsRNA of different lengths and 5' chemistry. Diversity of dsRNAs made in bacteria and upon in vitro transcription was visualized using Northern blotting. Analysis of *sid-1* mutants generated from the same wild-type cohort and a revertant was used for better control of genetic background, aiding in the identification of **s**id-1-**d**ependent **g**enes (*sdg*). Separate measurement of *sdg-1*

expression in descendants of independently edited isolates, along different lineages after perturbations, and in different gonads within single animals demonstrated stochasticity in gene expression and revealed establishment of different heritable epigenetic states. Co-localization of SDG-1::mCherry in perinuclear foci with the Z-granule marker GFP::ZNFX-1, its reported association with the Z-granule component ZSP-1/PID-2 and DEPS-1, changes in its levels in response to loss of SID-1 or the dsRNA-binding protein RDE-4 and its dynamic nuclear localization similar to CSR-1b was used to propose that SDG-1 plays a role in small RNA regulation while also being modulated by the activity of SID-1 and RDE-4.

## Acknowledgements

We thank Sindhuja Devanapally for data on silencing by neuronal dsRNA in *hrde-1(-)* animals and mating-induced silencing in *sid-1(-)* animals; Daphne Knudsen for the generation of *mut-16(jam148[nonsense])*; Mary Chey, Samiha Tasnim, and Daphne Knudsen for comments on the manuscript; the *Caenorhabditis elegans* Genetic Stock Center, the Seydoux laboratory (Johns Hopkins University) and the Hunter laboratory (Harvard University) for strains; Quentin Gaudry for help in creating our optogenetics apparatus; the Andrews laboratory for use of their Nikon Eclipse Ti spinning disk confocal microscope; Amy Beaven and the Imaging Core Facility for temporary use of a Leica TCS SP8 DLS microscope with HyVolution and the Zeiss LSM 980 AiryScan 2 Laser Scanning Confocal microscope (supported by grant 1S10OD025223-01A1 from the NIH); Lanelle Edwards, Rex Ledesma, Carlos Machado, and Omega Bioservices for help with RNA sequencing and analysis. This work was supported by UMD CMNS Dean's Matching Award for "Training Program in Cell and Molecular Biology" T32GM080201 to NMSD. and in part by National Institutes of Health Grants R01GM111457 and R01GM124356, and National Science Foundation Grant 2120895 to AMJ.

## Additional information

### Funding

| Funder | Grant reference number | Author |
|---|---|---|
| National Institute of General Medical Sciences | R01GM111457 | Antony M Jose |
| National Institute of General Medical Sciences | R01GM124356 | Antony M Jose |
| University of Maryland | UMD CMNS Dean's Matching Award for "Training Program in Cell and Molecular Biology" T32GM080201 | Nathan M Shugarts Devanapally |
| National Science Foundation | 2120895 | Antony M Jose |

The funders had no role in study design, data collection and interpretation, or the decision to submit the work for publication.

### Author contributions

Nathan M Shugarts Devanapally, Conceptualization, Data curation, Formal analysis, Investigation, Visualization, Methodology, Writing – original draft, Writing – review and editing; Aishwarya Sathya, Conceptualization, Data curation, Formal analysis, Validation, Investigation, Visualization, Methodology, Writing – original draft, Writing – review and editing; Andrew L Yi, Winnie M Chan, Data curation, Formal analysis, Investigation, Methodology, Writing – review and editing; Julia A Marre, Formal analysis, Investigation, Methodology; Antony M Jose, Conceptualization, Data curation, Formal analysis, Supervision, Funding acquisition, Investigation, Visualization, Methodology, Writing – original draft, Project administration, Writing – review and editing

### Author ORCIDs

Nathan M Shugarts Devanapally https://orcid.org/0000-0003-4140-7010

Aishwarya Sathya 🆔 https://orcid.org/0000-0001-6128-0040
Antony M Jose 🆔 https://orcid.org/0000-0003-1405-0618

Reviewer #1 (Public review): https://doi.org/10.7554/eLife.99149.3.sa1
Reviewer #2 (Public review): https://doi.org/10.7554/eLife.99149.3.sa2
Author response https://doi.org/10.7554/eLife.99149.3.sa3

## Additional files

### Supplementary files
MDAR checklist

### Data availability
All data are available on figshare (https://doi.org/10.6084/m9.figshare.25036142.v1). RNA-seq data has been deposited to Gene Expression Omnibus (GEO) with the accession number GSE185385. All code is available within the manuscript or at https://github.com/AntonyJose-Lab/Shugarts_et_al_2023 on GitHub (*AntonyJose-Lab, 2023*).

The following datasets were generated:

| Author(s) | Year | Dataset title | Dataset URL | Database and Identifier |
| --- | --- | --- | --- | --- |
| Shugarts N, Yi A, Chan W, Marré JA, Sathya A, Jose AM | 2025 | Intergenerational transport of double-stranded RNA in *C. elegans* can limit heritable epigenetic changes | https://www.ncbi.nlm.nih.gov/geo/query/acc.cgi?acc=GSE185385 | NCBI Gene Expression Omnibus, GSE185385 |
| Shugarts N, Sathya A, Yi A, Chan W, Marre J, Jose A | 2024 | Intergenerational transport of double-stranded RNA in *C. elegans* can limit heritable epigenetic changes | https://doi.org/10.6084/m9.figshare.25036142.v1 | figshare, 10.6084/m9.figshare.25036142.v1 |

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
