## [Editor Report · eLife Assessment]

In this report, the authors present **valuable** findings identifying a novel worm-specific protein (sdg-1) that is induced upon loss of dsRNA import via SID-1, but is not required to mediate SID-1 RNA regulatory effects. The genetic and genomic approaches are well-executed and the revision contains generally **solid** support for the central findings of the work. These findings will be of interest to those working in the germline epigenetic inheritance field.

---

## [Referee Report · Reviewer #1 (Public review)]

Summary:

In the manuscript "Intergenerational transport of double-stranded RNA limits heritable epigenetic changes," Shugarts and colleagues investigate intergenerational dsRNA transport in the nematode *C. elegans*. By inducing oxidative damage, they block dsRNA import into cells, which affects heritable gene regulation in the adult germline (Fig. 2). They identify a novel gene, sid-1-dependent gene-1 (sdg-1), upregulated upon SID-1 inhibition (Fig. 3). Both transient and genetic depletion of SID-1 lead to the upregulation of sdg-1 and a second gene, sdg-2 (Fig. 5). Interestingly, while sdg-1 expression suggests a potential role in dsRNA transport, neither its overexpression nor loss-of-function impacts dsRNA-mediated silencing in the germline (Fig. 7).

Strengths:

• The authors employ a robust neuronal stress model to systematically explore SID-1 dependent intergenerational dsRNA transport in *C. elegans*.

• They discover two novel SID-1-dependent genes, sdg-1 and sdg-2.

• The manuscript is well-written and addresses the compelling topic of dsRNA signaling in *C. elegans*.

Weaknesses:

• The molecular mechanism downstream of SDG-1 remains unclear. Testing whether sdg-2 functions redundantly with sdg-1could provide further insights.

• SDG-1 dependent genes in other nematodes remain unknown.

---

## [Referee Report · Reviewer #2 (Public review)]

Summary:

RNAs can function across cell borders and animal generations as sources of epigenetic information for development and immunity. The specific mechanistic pathways how RNA travels between cells and progeny remains an open question. Here, Shugarts, et al. use molecular genetics, imaging, and genomics methods to dissect specific RNA transport and regulatory pathways in the *C. elegans* model system. Larvae ingesting double-stranded RNA is noted to not cause continuous gene silencing throughout adulthood. Damage of neuronal cells expressing double-stranded target RNA is observed to repress target gene expression in the germline. Exogenous short or long double-stranded RNA required different genes for entry into progeny. It was observed that the SID-1 double-stranded RNA transporter showed different expression over animal development. Removal of the sid-1 gene caused upregulation of two genes, the newly described sid-1-dependent gene sdg-1 and sdg-2. Both genes were observed to be negatively regulated by other small RNA regulatory pathways. Strikingly, loss then gain of sid-1 through breeding still caused variability of sdg-1 expression for many, many generations. SDG-2 protein co-localizes with germ granules, intracellular sites for heritable RNA silencing machinery. Collectively, sdg-1 presents a model to study how extracellular RNAs can buffer gene expression in germ cells and other tissues.

Strengths:

(1) Very cleaver molecular genetic methods and genomic analyses, paired with thorough genetics, were employed to discover insights into RNA transport, sdg-1 and sdg-2 as sid-1-dependent genes, and sdg-1's molecular phenotype.

(2) The manuscript is well cited, and figures reasonably designed.

(3) The discovery of the sdg genes being responsive to the extracellular RNA cell import machinery provides a model to study how exogenous somatic RNA is used to regulate gene expression in progeny. The discovery of genes within retrotransposons stimulates tantalizing models how regulatory loops may actually permit the genetic survival of harmful elements.

Weaknesses:

(1) The manuscript is broad, making it challenging to read and consider the data presented. Of note, since the original submission, the authors have improved the clarity of the writing and presentation.

Comments on revised version:

This reviewer thanks the authors for their efforts in revising the manuscript. In their rebuttal, the authors acknowledged the broad scope of their manuscript. I concur. While I still think the manuscript is a challenge to read due to its expansive nature, the current draft is substantially improved when compared to the previous one. This work will contribute to our general knowledge of RNA biology, small RNA regulatory pathways, and RNA inheritance.

---

## [Author Response]

The following is the authors’ response to the current reviews.

**Reviewer #1 (Public review):**
Summary:In the manuscript "Intergenerational transport of double-stranded RNA limits heritable epigenetic changes," Shugarts and colleagues investigate intergenerational dsRNA transport in the nematode *C. elegans*. By inducing oxidative damage, they block dsRNA import into cells, which affects heritable gene regulation in the adult germline (Fig. 2). They identify a novel gene, sid-1-dependent gene-1 (sdg-1), upregulated upon SID-1 inhibition (Fig. 3). Both transient and genetic depletion of SID-1 lead to the upregulation of sdg-1 and a second gene, sdg-2 (Fig. 5). Interestingly, while sdg-1 expression suggests a potential role in dsRNA transport, neither its overexpression nor loss-of-function impacts dsRNA-mediated silencing in the germline (Fig. 7).Strengths:• The authors employ a robust neuronal stress model to systematically explore SID-1 dependent intergenerational dsRNA transport in *C. elegans*.• They discover two novel SID-1-dependent genes, sdg-1 and sdg-2.• The manuscript is well-written and addresses the compelling topic of dsRNA signaling in *C. elegans*.Weaknesses:• The molecular mechanism downstream of SDG-1 remains unclear. Testing whether sdg-2 functions redundantly with sdg-1could provide further insights.• SDG-1 dependent genes in other nematodes remain unknown.

We thank the reviewer for highlighting the strengths of the work along with a couple of the interesting future directions inspired by the reported discoveries. The restricted presence of genes encoding SDG-1 and its paralogs within retrotransposons suggests intriguing evolutionary roles for these proteins. Future work could examine whether such fast-evolving or newly evolved proteins with potential roles in RNA regulation are more broadly associated with retrotransposons. Multiple SID-1-dependent proteins (including SDG-1 and SDG-2) could act together to mediate downstream effects. This possibility can be tested using combinatorial knockouts and overexpression strains. Both future directions have the potential to illuminate the evolutionarily selected roles of dsRNA-mediated signaling through SID-1, which remain a mystery.

**Reviewer #2 (Public review):**
Summary:RNAs can function across cell borders and animal generations as sources of epigenetic information for development and immunity. The specific mechanistic pathways how RNA travels between cells and progeny remains an open question. Here, Shugarts, et al. use molecular genetics, imaging, and genomics methods to dissect specific RNA transport and regulatory pathways in the *C. elegans* model system. Larvae ingesting double-stranded RNA is noted to not cause continuous gene silencing throughout adulthood. Damage of neuronal cells expressing double-stranded target RNA is observed to repress target gene expression in the germline. Exogenous short or long double-stranded RNA required different genes for entry into progeny. It was observed that the SID-1 double-stranded RNA transporter showed different expression over animal development. Removal of the sid-1 gene caused upregulation of two genes, the newly described sid-1-dependent gene sdg-1 and sdg-2. Both genes were observed to be negatively regulated by other small RNA regulatory pathways. Strikingly, loss then gain of sid-1 through breeding still caused variability of sdg-1 expression for many, many generations. SDG-2 protein co-localizes with germ granules, intracellular sites for heritable RNA silencing machinery. Collectively, sdg-1 presents a model to study how extracellular RNAs can buffer gene expression in germ cells and other tissues.Strengths:(1) Very cleaver molecular genetic methods and genomic analyses, paired with thorough genetics, were employed to discover insights into RNA transport, sdg-1 and sdg-2 as sid-1-dependent genes, and sdg-1's molecular phenotype.(2) The manuscript is well cited, and figures reasonably designed.(3) The discovery of the sdg genes being responsive to the extracellular RNA cell import machinery provides a model to study how exogenous somatic RNA is used to regulate gene expression in progeny. The discovery of genes within retrotransposons stimulates tantalizing models how regulatory loops may actually permit the genetic survival of harmful elements.Weaknesses:(1) The manuscript is broad, making it challenging to read and consider the data presented. Of note, since the original submission, the authors have improved the clarity of the writing and presentation.Comments on revised version:This reviewer thanks the authors for their efforts in revising the manuscript. In their rebuttal, the authors acknowledged the broad scope of their manuscript. I concur. While I still think the manuscript is a challenge to read due to its expansive nature, the current draft is substantially improved when compared to the previous one. This work will contribute to our general knowledge of RNA biology, small RNA regulatory pathways, and RNA inheritance.

We thank the reviewer for highlighting the strengths of the manuscript and for helping us improve the presentation of our results and discussion.

The following is the authors’ response to the original reviews.

**Public Reviews:**

**Reviewer #1 (Public review):**
Summary:In the manuscript "Intergenerational transport of double-stranded RNA limits heritable epigenetic changes" Shugarts and colleagues investigate intergenerational dsRNA transport in the nematode *C. elegans*. They induce oxidative damage in worms, blocking dsRNA import into cells (and potentially affecting the worms in other ways). Oxidative stress inhibits dsRNA import and the associated heritable regulation of gene expression in the adult germline (Fig. 2). The authors identify a novel gene, sid-1-dependent gene-1 (sdg-1), which is induced upon inhibition of SID-1 (Fig. 3). Both transient inhibition and genetic depletion of SID-1 lead to the upregulation of sdg-1 and a second gene, sdg-2 (Fig. 5). The expression of SDG-1 is variable, potentially indicating buffering regulation. While the expression of Sdg-1 could be consistent with a role in intergenerational transport of dsRNA, neither its overexpression nor loss-of-function impacts dsRNA-mediated silencing (Fig. 7) in the germline. It would be interesting to test if sdg-2 functions redundantly.In summary, the authors have identified a novel worm-specific protein (sdg-1) that is induced upon loss of dsRNA import via SID-1, but is not required to mediate SID-1 RNA regulatory effects.

We thank the reviewer for highlighting our findings on SDG-1. We found that oxidative damage in neurons enhanced dsRNA transport into the germline and/or subsequent silencing.

Remaining Questions:• The authors use an experimental system that induces oxidative damage specifically in neurons to release dsRNAs into the circulation. Would the same effect be observed if oxidative damage were induced in other cell types?

It is possible that oxidative damage of other tissues using miniSOG (as demonstrated in Xu and Chisholm, 2016) could also enhance the release of dsRNA into the circulation from those tissues. However, future experiments would be needed to test this empirically because it is also possible that the release of dsRNA depends on physiological properties (e.g., the molecular machinery promoting specific secretion) that are particularly active in neurons. We chose to use neurons as the source of dsRNA because by expressing dsRNA in a variety of tissues, neurons appeared to be the most efficient at the export of dsRNA as measured using SID-1-dependent silencing in other tissues (Jose et al., PNAS, 2009).

• Besides dsRNA, which other RNAs and cellular products (macromolecules and small signalling molecules) are released into the circulation that could affect the observed changes in germ cells?

We do not yet know all the factors that could be released either in naive animals or upon oxidative damage of neurons that influence the uptake of dsRNA into other tissues. The dependence on SID-1 for the observed enhancement of silencing (Fig. 2) shows that dsRNA is necessary for silencing within the germline. Whether this import of dsRNA occurs in conjunction with other factors (e.g., the uptake of short dsRNA along with yolk into oocytes [Marré et al., PNAS, 2016]) before silencing within the germline will require further study. A possible approach could be the isolation of extracellular fluid (Banse and Hunter, J Vis Exp., 2012) followed by characterization of its contents. However, the limited material available using this approach and the difficulty in avoiding contamination from cellular damage by the needle used for isolating the material make it challenging.

• SID-1 modifies RNA regulation within the germline (Fig. 7) and upregulates sdg-1 and sdg-2 (Fig. 5). However, SID-1's effects do not appear to be mediated via sdg-1. Testing the role of sdg-2 would be intriguing.

We observe the accumulation of *sdg-1* and *sdg-2* RNA in two different mutants lacking SID-1, which led us to conservatively focus on the analysis of one of these proteins for this initial paper. We expect that more sensitive analyses of the RNA-seq data will likely reveal additional genes regulated by SID-1. With the ability to perform multiplexed genome-editing, we hope in future work to generate strains that have mutations in many SID-1-dependent genes to recapitulate the defects observed in *sid-1(-)* animals. Indeed, as surmised by the reviewer, we are focusing on *sdg-2* as the first such SID-1-dependent gene to analyze using mutant combinations.

• Are sdg-1 or sdg-2 conserved in other nematodes or potentially in other species? appears to be encoded or captured by a retro-element in the *C. elegans* genome and exhibits stochastic expression in different isolates. Is this a recent adaptation in the *C. elegans* genome, or is it present in other nematodes? Does loss-of-function of sdg-1 or sdg-2 have any observable effect?

Clear homologs of SDG-1 and SDG-2 are not detectable outside of *C. elegans*. Consistent with the location of the *sdg-1* gene within a *Cer9* retrotransposon that appears to have integrated only within the *C. elegans* genome, sequence conservation between the genomes of related species is only observed outside the region of the retrotransposon (see Author response image 1, screenshot from UCSC browser). There were no obvious defects detected in animals lacking *sdg-1* (Fig. 7) or in animals lacking *sdg-2* (data not shown). It is possible that further exploration of both mutants and mutant combinations lacking additional SID-1-dependent genes would reveal defects. We also plan to examine these mutants in sensitized genetic backgrounds where one or more members of the RNA silencing pathway have been compromised.

Clarification for Readability:To enhance readability and avoid misunderstandings, it is crucial to specify the model organism and its specific dsRNA pathways that are not conserved in vertebrates:

We agree with the reviewer and thank the reviewer for the specific suggestions provided below. To take the spirit of the suggestion to heart we have instead changed the title of our paper to clearly signal that the entire study only uses *C. elegans*. We have titled the study ‘Intergenerational transport of double-stranded RNA in *C. elegans* can limit heritable epigenetic changes’

• In the first sentence of the paragraph "Here, we dissect the intergenerational transport of extracellular dsRNA ...", the authors should specify "in the nematode *C. elegans*". Unlike vertebrates, which recognise dsRNA as a foreign threat, worms and other invertebrates pervasively use dsRNA for signalling. Additionally, worms, unlike vertebrates and insects, encode RNA-dependent RNA polymerases that generate dsRNA from ssRNA substrates, enabling amplification of small RNA production. Especially in dsRNA biology, specifying the model organism is essential to avoid confusion about potential effects in humans.

We agree with most statements made by the reviewer, although whether dsRNA is exclusively recognized as a foreign threat by all vertebrates of all stages remains controversial. Our changed title now eliminates all ambiguity regarding the organism used in the study.

• Similarly, the authors should specify "in *C. elegans*" in the sentence "Therefore, we propose that the import of extracellular dsRNA into the germline tunes intracellular pathways that cause heritable RNA silencing." This is important because *C. elegans* small RNA pathways differ significantly from those in other organisms, particularly in the PIWI-interacting RNA (piRNA) pathways, which depend on dsRNA in *C. elegans* but uses ssRNA in vertebrates. Specification is crucial to prevent misinterpretation by the reader. It is well understood that mechanisms of transgenerational inheritance that operate in nematodes or plants are not conserved in mammals.

The piRNAs of *C. elegans* are single-stranded but are encoded by numerous independent genes throughout the genome. The molecules used for transgenerational inheritance of epigenetic changes that have been identified thus far are indeed different in different organisms. However, the regulatory principles required for transgenerational inheritance are general (Jose, *eLife*, 2024). Nevertheless, we have modified the title to clearly state that the entire study is using *C. elegans*.

• The first sentence of the discussion, "Our analyses suggest a model for ...", would also benefit from specifying "in *C. elegans*". The same applies to the figure captions. Clarification of the model organism should be added to the first sentence, especially in Figure 1.

With the clarification of the organism used in the title, we expect that all readers will be able to unambiguously interpret our results and the contexts where they apply.

**Reviewer #2 (Public review):**
Summary:RNAs can function across cell borders and animal generations as sources of epigenetic information for development and immunity. The specific mechanistic pathways how RNA travels between cells and progeny remains an open question. Here, Shugarts, et al. use molecular genetics, imaging, and genomics methods to dissect specific RNA transport and regulatory pathways in the *C. elegans* model system. Larvae ingesting double stranded RNA is noted to not cause continuous gene silencing throughout adulthood. Damage of neuronal cells expressing double stranded target RNA is observed to repress target gene expression in the germline. Exogenous supply of short or long double stranded RNA required different genes for entry into progeny. It was observed that the SID-1 double-stranded RNA transporter showed different expression over animal development. Removal of the sid-1 gene caused upregulation of two genes, the newly described sid-1-dependent gene sdg-1 and sdg-2. Both genes were observed to also be negatively regulated by other small RNA regulatory pathways. Strikingly, loss then gain of sid-1 through breeding still caused variability of sdg-1 expression for many, many generations. SDG-2 protein co-localizes with a Z-granule marker, an intracellular site for heritable RNA silencing machinery. Collectively, sdg-1 presents a model to study how extracellular RNAs can buffer gene expression in germ cells and other tissues.

We thank the reviewer for highlighting our findings and underscoring the striking nature of the discovery that mutating *sid-1* using genome-editing resulted in a transgenerational change that could not be reversed by changing the *sid-1* sequence back to wild-type.

Strengths:(1) Very clever molecular genetic methods and genomic analyses, paired with thorough genetics, were employed to discover insights into RNA transport, sdg-1 and sdg-2 as sid-1-dependent genes, and sdg-1's molecular phenotype.(2) The manuscript is well cited, and figures reasonably designed.(3) The discovery of the sdg genes being responsive to the extracellular RNA cell import machinery provides a model to study how exogenous somatic RNA is used to regulate gene expression in progeny. The discovery of genes within retrotransposons stimulates tantalizing models how regulatory loops may actually permit the genetic survival of harmful elements.

We thank the reviewer for the positive comments.

Weaknesses:(1) As presented, the manuscript is incredibly broad, making it challenging to read and consider the data presented. This concern is exemplified in the model figure, that requires two diagrams to summarize the claims made by the manuscript.

RNA interference (RNAi) by dsRNA is an organismal response where the delivery of dsRNA into the cytosol of some cell precedes the processing and ultimate silencing of the target gene within that cell. These two major steps are often not separately considered when explaining observations. Yet, the interpretation of every RNAi experiment is affected by both steps. To make the details that we have revealed in this work for both steps clearer, we presented the two models separated by scale - organismal vs. intracellular. We agree that this integrative manuscript appears very broad when the many different findings are each considered separately. The overall model revealed here forms the necessary foundation for the deep analysis of individual aspects in the future.

(2) The large scope of the manuscript denies space to further probe some of the ideas proposed. The first part of the manuscript, particularly Figures 1 and 2, presents data that can be caused by multiple mechanisms, some of which the authors describe in the results but do not test further. Thus, portions of the results text come across as claims that are not supported by the data presented.

We agree that one of the consequences of addressing the joint roles of transport and subsequent silencing during RNAi is that the scope of the manuscript appears large. We had suggested multiple interpretations for specific observations in keeping with the need for further work. To avoid any misunderstandings that our listing of possible interpretations be taken as claims by the reader, we have followed the instructions of the reviewer (see below) and moved some of the potential explanations we raised to the discussion section.

(3) The manuscript focuses on the genetics of SDGs but not the proteins themselves. Few descriptions of the SDGs functions are provided nor is it clarified why only SDG-1 was pursued in imaging and genetic experiments. Additionally, the SDG-1 imaging experiments could use additional localization controls.

We agree that more work on the SDG proteins will likely be informative, but are beyond the scope of this already expansive paper. We began with the analysis of SDG-1 because it had the most support as a regulator of RNA silencing (Fig. 5f). Indeed, in other work (Lalit and Jose, bioRxiv, 2024), we find that AlphaFold 2 predicts the SDG-1 protein to be a regulator of RNA silencing that directly interacts with the dsRNA-editing enzyme ADR-2 and the endonuclease RDE-8. Furthermore, we expect that more sensitive analyses of the RNA-seq data are likely to reveal additional genes regulated by SID-1. Using multiplexed genome editing, we hope to generate mutant combinations lacking multiple *sdg* genes to reveal their function(s).

We agree that given the recent discovery of many components of germ granules, our imaging data does not have sufficient resolution to discriminate between them. We have modified our statements and our model regarding the colocalization of SDG-1 with Z-granules to indicate that the overlapping enrichment of SDG-1 and ZNFX-1 in the perinuclear region is consistent with interactions with other nearby granule components.

**Recommendations for the authors:**

**Reviewer #2 (Recommendations for the authors):**
Major(1) As presented, the manuscript is almost two manuscripts combined into one. This point is highlighted in Figure 7h, which basically presents two separate models. The key questions addressed in the manuscript starts at Figure 3. Figures 1 and 2 are interesting observations but require more experiments to define further. For example, as the Results text describes for Figure 1, "These differences in the entry of ingested dsRNA into cells and/or subsequent silencing could be driven by a variety of changes during development. These include changes in the uptake of dsRNA into the intestine, distribution of dsRNA to other tissues from the intestine, import of dsRNA into the germline, and availability of RNA silencing factors within the germline." Presenting these (reasonable) mechanistic ideas detracted from the heritable RNA epigenetic mechanism explored in the later portion of the manuscript. There are many ways to address this issue, one being moving Figures 1 and 2 to the Supplement to focus on SID-1 related pathways.

Since this manuscript addresses the interaction between intercellular transport of dsRNA and heritable epigenetic changes, it was necessary to establish the possible route(s) that dsRNA could take to the germline before any inference could be made regarding heritable epigenetic changes. As suggested below (pt. 2), we have now moved the alternatives we enumerated as possible explanations for some experimental results (e.g., for the differences quoted here) to the discussion section.

(2) The manuscript includes detailed potential interpretations in the Results, making them seem like claims. Here is an example:"Thus, one possibility suggested by these observations is that reduction of sdg-1 RNA via SID-1 alters the amount of SDG-1 protein, which could interact with components of germ granules to mediate RNA regulation within the germline of wild-type animals."This mechanism is a possibility, but placing these ideas in the citable results makes it seem like an overinterpretation of imaging data. This text and others should be in the Discussion, where speculation is encouraged. Results sections like this example and others should be moved to the discussion.

We have rephrased motivating connections between experiments like the one quoted above and also moved such text to the discussion section wherever possible.

(3) A paragraph describing the SDG proteins will be helpful. Homologs? Conserved protein domains? mRNA and/or protein expression pattern across worm, not just the germline? Conservation across Caenorhabditis sp? These descriptions may help establish context why SDG-1 localizes to Z-granules.

We have now added information about the conservation of the sdg-1 gene in the manuscript. AlphaFold predicts domains with low confidence for the SDG-1 protein, consistent with the lack of conservation of this protein (AlphaFold requires multiple sequence alignments to predict confidently). In the adult animal, the SDG-1 protein was only detectable in the germline. Future work focused on SDG-1, SDG-2 and other SDG proteins will further examine possible expression in other tissues and functional domains if any. Unfortunately, in multiple attempts of single-molecule FISH experiments using probes against the sdg-1 open reading frame, we were unable to detect a specific signal above background (data not shown). Additional experiments are needed for the sensitive detection of sdg-1 expression outside the germline, if any.

(4) Based on the images shown, SDG-1 could be in other nearby granules, such as P granules or mutator foci. Additional imaging controls to rule out these granules/condensates will greatly strengthen the argument that SDG-1 protein localizes to Z-granules specifically.

We have modified the final model to indicate that the perinuclear colocalization is with germ granules broadly and we agree that we do not have the resolution to claim that the observed overlap of SDG-1::mCherry with GFP::ZNFX-1 that we detect using Airyscan microscopy is specifically with Z granules. Our initial emphasis of Z-granule was based on the prior report of SDG-1 being co-immunoprecipitated with the Z-granule surface protein PID-2/ZSP-1. However, through other work predicting possible direct interactions using AlphaFold (Lalit and Jose, bioRxiv, 2024), we were unable to detect any direct interactions between PID-2 and SDG-1. Indeed, many additional granules have been recently reported (Chen et al., Nat. Commun., 2024; Huang et al., bioRxiv 2024), making it possible that SDG-1 has specific interactions with a component of one of the other granules (P, Z, M, S, E, or D) or adjacent P bodies.

Minor(1) "This entry into the cytosol is distinct from and can follow the uptake of dsRNA into cells, which can rely on other receptors." Awkard sentence. Please revise.

We have now revised this sentence to read “This entry into the cytosol is distinct from the uptake of dsRNA into cells, which can rely on other receptors”

(2) Presumably, the dsRNA percent of the in vitro transcribed RNA is different than the 50 bp oligos that can be reliably annealed by heating and cooling. Other RNA secondary structure possibilities warrant further discussion.

We agree that in vitro transcribed RNA could include a variety of undefined secondary structures in addition to dsRNAs of mixed length. Such structures could recruit or titrate away RNA-binding proteins in addition to the dsRNA structures engaging the canonical RNAi pathway, resulting in mixed mechanisms of silencing. Future work identifying such structures and exploring their impact on the efficacy of RNAi could be informative. We have now added these considerations to the discussion and thank the reviewer for highlighting these possibilities.